# The Smoothed Possibility of Social Choice

**Lirong Xia, RPI, xialirong@gmail.com**

## Abstract

We develop a framework that leverages the smoothed complexity analysis by Spielman and Teng [60] to circumvent paradoxes and impossibility theorems in social choice, motivated by modern applications of social choice powered by AI and ML. For Condrocet's paradox, we prove that the smoothed likelihood of the paradox either vanishes at an exponential rate as the number of agents increases, or does not vanish at all. For the ANR impossibility on the non-existence of voting rules that simultaneously satisfy anonymity, neutrality, and resolvability, we characterize the rate for the impossibility to vanish, to be either polynomially fast or exponentially fast. We also propose a novel easy-to-compute tie-breaking mechanism that optimally preserves anonymity and neutrality for even number of alternatives in natural settings. Our results illustrate the smoothed possibility of social choice—even though the paradox and the impossibility theorem hold in the worst case, they may not be a big concern in practice.

## 1 Introduction

Dealing with paradoxes and impossibility theorems is a major challenge in social choice theory, because "*the force and widespread presence of impossibility results generated a consolidated sense of pessimism, and this became a dominant theme in welfare economics and social choice theory in general*", as the eminent economist Amartya Sen commented in his Nobel prize lecture [57].

Many paradoxes and impossibility theorems in social choice are based on worst-case analysis. Take perhaps the earliest one, namely *Condorcet's (voting) paradox* [15], for example. Condorcet's paradox states that, when there are at least three alternatives, it is impossible for pairwise majority aggregation to be transitive. The proof is done by explicitly constructing a worst-case scenario—a profile $P$ that contains a *Condorcet cycle*. For example, in $P = \{a \succ b \succ c, b \succ c \succ a, c \succ a \succ b\}$, there is a cycle $a \succ b$, $b \succ c$, and $c \succ a$ of pairwise majority. Condorcet's paradox is closely related to the celebrated Arrow's impossibility theorem: if Condorcet's paradox can be avoided, then the pairwise majority rule can avoid Arrow's impossibility theorem.

As another example, the *ANR impossibility theorem* (e.g. [49, Problem 1] and [53, 22, 12]) states that no voting rule $r$ can simultaneously satisfy anonymity ($r$ is insensitive to the identities of agents) and neutrality ($r$ is insensitive to the identities of alternatives), and resolvability ($r$ always chooses a single winner). The proof is done by analyzing a worst-case scenario $P = \{a \succ b, b \succ a\}$. Suppose for the sake of contradiction that a resolvable $r$ satisfies anonymity and neutrality, and without loss of generality let $r(P) = a$. After exchanging $a$ and $b$, the winner ought to be $b$ due to neutrality. But since the permuted profile still contains one vote for $a \succ b$ and one vote for $b \succ a$, the winner ought to be $a$ due to anonymity, which is a contradiction.

There is an enormous literature in social choice on circumventing the impossibilities, most of which belongs to the following two approaches. *(1) Domain restrictions*, namely, agents' reported preferences are assumed to come from a subset of all linear orders such as single-peaked preferences [6, 2, 58, 48, 16, 8, 24]; and *(2) likelihood analysis*, where impossibility theorems are evaluated by the likelihood of their occurrence in profiles randomly generated from a distribution such as the i.i.d. uniform distribution, a.k.a. *Impartial Culture (IC)* [28, 32]. Both approaches have been

criticized for making strong and unrealistic assumptions on the domain and on the probability distributions, respectively. In particular, IC has received much criticism, see e.g. [19], yet no widely-accepted probabilisitic model exists to the best of our knowledge.

The worst-case nature behind the impossibility theorems might be desirable for high-stakes, less-frequent applications such as political elections, but it may not be appropriate for modern low-stakes, frequently-used applications of social choice, many of which are supported by AI systems that learn agents' preferences to help them make group decisions [66]. While AI-powered social choice appears to be a promising solution to the long-standing turnout problem [57] and can therefore promote democracy to a larger scale and with a higher frequency, it questions the relevance of worst-case analysis in social choice theory. This motivates us to ask the following key question:

**How serious are the impossibilities in frequently-used modern applications of social choice?**

The frequently-used feature naturally leads to the analysis of average likelihood of the impossibility theorems. But in light of the criticism of the classical likelihood analysis approach discussed above, what is a realistic model to answer the key question?

Interestingly, computer science has encountered a similar challenge and has gone through a similar path in the analysis of practical performance of algorithms. Initially, the analysis mostly focused on the worst case, as in the spirit of big $O$ notation and NP-hardness. Domain restrictions have been a popular approach beyond the worst-case analysis. For example, while SAT is NP-hard, its restriction 2-SAT is in P. Likelihood analysis, in particular average-case complexity analysis [7], has also been a popular approach, yet it suffers from the same criticism as its counterpart in social choice—the distribution used in the analysis may well be unrealistic [61].

The challenge was addressed by the *smoothed (complexity) analysis* introduced by Spielman and Teng [60], which focuses on the "worst average-case" scenario that combines the worst-case analysis and the average-case analysis. The idea is based on the fact that the input $\vec{x}$ of an algorithm is often a noisy perception of the ground truth $\vec{x}^*$. Therefore, the worst-case is analyzed by assuming that an adversary chooses a ground truth $\vec{x}^*$ and then Nature adds a noise $\vec{\epsilon}$ (e.g. a Gaussian noise) to it, such that the algorithm's input becomes $\vec{x} = \vec{x}^* + \vec{\epsilon}$. The smoothed runtime of an algorithm is therefore $\sup_{\vec{x}^*} \mathbb{E}_{\vec{\epsilon}} \text{RunTime}(\vec{x}^* + \vec{\epsilon})$, in contrast to the worst-case runtime $\sup_{\vec{x}^*} \text{RunTime}(\vec{x}^*)$ and the average-case runtime $\mathbb{E}_{\vec{x}^* \sim \pi} \text{RunTime}(\vec{x}^*)$, where $\pi$ is a given distribution over data.

**Our Contributions.** We propose a framework that leverages the elegant smoothed complexity analysis to answer the key question above. In social choice, the data is a *profile*, which consists of agents' reported preferences that are often represented by linear orders over a set $\mathcal{A}$ of $m$ alternatives. Like in the smoothed complexity analysis, in our framework there is an adversary who controls agents' "ground truth" preferences, which may be ordinal (as rankings over alternatives) or cardinal (as utilities over alternatives). Then, Nature adds a "noise" to the ground truth preferences and outputs a preference profile, which consists of linear orders over alternatives.

Following the convention in average-case complexity analysis [7], we use a statistical model to model Nature's noising procedure. As in many smoothed-analysis approaches, we assume that noises in agents' preferences are independently generated, yet agents' ground truth preferences can be arbitrarily correlated, which constitutes the basis for the worst-case analysis. Using our smoothed analysis framework, we obtain the following two dichotomy theorems on the asymptotic smoothed likelihood of Condorcet's paradox and the ANR impossibility under mild assumptions, when the number of alternatives $m$ is fixed and the number of agents $n$ goes to infinity.

THEOREM 1. **(Smoothed Condorcet's paradox, informally put).** *The smoothed likelihood of Condorcet's Paradox either vanishes at an exponential rate, or does not vanish at all.*

THEOREM 2. **(Smoothed ANR (im)possibility theorem, informally put).** *The theorem has two parts. The* **smoothed possibility** *part states that there exist resolute voting rules under which the impossibility theorem either vanishes at an exponential rate or at a polynomial rate. The* **smoothed impossibility part** *states that there does not exists a resolute voting rule under which the impossibility theorem vanishes faster than under the rules in the smoothed possibility part.*

Both theorems are quite general and their formal statements also characterize conditions for each case. Such conditions in Theorem 1 tell us when Condorcet's Paradox vanishes (at an exponential rate), which is positive. While the theorem may be expected at a high level and part of it is easy to

prove, for example the exponential-rate part can be proved by using a similar idea as in the proof of minimaxity/sample complexity of MLE under a large class of distance-based models [13], we are not aware of a previous work that provides a *complete dichotomy* that draws a clear line between paradoxes and non-paradoxes as Theorem 1 does. In addition, we view such expectedness positive news, because it provides a theoretical confirmation of well-believed hypotheses under natural settings, as smoothed complexity analysis did for the runtime of a simplex algorithm.

The smoothed possibility part of Theorem 2 is also positive because it states that the ANR impossibility vanishes as the number of agents $n$ increases. The smoothed impossibility part of Theorem 2 is mildly negative, because it states that no voting rule can do better, though the impossibility may still vanish as $n$ increases. Together, Theorem 1 and 2 illustrate the smoothed possibility of social choice—even though the paradox and the impossibility theorem hold in the worst case, they may not be a big concern in practice in some natural settings.

Our framework also allows us to develop a novel easy-to-compute tie-breaking mechanism called *most popular singleton ranking (MPSR)* tie-breaking, which tries to break ties using a linear order that uniquely occurs most often in the profile (Definition 8). We prove that MPSR is better than the commonly-used lexicographic tie-breaking and fixed-agent tie-breaking mechanisms w.r.t. the smoothed likelihood of the ANR impossibility—MPSR reduces the smoothed likelihood from $n^{-0.5}$ to $n^{-\frac{m!}{4}}$ for many commonly-studied voting rules under natural assumptions (Proposition 1 and Theorem 3), and is optimal for even number of alternatives $m$ (Theorem 2 and Lemma 2).

**Proof Techniques.** Standard approximation techniques such as Berry-Esseen theorem and its high-dimensional counterparts, e.g. [64, 18, 21], due to their $O(n^{-0.5})$ error terms, are too coarse for the (tight) bound in Theorem 2. To prove our theorems, we first model various events of interest as systems of linear constraints. Then, we develop a technical tool (Lemma 1) to provide a dichotomy characterization for the *Poisson Multinomial Variables (PMV)* that corresponds to the histogram of a randomly generated profile to satisfy the constraints. We further show in Appendix I that Lemma 1 is a general and useful tool for analyzing smoothed likelihood of many other commonly-studied events in social choice (Table 4), which are otherwise hard to analyze.

## 1.1 Related Work and Discussions

**Smoothed analysis.** Smoothed analysis has been applied to a wide range of problems in mathematical programming, machine learning, numerical analysis, discrete math, combinatorial optimization, and equilibrium analysis and price of anarchy [14], see [61] for a survey. In a recent position paper, Baumeister et al. [4] proposed to conduct smoothed analysis on computational aspects of social choice and mentioned that their model can be used to analyze voting paradoxes and ties, but the paper does not contain technical results. Without knowing their work, we independently proposed and formulated the smoothed analysis framework for social choice in this paper.

**The worst average-case idea.** While our framework is inspired by the smoothed complexity analysis, the worst average-case idea is deeply rooted in (frequentist) statistics and can be viewed as a measure of robustness. Taking a statistical decision theory [5] point of view, the frequentist's loss of a decision rule $r$ : Data $\rightarrow$ Decision under a statistical model $\mathcal{M} = (\Theta, \mathcal{S}, \Pi)$ is measured by

$$\sup_{\theta \in \Theta} \mathbb{E}_{P \sim \pi_\theta} (\text{Loss}(\theta, r(P))),$$

where the expectation evaluates the average-case loss under the worst-case distribution $\pi_\theta \in \Pi$. There is a large literature in statistical aspects of social choice (e.g. [15, 13, 65]) and preference learning (e.g. [39, 59]) that study the frequentist loss w.r.t. classical loss functions in statistics that depend on both $P$ and $\theta$, leading to consistency and minimaxity results. The idea is also closely related to the "min of means" criteria in decision theory [31]. Our framework explicitly models smoothed likelihood of social choice events via loss functions that measure the dissatisfaction of axioms w.r.t. the data (profile) and do not depend on the "ground truth" $\theta$. This is similar to smoothed complexity analysis, where the loss function is the runtime of an algorithm, which also only depends on the input data $P$ but not on $\theta$.

**Correlations among agents' preferences.** In our model, agents' ground truth preferences can be arbitrarily correlated while the randomness comes from independent noises. This is a standard assumption in smoothed complexity analysis as well as in relevant literatures in psychology, economics, and behavioral science, as evident in random utility models, logistic regression, MLE inter-

pretation of the ordinary least squares method, etc. [62, 67]. As another justification, the adversary can be seen as a manipulator who wants to control agents' reported preferences, but is only able to do it in a probabilistic way.

**Generality of results and techniques.** Our technical results are quite general and can be immediately applied to classical likelihood analysis in social choice under i.i.d. distribution, to answer open questions, obtain new results, and provide new insights. For example, a straightforward application of Lemma 1 gives an asymptotic answer to an open question by Tsetlin et al. [63] (after Corollary 2 in Appendix I). As another example, we are not aware of a previous work on the asymptotic likelihood of the ANR impossibility even under IC, which is a special case of Theorem 2.

**Other related work.** As discussed above, there is a large literature on domain restrictions and the likelihood analysis toward circumventing impossibility theorems, see for example, the book by Gehrlein and Lepelley [30] for a recent survey. In particular, there is a large literature on the likelihood of Condorcet voting paradox and the likelihood of (non)-existence of Condorcet winner under i.i.d. distributions especially IC [20, 43, 29, 63, 35, 32, 10, 11]. The IC assumption has also been used to prove quantitative versions of other impossibility theorems in social choice such as Arrow's impossibility theorem [36, 37, 46] and Gibbard-Satterthwaite theorem [26, 47], as well as in judgement aggregation [51, 25]. Other works have studied social choice problems when each agent's preferences are represented by a probability distribution [3, 33, 55, 68, 52, 40]. These works focused on computing the outcome efficiently, which is quite different from our goal.

## 2   Preliminaries

**Basic Setting.** Let $\mathcal{A} = [m] = \{1, \ldots, m\}$ denote the set of $m \geq 3$ *alternatives*. Let $\mathcal{L}(\mathcal{A})$ denote the set of all linear orders (a.k.a. rankings) over $\mathcal{A}$. Let $n \in \mathbb{N}$ denote the number of agents. Each agent uses a linear order to represent his or her preferences. The vector of $n \in \mathbb{N}$ agents' votes $P$ is called a *(preference) profile*, or sometimes an $n$-profile. For any profile $P$, let $\text{Hist}(P) \in \mathbb{Z}_{\geq 0}^{m!}$ denote the anonymized profile of $P$, also called the *histogram* of $P$, which counts the multiplicity of each linear order in $P$. A *resolute voting rule* $r$ is a mapping from each profile to a single winner in $\mathcal{A}$. A *voting correspondence* $c$ is a mapping from each profile to a non-empty set of co-winners.

**Tie-Breaking Mechanisms.** Many commonly-studied voting rules are defined as correspondences combined with a tie-breaking mechanism. For example, a *positional scoring correspondence* is characterized by a scoring vector $\vec{s} = (s_1, \ldots, s_m)$ with $s_1 \geq s_2 \geq \cdots \geq s_m$ and $s_1 > s_m$. For any alternative $a$ and any linear order $R \in \mathcal{L}(\mathcal{A})$, we let $\vec{s}(R, a) = s_i$, where $i$ is the rank of $a$ in $R$. Given a profile $P$, the positional scoring correspondence $c_{\vec{s}}$ chooses all alternatives $a$ with maximum $\sum_{R \in P} \vec{s}(R, a)$. For example, *Plurality* uses the scoring vector $(1, 0, \ldots, 0)$ and *Borda* uses the scoring vector $(m - 1, m - 2, \ldots, 0)$. The positional scoring rule $r_{\vec{s}}$ chooses a single alternative by further applying a tie-breaking mechanism. The *lexicographic tie-breaking*, denoted by LEX-$R$ where $R \in \mathcal{L}(\mathcal{A})$, breaks ties in favor of alternatives ranked higher in $R$. The fixed-agent tie-breaking, denoted by FA-$j$ where $1 \leq j \leq n$, uses agent $j$'s preferences to break ties.

**(Un)weighted Majority Graphs.** For any profile $P$ and any pair of alternatives $a, b$, let $P[a \succ b]$ denote the number of rankings in $P$ where $a$ is preferred to $b$. Let $\text{WMG}(P)$ denote the *weighted majority graph* of $P$, which is a complete graph where the vertices are $\mathcal{A}$ and the edge weights are $w_P(a, b) = P[a \succ b] - P[b \succ a]$. For any distribution $\pi$ over $\mathcal{L}(\mathcal{A})$, let $\text{WMG}(\pi)$ denote the weighted majority graph where $\pi$ is treated as a *fractional* profile, where for each $R \in \mathcal{L}(\mathcal{A})$ there are $\pi(R)$ copies of $R$. The *unweighted majority graph (UMG)* of a profile $P$, denoted by $\text{UMG}(P)$, is the unweighted directed graph where the vertices are the alternatives and there is an edge $a \to b$ if and only if $P[a \succ b] > P[b \succ a]$. If $a$ and $b$ are tied, then there is no edge between $a$ and $b$. $\text{UMG}(\pi)$ is defined similarly. A *Condorcet cycle* of a profile $P$ is a cycle in $\text{UMG}(P)$. A *weak Condorcet cycle* of a profile $P$ is a cycle in any supergraph of $\text{UMG}(P)$.

**Axiomatic Properties.** A voting rule $r$ satisfies *anonymity*, if the winner is insensitive to the identity of the voters. That is, for any pair of profiles $P$ and $P'$ with $\text{Hist}(P) = \text{Hist}(P')$, we have $r(P) = r(P')$. $r$ satisfies *neutrality* if the winner is insensitive to the identity of the alternatives. That is, for any permutation $\sigma$ over $\mathcal{A}$, we have $r(\sigma(P)) = \sigma(r(P))$, where $\sigma(P)$ is the obtained from $P$ by permuting alternatives according to $\sigma$.

**Single-Agent Preference Models.** A statistical model $\mathcal{M} = (\Theta, \mathcal{S}, \Pi)$ has three components: the parameter space $\Theta$, which contains the "ground truth"; the sample space $\mathcal{S}$, which contains all possible data; and the set of probability distributions $\Pi$, which contains a distribution $\pi_\theta$ over $\mathcal{S}$ for each $\theta \in \Theta$. In this paper we adopt single-agent preference models, where $\mathcal{S} = \mathcal{L}(\mathcal{A})$.

**Definition 1.** *A* single-agent preference model *is denoted by* $\mathcal{M} = (\Theta, \mathcal{L}(\mathcal{A}), \Pi)$. *$\mathcal{M}$ is* strictly positive *if there exists $\epsilon > 0$ such that the probability of any ranking under any distribution in $\Pi$ is at least $\epsilon$. $\mathcal{M}$ is* closed *if $\Pi$ is a closed set in $\mathbb{R}_{\geq 0}^{m!}$, where each distribution in $\Pi$ is viewed as a vector in $m!$-probability simplex. $\mathcal{M}$ is* neutral *if for any $\theta \in \Theta$ and any permutation $\sigma$ over $\mathcal{A}$, there exists $\eta \in \Theta$ such that for all $R \in \mathcal{L}(\mathcal{A})$, we have $\pi_\theta(R) = \pi_\eta(\sigma(R))$.*

See Example 2 in Appendix A for two examples of single-agent preference models that correspond to the celebrated Mallows model and Plackett-Luce model, respectively.

## 3 Smoothed Analysis Framework and The Main Technical Lemma

Many commonly-studied axioms and events in social choice, denoted by $X$, are defined based on per-profile properties in the following way. Let $r$ denote a voting rule or correspondence and let $P$ denote a profile. There is a function $\mathrm{S}_X(r, P) \in \{0, 1\}$ that indicates whether $X$ holds for $r$ at $P$. Then, $r$ satisfies $X$ if $\forall P, \mathrm{S}_X(r, P) = 1$, or equivalently, $\inf_P \mathrm{S}_X(r, P) = 1$. For example, for *anonymity*, let $\mathrm{S}_{\mathrm{ano}}(r, P) = 1$ iff for all profiles $P'$ with $\mathrm{Hist}(P') = \mathrm{Hist}(P)$, $r(P') = r(P)$. For *neutrality*, let $\mathrm{S}_{\mathrm{neu}}(r, P) = 1$ iff for all permutation $\sigma$ over $\mathcal{A}$, $\sigma(r(P)) = r(\sigma(P))$. For *non-existence of Condorcet cycle*, let $\mathrm{S}_{\mathrm{NCC}}(P) = 1$ iff there is no Condorcet cycle in $P$.

Our smoothed analysis framework assumes that each of the $n$ agents' preferences are chosen from a single-agent preference model $\mathcal{M} = (\Theta, \mathcal{L}(\mathcal{A}), \Pi)$ by the adversary.

**Definition 2 (Smoothed likelihood of events).** *Given a single-agent preference model $\mathcal{M} = (\Theta, \mathcal{L}(\mathcal{A}), \Pi)$, $n \in \mathbb{N}$ agents, a function $\mathrm{S}_X$ that characterizes an axiom or event $X$, and a voting rule (or correspondence) $r$, the* smoothed likelihood *of $X$ is defined as $\inf_{\vec{\pi} \in \Pi^n} \mathbb{E}_{P \sim \vec{\pi}} \mathrm{S}_X(r, P)$.*

For example, $\inf_{\vec{\pi} \in \Pi^n} \mathbb{E}_{P \sim \vec{\pi}} \mathrm{S}_{\mathrm{NCC}}(P)$ is the smoothed likelihood of non-existence of Condorcet cycle, which corresponds to the avoidance of Condorcet's paradox. $\inf_{\vec{\pi} \in \Pi^n} \Pr_{P \sim \vec{\pi}}(\mathrm{S}_{\mathrm{ano}}(P) + \mathrm{S}_{\mathrm{neu}}(P) = 2)$ is the smoothed likelihood of satisfaction of anonymity+neutrality, which corresponds to the avoidance of the ANR impossibility.

We use the following simple example to show how to model events of interest as a system of linear constraints. The first event is closely related to the smoothed Condorcet's paradox (Theorem 1) and the second event is closely related to the smoothed ANR theorem (Theorem 2).

**Example 1.** *Let $m = 3$ and $\mathcal{A} = \{1, 2, 3\}$. For any profile $P$, let $x_{123}$ denote the number of $1 \succ 2 \succ 3$ in $P$. The event "there is a Cordorcet cycle $1 \rightarrow 2 \rightarrow 3 \rightarrow 1$" can be represented by:*

$$(x_{213} + x_{231} + x_{321}) - (x_{123} + x_{132} + x_{312}) < 0 \qquad (1)$$

$$(x_{312} + x_{321} + x_{132}) - (x_{231} + x_{213} + x_{123}) < 0 \qquad (2)$$

$$(x_{123} + x_{132} + x_{213}) - (x_{312} + x_{321} + x_{231}) < 0 \qquad (3)$$

*Equation (1) (respectively, (2) and (3)) states that $\mathrm{UMG}(P)$ has edge $1 \rightarrow 2$ (respectively, $2 \rightarrow 3$ and $3 \rightarrow 1$). As another example, the event "$\mathrm{Hist}(P)$ is invariant to the permutation $\sigma$ over $\mathcal{A}$ that exchanges 1 and 2" can be represented by $\{x_{123} - x_{213} = 0, x_{132} - x_{231} = 0, x_{312} - x_{321} = 0\}$.*

Notice that in this example each constraint has the form $\vec{E} \cdot \vec{x} = 0$ or $\vec{S} \cdot \vec{x} < 0$, where $\vec{E} \cdot \vec{1} = 0$ and $\vec{S} \cdot \vec{1} = 0$. More generally, our main technical lemma upper-bounds the smoothed likelihood for the Poisson multinomial variable $\mathrm{Hist}(P)$ to satisfy a similar system of linear inequalities below.

**Definition 3.** *Let $q, n \in \mathbb{N}$. For any vector of $n$ distributions $\vec{\pi} = (\pi_1, \ldots, \pi_n)$, each of which is over $[q]$, let $\vec{Y} = (Y_1, \ldots, Y_n)$ denote the vector of $n$ random variables distributed as $\pi_1, \ldots, \pi_n$, respectively, and let $\vec{X}_{\vec{\pi}} = \mathrm{Hist}(\vec{Y})$, i.e. the Poisson multinomial variable that corresponds to $\vec{Y}$.*

**Definition 4.** *Let $C^{\mathbf{ES}}(\vec{x}) = \{\mathbf{E} \cdot (\vec{x})^\top = (\vec{0})^\top \text{ and } \mathbf{S} \cdot (\vec{x})^\top < (\vec{0})^\top\}$, where $\mathbf{E}$ is a $K \times q$ integer matrix that represents the* equations *and $\mathbf{S}$ is an $L \times q$ integer matrix with $K + L \geq 1$ that represents the* strict inequalities. *Let $C_{\leq 0}^{\mathbf{ES}}(\vec{x}) = \{\mathbf{E} \cdot (\vec{x})^\top = (\vec{0})^\top \text{ and } \mathbf{S} \cdot (\vec{x})^\top \leq (\vec{0})^\top\}$ denote the relaxation of $C^{\mathbf{ES}}(\vec{x})$. Let $\mathcal{H}$ and $\mathcal{H}_{\leq 0}$ denote the solutions to $C^{\mathbf{ES}}(\vec{x})$ and $C_{\leq 0}^{\mathbf{ES}}(\vec{x})$, respectively.*

**Lemma 1 (Main technical lemma).** *Let $q \in \mathbb{N}$ and let $\Pi$ be a closed set of strictly positive distributions over $[q]$. Let $CH(\Pi)$ denote the convex hull of $\Pi$.*

**Upper bound.** *For any $n \in \mathbb{N}$ and any $\vec{\pi} \in \Pi^n$,*

$$\Pr\left( \vec{X}_{\vec{\pi}} \in \mathcal{H} \right) = \begin{cases} 0 & \text{if } \mathcal{H} = \emptyset \\ \exp(-\Omega(n)) & \text{if } \mathcal{H} \neq \emptyset \text{ and } \mathcal{H}_{\leq 0} \cap CH(\Pi) = \emptyset \\ O(n^{-\frac{Rank(\mathbf{E})}{2}}) & \text{if } \mathcal{H} \neq \emptyset \text{ and } \mathcal{H}_{\leq 0} \cap CH(\Pi) \neq \emptyset \end{cases}$$

**Tightness of the upper bound.** *There exists a constant $C$ such that for any $n' \in \mathbb{N}$, there exists $n' \leq n \leq Cn'$ and $\vec{\pi} \in \Pi^n$ such that*

$$\Pr\left( \vec{X}_{\vec{\pi}} \in \mathcal{H} \right) = \begin{cases} \exp(-O(n)) & \text{if } \mathcal{H} \neq \emptyset \text{ and } \mathcal{H}_{\leq 0} \cap CH(\Pi) = \emptyset \\ \Omega(n^{-\frac{Rank(\mathbf{E})}{2}}) & \text{if } \mathcal{H} \neq \emptyset \text{ and } \mathcal{H}_{\leq 0} \cap CH(\Pi) \neq \emptyset \end{cases}$$

Lemma 1 is quite general because the assumptions on $\Pi$ are mild and $\mathbf{E}$ and $\mathbf{S}$ are general enough to model a wide range of events in social choice as we will see later in the paper. It provides asymptotically tight upper bounds on the probability for the histogram of a randomly generated profile from $\vec{\pi} \in \Pi^n$ to satisfy all constraints in $C^{\mathbf{ES}}$. The bounds provide a trichotomy: if no vector satisfies all constraints in $C^{\mathbf{ES}}$, i.e. $\mathcal{H} = \emptyset$, then the upper bound is 0; otherwise if $\mathcal{H} \neq \emptyset$ and its relaxation $\mathcal{H}_{\leq 0}$ does not contain a vector in the convex hull of $\Pi$, then the upper bound is exponentially small; otherwise the upper bound is polynomially small in $n$, and the degree of polynomial is determined by the rank of $\mathbf{E}$, specifically $-\frac{\text{Rank}(\mathbf{E})}{2}$. The tightness part of the lemma states that the upper bounds cannot be improved for all $n$.

At a high level the lemma is quite natural and follows the intuition of multivariate central limit theorem as follows. Roughly, $\vec{X}_{\vec{\pi}}$ is distributed like a multinomial Gaussian whose expectation is $\vec{\pi} \cdot \vec{1} = \sum_{j=1}^{n} \pi_j$ (which is a $q$-dimensional vector). Then, the zero part of Lemma 1 is trivial; the exponential part makes sense because the expectation $\vec{\pi} \cdot \vec{1}$ is $\Theta(n)$ away from any vector in $\mathcal{H}$; and the last part is expected to be $O(n^{-0.5})$ because the center $\vec{\pi} \cdot \vec{1}$ satisfies the $\mathbf{E}$ part of $C^{\mathbf{ES}}$.

The surprising part of the lemma is the degree of polynomial $-\frac{\text{Rank}(\mathbf{E})}{2}$ and its tightness. As discussed in the Introduction, all central limit theorems we are aware of are too coarse for proving the $O(n^{-\frac{\text{Rank}(\mathbf{E})}{2}})$ bound. To prove the polynomial upper bound, we introduce an alternative representation of $\vec{X}_{\vec{\pi}}$ to tackle the dependencies among its components, prove novel fine-grained concentration and anti-concentration bounds, focus on the reduced row echelon form of $\mathbf{E}$ plus an additional constraint on the total number of agents to characterize $\mathcal{H}$, and then do a weighted counting of vectors that satisfy $C^{\mathbf{ES}}$. The full proof can be found in Appendix B.

## 4 Smoothed Condorcet's Paradox and ANR (Im)possibility Theorem

We first apply the main technical lemma (Lemma 1) to characterize the smoothed likelihood of Condorcet's paradox in the following dichotomy theorem, which holds for any fixed $m \geq 3$.

**Theorem 1 (Smoothed likelihood of Codorcet's paradox).** *Let $\mathcal{M} = (\Theta, \mathcal{L}(\mathcal{A}), \Pi)$ be a strictly positive and closed single-agent preference model.*

**Smoothed avoidance of Condorcet's paradox.** *Suppose for all $\pi \in CH(\Pi)$, $UMG(\pi)$ does not contain a weak Condorcet cycle. Then, for any $n \in \mathbb{N}$, we have:*

$$\inf_{\vec{\pi} \in \Pi^n} \mathbb{E}_{P \sim \vec{\pi}} S_{NCC}(P) = 1 - \exp(-\Omega(n))$$

**Smoothed Condorcet's paradox.** *Suppose there exists $\pi \in CH(\Pi)$ such that $UMG(\pi)$ contains a weak Condorcet cycle. Then, there exist infinitely many $n \in \mathbb{N}$ such that:*

$$\inf_{\vec{\pi} \in \Pi^n} \mathbb{E}_{P \sim \vec{\pi}} S_{NCC}(P) = 1 - \Omega(1)$$

The smoothed avoidance part of the theorem is positive news: if there is no weak Condorcet cycle in the UMG of any distribution in the convex hull of $\Pi$, then no matter how the adversary sets agents' ground truth preferences, the probability for Condorcet's paradox to hold, which is $1 - \inf_{\vec{\pi} \in \Pi^n} \mathbb{E}_{P \sim \vec{\pi}} S_{\text{NCC}}(P) = \sup_{\vec{\pi} \in \Pi^n} \Pr_{P \sim \vec{\pi}}(S_{\text{NCC}}(P) = 0)$, vanishes at an exponential rate as $n \to \infty$. Consequently, in such cases Arrows' impossibility theorem can be avoided because the pairwise

majority rule satisfies all desired properties mentioned in the theorem. The second part (smoothed paradox) states that otherwise the adversary can make Condorcet's paradox occur with constant probability. The proof is done by modeling $S_{\mathrm{NCC}}(P) = 0$ as systems of linear constraints as in Definition 4, each of which represents a target UMG with a weak Codorcet cycle as in Example 1, then applying Lemma 1. The full proof is in Appendix C.

We now turn to the smoothed ANR impossibility. We will reveal a relationship between all $n$-profiles and all permutation groups over $\mathcal{A}$ after recalling some basic notions in group theory. The *symmetric group* over $\mathcal{A} = [m]$, denoted by $\mathcal{S}_{\mathcal{A}}$, is the set of all permutations over $\mathcal{A}$.

**Definition 5.** *For any profile $P$, let $Perm(P)$ denote the set of all permutations $\sigma$ over $\mathcal{A}$ that maps $Hist(P)$ to itself. Formally, $Perm(P) = \{\sigma \in \mathcal{S}_{\mathcal{A}} : Hist(P) = \sigma(Hist(P))\}$.*

See Appendix D for additional notation and examples about group theory.[1] It is not hard to see that $Perm(P)$ is a permutation group. We now define a special type of permutation groups that "cover" all alternatives in $\mathcal{A}$, which are closely related to the impossibility theorem.

**Definition 6.** *For any permutation group $U \subseteq \mathcal{S}_{\mathcal{A}}$ and any alternative $a \in \mathcal{A}$, we say that $U$ covers $a$ if there exists $\sigma \in U$ such that $a \neq \sigma(a)$. We say that $U$ covers $\mathcal{A}$ if it covers all alternatives in $\mathcal{A}$. For any $m$, let $\mathcal{U}_m$ denote the set of all permutation groups that cover $\mathcal{A}$.*

For example, when $m = 3$, $\mathcal{U}_3 = \{\mathrm{Id}, (1, 2, 3), (1, 3, 2)\}$, where Id is the identity permutation and $(1, 2, 3)$ is the circular permutation $1 \rightarrow 2 \rightarrow 3 \rightarrow 1$. See Example 5 in Appendix D for the list of all permutation groups for $m = 3$. In general $|\mathcal{U}_m| > 1$.

**Theorem 2** (**Smoothed ANR (im)possibility**). *Let $\mathcal{M} = (\Theta, \mathcal{L}(\mathcal{A}), \Pi)$ be a strictly positive and closed single-agent preference model. Let $\mathcal{U}_m^{\Pi} = \{U \in \mathcal{U}_m : \exists \pi \in CH(\Pi), \forall \sigma \in U, \sigma(\pi) = \pi\}$, and when $\mathcal{U}_m^{\Pi} \neq \emptyset$, let $l_{\min} = \min_{U \in \mathcal{U}_m^{\Pi}} |U|$ and $l_{\Pi} = \frac{l_{\min} - 1}{l_{\min}} m!$.*

**Smoothed possibility.** *There exist an anonymous voting rule $r_{ano}$ and a neutral voting rule $r_{neu}$ such that for any $r \in \{r_{ano}, r_{neu}\}$, any $n$, and any $\vec{\pi} \in \Pi^n$, we have:*

$$\Pr_{P \sim \vec{\pi}}(S_{ano}(r, P) + S_{neu}(r, P) < 2) = \begin{cases} O(n^{-\frac{l_{\Pi}}{2}}) & \text{if } \mathcal{U}_m^{\Pi} \neq \emptyset \\ \exp(-\Omega(n)) & \text{otherwise} \end{cases}$$

**Smoothed impossibility.** *For any voting rule $r$, there exist infinitely many $n \in \mathbb{N}$ such that:*

$$\sup_{\vec{\pi} \in \Pi^n} \Pr_{P \sim \vec{\pi}}(S_{ano}(r, P) + S_{neu}(r, P) < 2) = \begin{cases} \Omega(n^{-\frac{l_{\Pi}}{2}}) & \text{if } \mathcal{U}_m^{\Pi} \neq \emptyset \\ \exp(-O(n)) & \text{otherwise} \end{cases}$$

Again, Theorem 2 holds for fixed $m \geq 3$. We note that for any profile $P$, $S_{\mathrm{ano}}(r, P) + S_{\mathrm{neu}}(r, P) < 2$ if and only if at least one of anonymity or neutrality is violated at $P$. In other words, if $S_{\mathrm{ano}}(r, P) + S_{\mathrm{neu}}(r, P) = 2$ then both anonymity and neutrality are satisfied at $P$. Therefore, the first part of Theorem 2 is called "smoothed possibility" because it states that no matter how the adversary sets agents' ground truth preferences, the probability for $r_{\mathrm{ano}}$ (respectively, $r_{\mathrm{neu}}$) to satisfy both anonymity and neutrality converges to 1. The second part (smoothed impossibility) shows that the rate of convergence in the first part is asymptotically tight for all $n$. This is a mild impossibility theorem because violations of anonymity or neutrality may still vanish (at a slower rate) as $n \rightarrow \infty$.

The proof proceeds in the following three steps. Step 1. For any $m$ and $n$, we define a set of profiles, denoted by $\mathcal{T}_{m,n}$, that represent the source of impossibility. In fact, $\mathcal{T}_{m,n}$ is the set of all $n$-profiles $P$ such that $Perm(P)$ covers $\mathcal{A}$, i.e. $Perm(P) \in \mathcal{U}_m$. Step 2. To prove the smoothed possibility part, we define $r_{\mathrm{ano}}$ and $r_{\mathrm{neu}}$ that satisfy both anonymity and neutrality for all profiles that are not in $\mathcal{T}_{m,n}$. Then, we apply Lemma 1 to upper-bound the probability of $\mathcal{T}_{m,n}$. Step 3. The smoothed impossibility part is proved by applying the tightness part of Lemma 1 to the probability of $\mathcal{T}_{m,n}$. The full proof can be found in Appendix E.

In general $l_{\min}$ in Theorem 2 can be hard to characterize. The following lemma provides a lower bound on $l_{\min}$ by characterizing $\min_{U \in \mathcal{U}_m} |U|$, whose group-theoretic proof is in Appendix F.

**Lemma 2.** *For any $m \geq 2$, let $l^* = \min_{U \in \mathcal{U}_m} |U|$. We have $l^* = 2$ if $m$ is even; $l^* = 3$ if $m$ is odd and $3 \mid m$; $l^* = 5$ if $m$ is odd, $3 \nmid m$, and $5 \mid m$; and $l^* = 6$ for other $m$.*

A notable special case of Theorem 2 is $\pi_{\text{uni}} \in \text{CH}(\Pi)$, where $\pi_{\text{uni}}$ is the uniform distribution over $\mathcal{L}(\mathcal{A})$. We note that for any permutation $\sigma$, $\pi_{\text{uni}} = \sigma(\pi_{\text{uni}})$, which means that $\mathcal{U}_m^\Pi = \mathcal{U}_m$. Therefore, only the polynomial bound in Theorem 2 remains, with $l_\Pi = \frac{l^*-1}{l^*}m!$. In particular, $\pi_{\text{uni}} \in \text{CH}(\Pi)$ for all neutral single-agent preference models under IC, which corresponds to $\Pi = \{\pi_{\text{uni}}\}$. See Corollary 1 in Appendix F.1 for the formal statement.

## 5 Optimal Tie-Breaking for Anonymity + Neutrality

While $r_{\text{ano}}$ and $r_{\text{neu}}$ in Theorem 2 are asymptotically optimal w.r.t. anonymity + neutrality, they may be hard to compute. The following proposition shows that the commonly-used LEX and FA mechanisms are far from being optimal for positional scoring rules.

**Proposition 1.** *Let $r$ be a voting rule obtained from a positional scoring correspondence by applying* LEX *or* FA. *Let $\mathcal{M} = (\Theta, \mathcal{L}(\mathcal{A}), \Pi)$ be a strictly positive and closed single-agent preference model with $\pi_{uni} \in CH(\Pi)$. There exist infinitely many $n \in \mathbb{N}$ such that:*

$$\sup_{\vec{\pi} \in \Pi^n} \Pr_{P \sim \vec{\pi}} \left( S_{ano}(r, P) + S_{neu}(r, P) < 2 \right) = \Omega(n^{-0.5})$$

The proof is done by modeling ties under positional scoring correspondences as systems of linear constraints, then applying the tightness of the polynomial bound in Lemma 1. The full proof can be found in Appendix G. We now introduce a new class of easy-to-compute tie-breaking mechanisms that achieve the optimal upper bound $O(n^{-\frac{m!}{4}})$ in Theorem 2 when $m$ is even.

**Definition 7** (**Most popular singleton ranking**). *Given a profile $P$, we define its* most popular singleton ranking (MPSR) *as $MPSR(P) = \arg\max_R (P[R] : \nexists W \neq R \text{ s.t. } P[W] = P[R])$.*

Put differently, a ranking $R$ is called a *singleton* in a profile $P$, if there does not exist another linear order that occurs for the same number of times in $P$. $\text{MPSR}(P)$ is the singleton that occurs most frequently in $P$. If no singleton exists, then we let $\text{MPSR}(P) = \emptyset$. We now define tie-breaking mechanisms based on MPSR.

**Definition 8** (**MPSR tie-breaking mechanism**). *For any voting correspondence $c$, any profile $P$, and any backup tie-breaking mechanism TB, the* MPSR-then-TB *mechanism uses $MPSR(P)$ to break ties whenever $MPSR(P) \neq \emptyset$; otherwise it uses TB to break ties.*

**Theorem 3.** *Let $\mathcal{M} = (\Theta, \mathcal{L}(\mathcal{A}), \Pi)$ be a strictly positive and closed single-agent preference model with $\pi_{uni} \in CH(\Pi)$. For any voting correspondence $c$ that satisfies anonymity and neutrality, let $r_{MPSR}$ denote the voting rule obtained from $c$ by MPSR-then-TB. For any $n$ and any $\vec{\pi} \in \Pi^n$,*

$$\Pr_{P \sim \vec{\pi}}(S_{ano}(r_{MPSR}, P) + S_{neu}(r_{MPSR}, P) < 2) = O(n^{-\frac{m!}{4}})$$

*Moreover, if TB satisfies anonymity (respectively, neutrality) then so does $r_{MPSR}$.*

The proof is done by showing that (1) anonymity and neutrality are preserved when $\text{MPSR}(P) \neq \emptyset$, and (2) any profile $P$ with $\text{MPSR}(P) = \emptyset$ can be represented by a system of linear constraints, whose smoothed likelihood is upper-bounded by the polynomial upper bound in Lemma 1. The full proof can be found in Appendix H.

Note that when $m$ is even, the $O(n^{-\frac{m!}{4}})$ upper bound in Theorem 3 matches the optimal upper bound in light of Theorem 2 and Lemma 2. This is good news because it implies that any anonymous and neutral correspondence can be made an asymptotically optimal voting rule w.r.t. anonymity+neutrality by MPSR tie-breaking. When $m$ is odd, the $O(n^{-\frac{m!}{4}})$ upper bound in Theorem 3 is suboptimal but still significantly better than that of the lexicographic or fixed-agent tie-breaking mechanism, which is $\Omega(n^{-0.5})$ (Proposition 1).

## 6 Future work

We have only touched the tip of the iceberg of smoothed analysis in social choice. There are at least three major dimensions for future work: (1) other social choice axioms and impossibility theorems, for example Arrow's impossibility theorem [36, 37, 46] and the Gibbard-Satterthwaite theorem [26, 47], (2) computational aspects in social choice [9, 4] such as the smoothed complexity of winner determination and complexity of manipulation, and (3) other social choice problems such as judgement aggregation [51, 25], distortion [54, 1, 41, 38, 42], matching, resource allocation, etc.

# 7 Acknowledgements

We thank Elliot Anshelevich, Rupert Freeman, Herve Moulin, Marcus Pivato, Nisarg Shah, Rohit Vaish, Bill Zwicker, participants of the COMSOC video seminar, and anonymous reviewers for helpful discussions and comments. This work is supported by NSF #1453542, ONR #N00014-17-1-2621, and a gift fund from Google.

## Broader Impact

In this paper we aim to provide smoothed possibilities of social choice, which is an important problem in the society. Therefore, success of the research will benefit general public beyond the CS research community because better solutions are now available for a wide range of group decision-making scenarios.

## Footnotes

[1] Some group theoretic notation and ideas in this paper are similar to those in a 2015 working paper by Doğan and Giritligil [22] whose main results are different. See Appendix D for more details and discussions.

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
