[Supplementary Material]

# A    Appendix: An Example of Single-Agent Preference Models

**Example 2.** *In a* single-agent Mallows' model $\mathcal{M}_{Ma}$, $\Theta = \mathcal{L}(\mathcal{A}) \times [0, 1]$, *where in each* $(R, \varphi) \in \Theta$, *$R$ is the central ranking and $\varphi$ is the dispersion parameter. For any* $W \in \mathcal{L}(\mathcal{A})$, *we have* $\pi_{(R,\varphi)} = \varphi^{KT(R,W)}/Z_\varphi$, *where* $KT(R, W)$ *is the* Kendall Tau distance *between $R$ and $W$, namely the number of pairwise disagreements between $R$ and $W$, and* $Z_\varphi = \sum_{W \in \mathcal{L}(\mathcal{A})} \varphi^{KT(R,W)}$ *is the normalization constant. For any* $0 < \underline{\varphi} \leq 1$, *we let* $\mathcal{M}_{Ma}^{[\underline{\varphi},1]}$ *denote the Mallows' model where the parameter space is* $\mathcal{L}(\mathcal{A}) \times [\underline{\varphi}, 1]$.

*As another example, in the* single-agent Plackett-Luce model $\mathcal{M}_{Pl}$, $\Theta = \{\vec{\theta} \in [0, 1]^m : \vec{\theta} \cdot \vec{1} = 1\}$. *For any* $\vec{\theta} \in \Theta$ *and any* $R = \sigma(1) \succ \sigma(2) \succ \cdots \succ \sigma(m)$, *we have* $\pi_{\vec{\theta}}(R) = \prod_{i=1}^{m-1} \frac{\theta_{\sigma(i)}}{\sum_{l=i}^m \theta_{\sigma(l)}}$. *For any* $0 < \underline{\varphi} \leq 1$, *we let* $\mathcal{M}_{Pl}^{[\underline{\varphi},1]}$ *denote the Plackett-Luce model where* $\Theta = \{\vec{\theta} \in [\underline{\varphi}, 1]^m : \vec{\theta} \cdot \vec{1} = 1\}$.

*It follows that for any* $0 < \underline{\varphi} \leq 1$, $\mathcal{M}_{Ma}^{[\underline{\varphi},1]}$ *and* $\mathcal{M}_{Pl}^{[\underline{\varphi},1]}$ *are strictly positive, closed, and neutral.*

# B    Appendix: Proof of Lemma 1

**Lemma 1. (Main technical lemma).** *Let $q \in \mathbb{N}$ and $\Pi$ be a closed set of strictly positive distributions over $[q]$. Let $CH(\Pi)$ denote the convex hull of $\Pi$.*

**Upper bound.** *For any $n \in \mathbb{N}$ and any $\vec{\pi} \in \Pi^n$,*

$$\Pr\left(\vec{X}_{\vec{\pi}} \in \mathcal{H}\right) = \begin{cases} 0 & \text{if } \mathcal{H} = \emptyset \\ \exp(-\Omega(n)) & \text{if } \mathcal{H} \neq \emptyset \text{ and } \mathcal{H}_{\leq 0} \cap CH(\Pi) = \emptyset \\ O(n^{-\frac{Rank(\mathbf{E})}{2}}) & \text{if } \mathcal{H} \neq \emptyset \text{ and } \mathcal{H}_{\leq 0} \cap CH(\Pi) \neq \emptyset \end{cases}$$

**Tightness of the upper bound.** *There exists a constant $C$ such that for any $n' \in \mathbb{N}$, there exists $n' \leq n \leq Cn'$ and $\vec{\pi} \in \Pi^n$ such that*

$$\Pr\left(\vec{X}_{\vec{\pi}} \in \mathcal{H}\right) = \begin{cases} \exp(-O(n)) & \text{if } \mathcal{H} \neq \emptyset \text{ and } \mathcal{H}_{\leq 0} \cap CH(\Pi) = \emptyset \\ \Omega(n^{-\frac{Rank(\mathbf{E})}{2}}) & \text{if } \mathcal{H} \neq \emptyset \text{ and } \mathcal{H}_{\leq 0} \cap CH(\Pi) \neq \emptyset \end{cases}$$

*Proof.* The $\mathcal{H} = \emptyset$ case trivially holds. Let $o = \text{Rank}(\mathbf{E})$. Let $\vec{X}_{\vec{\pi}} = \text{Hist}(\vec{Y}) = (X_{\vec{\pi},1} \ldots, X_{\vec{\pi},q})$. That is, for any $i \leq q$, $X_{\vec{\pi},i}$ represents the number of occurrences of outcome $i$ in $\vec{Y}$. For any $n \in \mathbb{N}$ and any $\vec{\pi} \in \Pi^n$, let $\vec{\mu}_{\vec{\pi}} = (\mu_{\vec{\pi},1}, \ldots, \mu_{\vec{\pi},q}) = \mathbb{E}(\sum_{j=1}^n \vec{X}_{\vec{\pi}}/n)$ denote the mean of $\vec{X}_{\vec{\pi}}/n$ and let $\vec{\sigma}_{\vec{\pi}} = (\sigma_{\vec{\pi},1}, \ldots, \sigma_{\vec{\pi},q})$, where for each $i \leq q$, $\sigma_{\vec{\pi},i} = \sqrt{\text{Var}(X_{\vec{\pi},i})/n}$. Because $\Pi$ is strictly positive, there exists $\epsilon_1 > 0, \epsilon_2 > 0$ such that for all $n$, all $\vec{\pi} \in \Pi^n$, and all $i \leq q$, we have $\epsilon_1 < \mu_{\vec{\pi},i} < \epsilon_2$ and $\frac{\epsilon_1}{\sqrt{n}} < \sigma_{\vec{\pi},i} < \frac{\epsilon_2}{\sqrt{n}}$.

**Upper bound when $\mathcal{H} \neq \emptyset$ and $\mathcal{H}_{\leq 0} \cap CH(\Pi) = \emptyset$.** It is not hard to see that $\mathcal{H}_{\leq 0}$ is convex and closed. Because $\Pi$ is closed and bounded, $CH(\Pi)$ is convex, closed and compact. Because $\mathcal{H}_{\leq 0} \cap CH(\Pi) = \emptyset$, by the strict hyperplane separation theorem, there exists a hyperplane that strictly separates $\mathcal{H}_{\leq 0}$ and $CH(\Pi)$. Therefore, there exists $\epsilon' > 0$ such that for any $\vec{x}_1 \in \mathcal{H}_{\leq 0}$ and any $\vec{x}_2 \in CH(\Pi)$, we have $|\vec{x}_1 - \vec{x}_2|_\infty > \epsilon'$, where $|\cdot|_\infty$ is the $L_\infty$ norm. This means that any solution to $CH(\vec{x})$ is at least $\epsilon'n$ away from $n \cdot \vec{\mu}_{\vec{\pi}}$ in $L_\infty$. Therefore, we have:

$$\Pr\left(\vec{X}_{\vec{\pi}} \in \mathcal{H}\right) \leq \Pr\left(|\vec{X}_{\vec{\pi}} - n \cdot \vec{\mu}_{\vec{\pi}}|_\infty > \epsilon'n\right) \leq \sum_{i=1}^q \Pr(|X_{\vec{\pi},i} - n\mu_{\pi,i}| > \epsilon'n)$$

$$\leq 2q \exp\left(-\frac{(\epsilon')^2 n}{(1 - 2\epsilon)^2}\right)$$

The last inequality follows after Hoeffding's inequality (Theorem 2 in [34]), where $\epsilon$ is a constant such that any distribution in $\Pi$ is above $\epsilon$.

**Upper bound when $\mathcal{H} \neq \emptyset$ and $\mathcal{H}_{\leq 0} \cap CH(\Pi) \neq \emptyset$.** Let $C^{\mathbf{E}}(\vec{x}) = \{\mathbf{E} \cdot \vec{x} = (\vec{0})^\top\}$ denote the relaxation of $C^{\mathbf{ES}}(\vec{x})$ by removing the $\mathbf{S}$ part. Let $C^{\mathbf{E}}(\vec{X}_{\vec{\pi}})$ denote the event that $\vec{X}_{\vec{\pi}}$ satisfies

all constraints in $C^E$. It follows that each vector in $\mathcal{H}$ is a solution to $C^E(\vec{x})$, which means that $\Pr(\vec{X}_{\vec{\pi}} \in \mathcal{H}) \leq \Pr(C^E(\vec{X}_{\vec{\pi}}))$. Therefore, it suffices to prove that for any $n \in \mathbb{N}$ and any $\vec{\pi} \in \Pi^n$, $\Pr(C^E(\vec{X}_{\vec{\pi}})) = O(n^{-\frac{o}{2}})$. Because $\mathbf{E} \cdot (\vec{1})^\top = (\vec{0})^\top$, it follows that $\vec{1} = \{1\}^q$ is linearly independent with the row vectors of $\mathbf{E}$. Therefore, the rank of $\mathbf{E}' = \begin{bmatrix} \mathbf{E} \\ \vec{1} \end{bmatrix}$ is $o + 1$. Let $C^{\mathbf{E}'}(\vec{x}) = \{\mathbf{E} \cdot \vec{x} = 0 \text{ and } \vec{1} \cdot \vec{x} = n\}$. From basic linear algebra, in particular the reduced row echelon form (a.k.a. row canonical form) [45] of $\mathbf{E}'$ computed by Gauss-Jordan elimination where $n$ is treated as a constant, we know that there exist $I_0 \subseteq [q]$ and $I_1 \subseteq [q]$ such that $I_0 \cap I_1 = \emptyset$, $|I_0| = \text{Rank}(\mathbf{E}') = \text{Rank}(\mathbf{E}) + 1$, and an $|I_0| \times (|I_1| + 1)$ matrix $\mathbf{D}$ in $\mathbb{Q}$ such that $C^{\mathbf{E}'}(\vec{x})$ is equivalent to $(\vec{x}_{I_0})^\top = \mathbf{D} \cdot [\vec{x}_{I_1}, n]^\top$, where $\vec{x}_{I_0}$ is the subvector of $\vec{x}$ that contains variables whose subscripts are in $I_0$. In other words, a vector $\vec{x}$ satisfies $C^{\mathbf{E}'}(\vec{x})$ if and only if $(\vec{x}_{I_0})^\top = \mathbf{D} \cdot [\vec{x}_{I_1}, n]^\top$. See Example 4 in Appendix B.1 for an example of deriving $\mathbf{D}$ from $\mathbf{E}'$.

We note that $I_1 \cup I_0 = [q]$. For the sake of contradiction suppose this is not true, which means that the reduced row echelon form of $\mathbf{E}'$ has a column of zeros for some variable $x_j$. However, this means that $\vec{1}$ is not a linear combination of the rows of the reduced row echelon form of $\mathbf{E}'$, which is a contradiction because $\mathbf{E}'$, which includes $\vec{1}$, can be obtained from a series of linear transformations on its reduced row echelon form. W.l.o.g. in the remainder of this proof we let $I_0 = \{1, \ldots, o+1\}$ and $I_1 = \{o+2, \ldots, q\}$.

The hardness in bounding $\Pr(C^E(\vec{X}_{\vec{\pi}}))$ is that elements of $\vec{X}_{\vec{\pi}}$ are not independent, and typical asymptotical tools such as Lyapunov-type bound are too coarse. To solve this issue, we use the following alternative representation of $Y_1, \ldots, Y_n$. For each $j \leq n$, we use a binary random variable $Z_j \in \{0, 1\}$ to represent whether the outcome of $Y_j$ is in $I_0$ (corresponding to $Z_j = 0$) or is in $I_1$ (corresponding to $Z_j = 1$). Then, we use another random variable $W_j \in [q]$ to represent the outcome of $Y_j$ conditioned on $Z_j$. See Figure 1 for an illustration.

Figure 1: The representation of $\vec{Y}$ as $\vec{Z}$ and $\vec{W}$, where $\vec{W}$ has the same distribution as $\vec{Y}$.

At a high level, this addresses the independent issue because components of $\vec{W}$ are conditionally independent given $\vec{Z}$, and as we will see below, concentration happens in $\vec{W}$ when $\vec{Z}$ contains $\Theta(n)$ many 0's.

**Definition 9 (Alternative representation of $Y_1, \ldots, Y_n$).** *For each $j \leq n$, we define a Bayesian network with two random variables $Z_j \in \{0, 1\}$ and $W_j \in [q]$, where $Z_j$ is the parent of $W_j$, and*

- *for each $l \in \{0, 1\}$, $\Pr(Z_j = l) = \Pr(Y_j \in I_l)$;*
- *for each $l \in \{0, 1\}$ and each $t \leq q$, $\Pr(W_j = t | Z_j = l) = \Pr(Y_j = t | Y_j \in I_l)$.*

*In particular, if $t \notin I_l$ then $\Pr(W_j = t | Z_j = l) = 0$.*

It follows that $W_j$ has the same distribution as $Y_j$. For any $\vec{z} \in \{0, 1\}^n$, we let $\text{Ind}_0(\vec{z}) \subseteq [n]$ denote the indices of $\vec{z}$ that equals to 0. Given $\vec{z}$, we let $\vec{W}_{\text{Ind}_0(\vec{z})}$ denote the set of all $W_j$'s with $z_j = 0$, and let $\text{Hist}(\vec{W}_{\text{Ind}_0(\vec{z})})$ denote the vector of $o + 1$ random variable that correspond to the histogram of $\vec{W}_{\text{Ind}_0(\vec{z})}$. Similarly, we let $\text{Hist}(\vec{W}_{\text{Ind}_1(\vec{z})})$ denote the vector of $q - o - 1$ random variables that correspond to the histogram of $\vec{W}_{\text{Ind}_1(\vec{z})}$. Recall that each $\pi_j$ is at least $\epsilon > 0$. We have the following

calculation on $\Pr(C^{\mathbf{E}}(\vec{X}_{\vec{\pi}}))$, i.e. the probability that $\text{Hist}(\vec{Y})$ satisfies all constraints in $C^{\mathbf{E}}$.

$$\Pr(C^{\mathbf{E}}(\vec{X}_{\vec{\pi}})) = \sum_{\vec{z}\in\{0,1\}^n} \Pr(\vec{Z}=\vec{z})\Pr\left(C^{\mathbf{E}}(\text{Hist}(\vec{W})) \,\Big|\, \vec{Z}=\vec{z}\right) \text{ (total probability)}$$

$$= \sum_{\vec{z}\in\{0,1\}^n} \Pr(\vec{Z}=\vec{z})\Pr\left(\text{Hist}(\vec{W}_{\text{Ind}_0(\vec{z})})^\top = \mathbf{D}\cdot[\text{Hist}(\vec{W}_{\text{Ind}_1(\vec{z})}),n]^\top \,\Big|\, \vec{Z}=\vec{z}\right)$$

$$= \sum_{\vec{z}\in\{0,1\}^n} \Pr(\vec{Z}=\vec{z}) \sum_{\vec{x}\in\mathbb{Z}_{\geq0}^{q-o-1}} \Pr\left(\text{Hist}(\vec{W}_{\text{Ind}_1(\vec{z})}) = \vec{x} \,\Big|\, \vec{Z}=\vec{z}\right)$$

$$\times \Pr\left(\text{Hist}(\vec{W}_{\text{Ind}_0(\vec{z})})^\top = \mathbf{D}\cdot[\vec{x},n]^\top \,\Big|\, \vec{Z}=\vec{z}\right) \qquad (4)$$

$$= \sum_{\vec{z}\in\{0,1\}^n} \Pr(\vec{Z}=\vec{z}) \sum_{\vec{x}\in\mathbb{Z}_{\geq0}^{q-o-1}} \Pr\left(\text{Hist}(\vec{W}_{\text{Ind}_1(\vec{z})}) = \vec{x} \,\Big|\, [\vec{Z}]_{\text{Ind}_1(\vec{z})} = \vec{1}\right)$$

$$\times \Pr\left(\text{Hist}(\vec{W}_{\text{Ind}_0(\vec{z})})^\top = \mathbf{D}\cdot[\vec{x},n]^\top \,\Big|\, [\vec{Z}]_{\text{Ind}_0(\vec{z})} = \vec{0}\right) \qquad (5)$$

$$\leq \sum_{\vec{z}\in\{0,1\}^n:[\text{Hist}(\vec{z})]_0\geq0.9\epsilon n} \Pr(\vec{Z}=\vec{z}) \sum_{\vec{x}\in\mathbb{Z}_{\geq0}^{q-o-1}} \Pr\left(\text{Hist}(\vec{W}_{\text{Ind}_1(\vec{z})}) = \vec{x} \,\Big|\, [\vec{Z}]_{\text{Ind}_1(\vec{z})} = \vec{1}\right)$$

$$\times \Pr\left(\text{Hist}(\vec{W}_{\text{Ind}_0(\vec{z})})^\top = \mathbf{D}\cdot[\vec{x},n]^\top \,\Big|\, [\vec{Z}]_{\text{Ind}_0(\vec{z})} = \vec{0}\right) + \Pr([\text{Hist}(\vec{Z})]_0 < 0.9\epsilon n) \qquad (6)$$

where $[\text{Hist}(\vec{z})]_0$ is the number of 0's in $\vec{z}$. (4) holds because $W_j$'s are independent of each other given $Z_j$'s, which means that for any $\vec{z}\in\{0,1\}^n$, $\text{Hist}(\vec{W}_{\text{Ind}_0(\vec{z})})$ and $\text{Hist}(\vec{W}_{\text{Ind}_1(\vec{z})})$ are independent given $\vec{z}$. (5) holds because $\vec{W}_{\text{Ind}_1(\vec{z})}$ (respectively, $\vec{W}_{\text{Ind}_0(\vec{z})}$) is independent of $[\vec{Z}]_{\text{Ind}_0(\vec{z})}$ (respectively, $[\vec{Z}]_{\text{Ind}_1(\vec{z})}$) given $[\vec{Z}]_{\text{Ind}_1(\vec{z})}$ (respectively, $[\vec{Z}]_{\text{Ind}_0(\vec{z})}$) in light of independence in the Bayesian network. To simplify notation, we write (6) as follows.

$$\sum_{\vec{z}\in\{0,1\}^n:[\text{Hist}(\vec{z})]_0\geq0.9\epsilon n} \Pr(\vec{Z}=\vec{z}) \sum_{\vec{x}\in\mathbb{Z}_{\geq0}^{q-o-1}} F_1(\vec{z},\vec{x})\times F_2(\vec{z},\vec{x}) + F_3, \text{ where}$$

$$F_1(\vec{z},\vec{x}) = \Pr\left(\text{Hist}(\vec{W}_{\text{Ind}_1(\vec{z})}) = \vec{x} \,\Big|\, [\vec{Z}]_{\text{Ind}_1(\vec{z})} = \vec{1}\right)$$

$$F_2(\vec{z},\vec{x}) = \Pr\left(\text{Hist}(\vec{W}_{\text{Ind}_0(\vec{z})})^\top = \mathbf{D}\cdot[\vec{x},n]^\top \,\Big|\, [\vec{Z}]_{\text{Ind}_0(\vec{z})} = \vec{0}\right)$$

$$F_3 = \Pr([\text{Hist}(\vec{Z})]_0 < 0.9\epsilon n)$$

We now show that given $[\text{Hist}(\vec{z})]_0 \geq 0.9\epsilon n$, for any $\vec{x}\in\mathbb{Z}_{\geq0}^{q-o-1}$,

$$F_2(\vec{z},\vec{x}) = O((0.9\epsilon n)^{-\frac{o}{2}}) = O(n^{-\frac{o}{2}}),$$

which follows after the following lemma, where $n = |\text{Ind}_0(\vec{z})|$ and $q^* = o+1$. The lemma can be seen as a Poisson multivariate extension of the Littlewood-Offord-Erdős anti-concentration bound, because it says that the probability for $\text{Hist}(\vec{W}_{\text{Ind}_0(\vec{z})})^\top$ to take a specific value $\mathbf{D}\cdot[\vec{x},n]^\top$ (which is a constant vector given $\vec{x}$) is bounded above by $O(n^{-\frac{o}{2}})$.

**Lemma 3 (Point-wise anti-concentration bound for Poisson multinomial variables).** *Given $q^* \in \mathbb{N}$ and $\epsilon > 0$. There exists a constant $C^* > 0$ such that for any $n \in \mathbb{N}$ and any vector $\vec{Y}' = (Y_1',\ldots,Y_n')$ of $n$ independent random variables over $[q^*]$, each of which is above $\epsilon$, and any vector $\vec{x}\in\mathbb{Z}_{\geq0}^{q^*}$, we have $\Pr(\text{Hist}(\vec{Y}') = \vec{x}) < C^* n^{\frac{1-q^*}{2}}$.*

*Proof.* When $\vec{x}\cdot\vec{1} \neq n$ the inequality holds for any $C^* > 0$. Suppose $\vec{x}\cdot\vec{1} = n$, we prove the claim by induction on $q^*$. When $q^* = 1$ the claim holds for any $C_1^* > 1$. Suppose the claim holds for $q^* = q'-1$ w.r.t. constant $C_{q'-1}^*$. W.l.o.g. suppose $x_1 \geq x_2\cdots \geq x_{q^*}$. When $q^* = q'$, we use the following representation of $Y_1',\ldots,Y_n'$ that is similar to Definition 9. For each $j \leq n$, let

$Y'_j$ be represented by the Bayesian network with two random variables $Z'_j \in \{0,1\}$ and its child $W'_j \in \{1,\ldots,q'\}$. Let $\Pr(Z'_j = 0) = \Pr(Y'_j \in \{1,\ldots,q'-1\})$ and $\Pr(Z'_j = 1) = \Pr(Y'_j = q')$ . Let $\Pr(W'_j = q'|Z'_j = 1) = 1$ and for all $l \leq q'-1$, $\Pr(W'_j = l|Z'_j = 0) = \Pr(Y'_j = l|Y'_j \in \{1,\ldots q'-1\})$.

It follows that $\Pr(Z'_j = 0, W'_j = q') = 0$, $\Pr(Y'_j = q') = \Pr(Z'_j = 1, W'_j = q')$, and for all $l \leq q'-1$, $\Pr(Y'_j = l) = \Pr(Z'_j = 0, W'_j = l)$ and $\Pr(Z'_j = 1, W'_j = l) = 0$. In other words, $Z'_j$ determines whether $Y'_j \in \{1,\ldots,q'-1\}$ (corresponding to $Z'_j = 0$) or $Y'_j = q'$ (corresponding to $Z'_j = 1$), and $W'_j$ determines the value of $Y'_j$ conditioned on $Z'_j$. By the law of total probability, we have:

$$\Pr(\text{Hist}(\vec{Y'}) = \vec{x}) = \sum_{\vec{z} \in \{0,1\}^n : \vec{z} \cdot \vec{1} = x_{q'}} \Pr\left(\text{Hist}(\vec{W'}) = \vec{x} \mid \vec{Z'} = \vec{z}\right) \cdot \Pr(\vec{Z'} = \vec{z}),$$

We note that $W'_j$'s are independent of each other given $\vec{Z'}$, and $\Pr(W_j = q'|Z_j = 1) = 1$. Therefore, for any $\vec{z} \in \{0,1\}^n$ with $\vec{z} \cdot \vec{1} = x_{q'}$, we have

$$\Pr\left(\text{Hist}(\vec{W'}) = \vec{x} \mid \vec{Z'} = \vec{z}\right) = \Pr\left(\text{Hist}(\vec{W'})_{-q'} = \vec{x}_{-q'} \mid \vec{Z'}_{\text{Ind}_0(\vec{z})} = \vec{0}\right)$$

$$\leq C^*_{q'-1}|n - x_{q'}|^{(2-q')/2} \leq C^*_{q'-1}(\frac{q'-1}{q'})^{(2-q')/2}n^{(2-q')/2}$$

where $\text{Hist}(\vec{Z'})_{-q'}$ is the subvector of $\text{Hist}(\vec{Z'})$ by taking out the $q'$-th component and $\vec{x}_{-q'} = (x_1,\ldots,x_{q'-1})$. The first inequality follows after the induction hypothesis, because each $W'_j$ with $j \in \text{Ind}_0(\vec{z})$ is a random variable over $\{1,\ldots,q'-1\}$ that is above $\epsilon$. The second step uses the assumption that $x_1 \geq x_2 \geq \cdots \geq x_{q'}$, which means that $x_{q'} \leq \frac{1}{q'}n$. The next claim, which can be seen as an extension of the Littlewood-Offord-Erdős anti-concentration bound to Poisson binomial distributions, proves that $\Pr(\vec{Z'} \cdot \vec{1} = x_{q'}) = O(n^{-1/2})$.

**Claim 1.** *There exists a constant $C'$ that does not depend on $n$ or $q'$ such that $\Pr(\vec{Z'} \cdot \vec{1} = x_{q'}) \leq C'n^{-1/2}$.*

*Proof.* For any $j \leq n$, recall that $\Pr(Z'_j = 0) \geq \epsilon$ and $\Pr(Z'_j = 1) \geq \epsilon$. Therefore, for all $j \leq n$, $\text{Var}(Z'_j) \geq \epsilon(1-\epsilon)$ and there exists a constant $\rho$ such that for all $j \leq n$, $\mathbb{E}(|Z'_j - \mathbb{E}(Z'_j)|^3) < \rho$. Let $\mu = \sum_{j=1}^n Z'_j$ and $\sigma = \sqrt{\sum_{j=1}^n \text{Var}(Z'_j)}$. We have $\sigma \geq \sqrt{n\epsilon(1-\epsilon)}$. By Berry-Esseen theorem (see e.g. [23]), there exists a constant $C_0$ such that:

$$\Pr(\vec{Z'} \cdot \vec{1} = x_{q'}) \leq \Pr(x_{q'} - 1 < \vec{Z'} \cdot \vec{1} \leq x_{q'} + 1) = \Pr(\frac{x_{q'}-1}{\sigma} \leq \frac{\vec{Z'} \cdot \vec{1}}{\sigma} \leq \frac{x_{q'}+1}{\sigma})$$

$$\leq (\text{Hist}(\frac{x_{q'}-\mu+1}{\sigma}) - \text{Hist}(\frac{x_{q'}-\mu-1}{\sigma})) + 2C_0(n\epsilon(1-\epsilon))^{-3/2}n\rho$$

$$\leq \frac{2}{\sigma} + 2C_0(n\epsilon(1-\epsilon))^{-3/2}n\rho \leq \left(\frac{2}{\sqrt{\epsilon(1-\epsilon)}} + \frac{2C_0\rho}{(\epsilon(1-\epsilon))^{3/2}}\right)n^{-1/2}$$

The claim follows by letting $C' = \frac{2}{\sqrt{\epsilon(1-\epsilon)}} + \frac{2C_0\rho}{(\epsilon(1-\epsilon))^{3/2}}$. $\square$

Finally, we have

$$\Pr(\text{Hist}(\vec{Y'}) = \vec{x}) \leq C^*_{q'-1}(\frac{q'-1}{q'})^{(2-q')/2}n^{(2-q')/2} \sum_{\vec{z} \in \{0,1\}^n : \vec{z} \cdot \vec{1} = x_{q'}} \Pr(\vec{Z'} = \vec{z})$$

$$\leq C'n^{-1/2}C^*_{q'-1}(\frac{q'-1}{q'})^{(2-q')/2}n^{(2-q')/2}$$

This proves the $q^* = q'$ case by letting $C^*_{q'} = C'C^*_{q'-1}(\frac{q'-1}{q'})^{(2-q')/2}$. This completes the proof of Lemma 3. $\square$

Because random variables in $\vec{Y}$ are above $\epsilon$, for all $j \leq n$, $Z_j$ takes 0 with probability at least $\epsilon$. Therefore, $\mathbb{E}(\vec{Z} \cdot \vec{1}) \geq \epsilon n$. By Hoeffding's inequality, $F_3 = \Pr([\mathrm{Hist}(\vec{Z})]_0 < 0.9\epsilon n)$ is exponentially small in $n$, which means that it is $O(n^{-\frac{o}{2}})$. We also note that for any $\vec{z}$ and $\vec{x}$ we have $F_1(\vec{z}, \vec{x}) \leq 1$. Therefore, continuing (6), we have:

$$\Pr(\mathbf{C^E}(\vec{X}_{\vec{\pi}})) \leq \sum_{\vec{z} \in \{0,1\}^n : [\mathrm{Hist}(\vec{z})]_0 \geq 0.9dn} \Pr(\vec{Z} = \vec{z}) \sum_{\vec{x} \in \mathbb{Z}_{\geq 0}^{q-o-1}} F_1(\vec{z}, \vec{x}) \times F_2(\vec{z}, \vec{x}) + F_3$$

$$\leq \sum_{\vec{z} \in \{0,1\}^n : [\mathrm{Hist}(\vec{z})]_0 \geq 0.9dn} \Pr(\vec{Z} = \vec{z}) O(n^{-\frac{o}{2}}) + O(n^{-\frac{o}{2}}) = O(n^{-\frac{o}{2}})$$

This proves the upper bound when $\mathcal{H} \neq \emptyset$ and $\mathcal{H}_{\leq 0} \cap \mathrm{CH}(\Pi) \neq \emptyset$.

**Tightness of the upper bound when $\mathcal{H} \neq \emptyset$ and $\mathcal{H}_{\leq 0} \cap \mathrm{CH}(\Pi) = \emptyset$.** We first prove the following claim that will be frequently used in this proof. The claim follows after a straightforward application of Theorem 1(i) in [17]. It states that for any $\vec{x} \in \mathcal{H}$ and any $l > 0$, $\mathcal{H}$ contains an integer vector that is close to $l \cdot \vec{x}$.

**Claim 2.** *Suppose $\mathcal{H} \neq \emptyset$. There exists a constant $C' > 0$ such that for any $\vec{x} \in \mathcal{H}$, there exists an integer vector $\vec{x}^* \in \mathcal{H}$ with $|\vec{x}^* - l \cdot \vec{x}|_\infty < C'$.*

*Proof.* We first prove that $\mathcal{H}$ contains an integer vector. Because $\mathcal{H} \neq \emptyset$ and $\mathbf{E}$ is an integer matrix, $\mathcal{H}$ contains a rational solution $\vec{x}'$, which can be chosen from the neighborhood of any vector in $\mathcal{H}$. It follows that for any $L > 0$, $L \cdot \vec{x}' \in \mathcal{H}$ because $\mathbf{E} \cdot (L \cdot \vec{x}')^\top = (\vec{0})^\top$ and $\mathbf{S} \cdot (L \cdot \vec{x}')^\top < (\vec{0})^\top$. It is not hard to see that there exists $L \in \mathbb{N}$ such that $L \cdot \vec{x}' \in \mathbb{Z}_{\geq 0}^q$.

The claim then follows after Theorem 1(i) in [17] by replacing $\mathbf{S} \cdot (\vec{x})^\top < (\vec{0})^\top$ with equivalent constraints $\mathbf{S} \cdot (L \cdot \vec{x}')^\top \leq (-\vec{1})^\top$. $\qquad\square$

Let $C'$ denote the constant in Claim 2. W.l.o.g. let $\vec{x} \in \mathcal{H}$ denote an arbitrary solution such that $\vec{x} > C' \cdot \vec{1}$; otherwise we consider $\vec{x} + (|\vec{x}|_\infty + C') \cdot \vec{1} \in \mathcal{H}$. For any $l \in \mathbb{N}$, let $\vec{x}_l^*$ denote the integer vector in $\mathcal{H}$ that is guaranteed by Claim 2; let $n_l = \vec{x}_l^* \cdot \vec{1}$; and let $\vec{\pi}^l \in \Pi^{n_l}$ denote an arbitrary vector of $n_l$ distributions, each of which is chosen from $\Pi$. Let $\vec{y}^l \in [q]^{n_l}$ denote an arbitrary vector with $\mathrm{Hist}(\vec{y}^l) = \vec{x}_l^*$. We note that for any $l \in \mathbb{N}$, each distribution in $\vec{\pi}^l$ is above $\epsilon$. Therefore, $\Pr(\vec{X}_{\vec{\pi}^l} = \vec{y}^l) \geq \epsilon^{n_l} = \exp(n_l \log \epsilon)$, which is $\exp(-O(n_l))$. It is not hard to verify that for any $l \in \mathbb{N}$, $\vec{x}_l^*$ is strictly positive and $\frac{n_{l+1}}{n_l}$ is bounded above by a constant, denoted by $\hat{C}$. Let $C = \max(n_1, \hat{C})$. This proves the tightness of the upper bound when $\mathcal{H} \neq \emptyset$ and $\mathcal{H}_{\leq 0} \cap \mathrm{CH}(\Pi) = \emptyset$.

**Tightness of the upper bound when $\mathcal{H} \neq \emptyset$ and $\mathcal{H}_{\leq 0} \cap \mathrm{CH}(\Pi) \neq \emptyset$.** Let $\vec{x}^* \in \mathcal{H}_{\leq 0} \cap \mathrm{CH}(\Pi)$ and write $\vec{x}^* = \sum_{i=1}^k \alpha_i \pi_i$ as a linear combination of vectors in $\Pi$. The tightness of the upper bound will be proved in the following four steps. Step 1. For every $l \in \mathbb{N}$ we prove that there exists an integer vector $\vec{y}^l \in \mathcal{H}$ that is $\Theta(\sqrt{l})$ away from $l \cdot \vec{x}^*$. Then, we define $\vec{\pi}^l = (\vec{\pi}_1^l, \ldots, \vec{\pi}_k^l) \in \Pi^{n_l}$, where for all $i \leq k$, $\vec{\pi}_i^l$ is approximately $l\alpha_i$ copies of $\pi_i$, which means that $n_l = \Theta(l)$. Step 2. We identify $\Omega(n_l^{\frac{(q-1)(k-1)+q-o-1}{2}})$ combinations of values of $\mathrm{Hist}(\vec{X}_{\vec{\pi}_1^l}), \ldots, \mathrm{Hist}(\vec{X}_{\vec{\pi}_k^l})$, such that the sum of each such combination is no more than $O(\sqrt{n_l})$ away from $\vec{y}^l$ and is a solution to $\mathbf{C^{ES}}(\vec{x})$. Step 3. We prove that the probability of each such combination is $\Omega(n_l^{\frac{(1-q)}{2}})$ by applying Lemma 4 below. Finally, we will have $\Pr(\vec{X}_{\vec{\pi}} \in \mathcal{H}) \geq \Omega(n_l^{\frac{(q-1)(k-1)+q-o-1}{2}}) \times \Omega(n_l^{\frac{(1-q)}{2}})^k = \Omega(n_l^{-\frac{o}{2}})$.

**Step 1.** Let $\vec{x}^\# \in \mathcal{H}$ and $\vec{x}^* \in \mathcal{H}_{\leq 0} \cap \mathrm{CH}(\Pi)$. W.l.o.g. suppose $\vec{x}^\#$ is strictly positive; otherwise let $l > 0$ denote an arbitrary number such that $\vec{x}^\# + l \cdot \vec{1}$ is strictly positive, and because $\mathbf{E} \cdot (\vec{1})^\top = (\vec{0})^\top$ and $\mathbf{S} \cdot (\vec{1})^\top = (\vec{0})^\top$, we have $\vec{x}^\# + l \cdot \vec{1} \in \mathcal{H}$. Because $\vec{x}^* \in \mathcal{H}_{\leq 0} \cap \mathrm{CH}(\Pi)$, we can write $\vec{x}^* = \sum_{i=1}^k \alpha_i \pi_i$, where for all $i \leq k$, $\alpha_i > 0$ and $\pi_i \in \Pi$, and $\sum_{i=1}^k \alpha_i = 1$. We note that $\vec{x}^* \geq \epsilon \cdot \vec{1}$, because $\Pi$ is above $\epsilon$.

For any sufficiently large $l \in \mathbb{N}$, we let $\vec{y}^{l\#} \in \mathcal{H}$ denote the integer approximation to $\sqrt{l} \cdot \vec{x}^\#$ that is guaranteed by Claim 2; let $\vec{y}^{l*}$ denote the integer approximation to $l \cdot \vec{x}^*$ that is guaranteed by

Claim 2, where we merge all rows $\vec{B}$ in $\mathbf{S}$ with $\vec{B} \cdot \vec{x}^* = 0$ to $\mathbf{E}$ before applying the claim. It follows that $\vec{y}^{l*} \in \mathcal{H}_{\leq 0}$. Let $C$ denote an arbitrary constant such that $|\vec{y}^{l\#} - \sqrt{l} \cdot \vec{x}^{\#}|_\infty < C$ and $|\vec{y}^{l*} - l \cdot \vec{x}^*|_\infty < C$.

Let $\vec{y}^l = \vec{y}^{l\#} + \vec{y}^{l*}$ and let $n_l = \vec{y}^l \cdot \vec{1}$. It follows that $\vec{y}^l$ is a strictly positive integer solution to $\mathbf{C}^{\mathbf{ES}}(\vec{x})$ and each of its element is $\Theta(n_l)$. We now define $\vec{\pi}^l$ that is approximately $l$ copies of $\vec{x}^*$. Formally, for each $i \leq k-1$, let $\vec{\pi}^l_i$ denote the vector of $\beta_i = \lfloor l\alpha_i \rfloor$ copies of $\pi_i$. Let $\vec{\pi}^l_k$ denote the vector of $\beta_k = n_l - \sum_{i=1}^{k-1} \beta_i$ copies of $\pi_k$. It follows that for any $i \leq k-1$, $|\beta_i - l\alpha_i| \leq 1$, and $|\beta_k - l\alpha_k| \leq k + \vec{y}^{l\#} \cdot \vec{1} = O(\sqrt{l}) = O(\sqrt{n_l})$. Let $\vec{\pi}^l = (\vec{\pi}^l_1, \dots, \vec{\pi}^l_k)$ denote the vector of $n_l$ distributions. It follows that $|\mathbb{E}(\vec{X}_{\vec{\pi}^l}) - \vec{y}^l|_\infty = O(\sqrt{n_l})$.

**Step 2.** We define a set $\mathcal{X}_l \subseteq \mathbb{N}^{qk}$ of vectors $(\vec{x}_1, \dots, \vec{x}_k)$ such that $\vec{x}_i = (x_{i1}, \dots, x_{iq}) \in \mathbb{N}^q$ will be used as a target value for $\vec{X}_{\vec{\pi}^l_i}$ soon in the proof. Let $\mathcal{X}_l$ denote the set of all integer vectors $(\vec{x}_1, \dots, \vec{x}_k)$ that satisfy the following three conditions.

(i) Let $\zeta > 0$ be a constant whose value will be specified later. For any $i \leq k-1$ and any $j \leq q-1$, we require $|x_{ij} - \pi^l_{ij}\beta_i| < \zeta\sqrt{n_l}$, where $\vec{\pi}^l_i = (\pi^l_{i1}, \dots, \pi^l_{iq})$. Therefore, $|x_{iq} - \pi^l_{iq}\beta_i| < (q-1)\zeta\sqrt{n_l}$, because $\sum_{j \leq q} x_{ij} = \beta_i$.

(ii) Let $\rho$ be the least common multiple of the denominators of all entries in $\mathbf{D}$. For example, in Example 4 we have $\rho = 2$. For each $o + 2 \leq j \leq q$, we require $|x_{kj} - (y^l_j - \sum_{i=1}^{k-1} x_{ij})| < \zeta\sqrt{n_l}$ and $\rho$ divides $x_{kj} - (y^l_j - \sum_{i=1}^{k-1} x_{ij})$.

(iii) $\vec{x} = \sum_{i=1}^k \vec{x}_i$ is a solution to $(\vec{x}_{I_0})^\top = \mathbf{D} \cdot [\vec{x}_{I_1}, n_l]^\top$.

Each element $(\vec{x}_1, \dots, \vec{x}_k)$ of $\mathcal{X}_l$ can be generated in the following three steps. First, $\vec{x}_1, \dots, \vec{x}_{k-1}$ are arbitrarily chosen according to condition (i) above. Second, $x_{k(o+2)}, \dots, x_{kq}$ are chosen according to condition (ii) above given $\vec{x}_1, \dots, \vec{x}_{k-1}$. This guarantees that for each $o + 2 \leq j \leq q$, $\sum_{i=1}^k x_{ij}$ is no more than $O(\sqrt{n_l})$ away from $y^l_j$ and $\sum_{i=1}^k x_{ij} - y^l_j$ is divisible by $\rho$. Finally, $x_{k1}, \dots, x_{k(o+1)}$ are determined by condition (iii) together with other components of $\vec{x}$ that are specified in the first and second step. More precisely,

$$
\begin{bmatrix} x_{k1} \\ \vdots \\ x_{k(o+1)} \end{bmatrix} = \mathbf{D} \cdot \begin{bmatrix} \sum_{i=1}^k x_{i(o+2)} \\ \vdots \\ \sum_{i=1}^k x_{iq} \\ n_l \end{bmatrix} - \sum_{i=1}^{k-1} \begin{bmatrix} x_{i1} \\ \vdots \\ x_{i(o+1)} \end{bmatrix}
$$

$x_{k1}, \dots, x_{k(o+1)}$ are integers because for all $o + 2 \leq j \leq q$, $\rho$ divides $y_j - \sum_{i=1}^k x_{ij}$ and $\mathbf{D} \cdot [\vec{y}_{I_1}, n_l]^T = [\vec{y}_{I_0}, n_l]^T$. We let $\zeta > 0$ be a sufficiently small constant that does not depend on $n_l$, such that the following two conditions hold.

(1) Each $(\vec{x}_1, \dots, \vec{x}_k) \in \mathcal{X}_l$ is strictly positive. This can be achieved by assigning $\zeta$ a small positive value, because $|\sum_{i=1}^k \vec{x}_i - \vec{y}^l|_\infty = O(\zeta)\sqrt{n_l}$ and each element in $\vec{y}^l$ is strictly positive and is $\Theta(n_l)$.

(2) For any $(\vec{x}_1, \dots, \vec{x}_k) \in \mathcal{X}_l$, we have $\sum_{i=1}^k \vec{x}_i \in \mathcal{H}$. Let $\vec{x} = \sum_{i=1}^k \vec{x}_i$. By definition we have $\mathbf{E} \cdot (\vec{x})^\top = (\vec{0})^\top$. We also have

$$
\mathbf{S} \cdot (\vec{x})^\top = \mathbf{S} \cdot (\vec{x} - \vec{y}^l)^\top + \mathbf{S} \cdot (\vec{y}^l)^\top = \mathbf{S} \cdot (\vec{x} - \vec{y}^l)^\top + \mathbf{S} \cdot (\vec{y}^{l\#} + \vec{y}^{l*})^\top
$$
$$
\leq \mathbf{S} \cdot (\vec{x} - \vec{y}^l)^\top + \mathbf{S} \cdot (\vec{y}^{l\#})^\top
$$

The inequality follows after recalling that $\vec{y}^{l*} \in \mathcal{H}_{\leq 0}$, which means that $\mathbf{S} \cdot (y^{l*})^\top \leq (\vec{0})^\top$. Notice that $|\sum_{i=1}^k \vec{x}_i - \vec{y}^l|_\infty = O(\zeta)\sqrt{n_l}$ and each element in $\mathbf{S} \cdot (\vec{y}^{l\#})^\top$ is $\Theta(n_l)$. Therefore, when $\zeta$ is sufficiently small we have $\mathbf{S} \cdot (\vec{x})^\top < (\vec{0})^\top$. This means that when $\zeta > 0$ is sufficiently small we have $\vec{x} \in \mathcal{H}$.

For any $l$ with $\frac{\zeta\sqrt{n_l}}{\rho} > 1$, we have:

$$|\mathcal{X}_l| \geq (\frac{1}{\rho})^{q-o-1}\zeta^{(q-1)(k-1)+q-o-1}n_l^{((q-1)(k-1)+q-o-1)/2} = \Omega(n_l^{((q-1)(k-1)+q-o-1)/2}).$$

This is because according to condition (i) there are at least $(\zeta\sqrt{n_l})^{(q-1)(k-1)}$ combinations of values for $\vec{x}_1, \ldots, \vec{x}_{k-1}$, and according to condition (ii) there are at least $(\frac{\zeta\sqrt{n_l}}{\rho})^{q-o-1}$ combinations of values for $x_{k(o+2)}, \ldots, x_{kq}$.

**Step 3.** By the definition of $\mathcal{X}_l$, there exists a constant $\alpha > 0$ that does not depend on $n_l$ such that for each $(\vec{x}_1, \ldots, \vec{x}_k) \in \mathcal{X}_l$ and each $i \leq k$, we have $|\vec{x}_i - \beta_i\pi_i|_\infty < \alpha\sqrt{n_l}$. Also because all agents' preferences are independently generated, we have: $\Pr(\forall i \leq k, \vec{X}_{\vec{\pi}_i} = \vec{x}_i) = \prod_{i=1}^k \Pr(\vec{X}_{\vec{\pi}_i} = \vec{x}_i)$.

We note that for each $i \leq k$, $\vec{X}_{\vec{\pi}_i}$ is the histogram of $\beta_i$ i.i.d. random variables, each of which is distributed as $\pi_i$. The next lemma implies that for each $i \leq k$, $\Pr(\vec{X}_{\vec{\pi}_i} = \vec{x}_i)$ is $\Omega(n_l^{(1-q)/2})$.

**Lemma 4 (Point-wise concentration bound for i.i.d. Poisson multinomial variables).** *Given $q \in \mathbb{N}$, $\epsilon > 0$, $\alpha > 0$. There exists a constant $\beta > 0$ such that for any distribution $\pi$ over $[q]$ that is above $\epsilon$, any $n \in \mathbb{N}$, and any vector $\vec{x} \in \mathbb{Z}_{\geq 0}^q$ with $\vec{x} \cdot \vec{1} = n$ and $|\vec{x} - n\pi|_\infty < \alpha\sqrt{n}$, we have $\Pr(\vec{X}_\pi = \vec{x}) > \beta n^{\frac{1-q}{2}}$, where $\vec{X}_\pi$ is the Poisson multinomial variables corresponding to $n$ i.i.d. random variables, each of which is distributed as $\pi$.*

*Proof.* Let $\vec{x} = (x_1, \ldots, x_q)$, and let $\vec{d} = \vec{x} - n\pi$. We have:

$$\Pr(\vec{X}_\pi = \vec{x}) = \binom{n}{x_1}\binom{n-x_1}{x_2}\cdots\binom{x_{q-1}+x_q}{x_{q-1}}\prod_{i=1}^q \pi_i^{x_i} = \frac{n!}{\prod_{i=1}^q x_i!}\prod_{i=1}^q \pi_i^{x_i}$$

$$\geq \frac{\frac{1}{\sqrt{2\lambda n}}(n/e)^n}{\prod_{i=1}^q(\frac{1}{\sqrt{2\lambda x_i}}e^{1/12}(x_i/e)^{x_i})}\prod_{i=1}^q \pi_i^{x_i} = Cn^{\frac{1-q}{2}}\frac{n^n}{\prod_{i=1}^q x_i^{x_i}}\prod_{i=1}^q \pi_i^{x_i} \tag{7}$$

$$= Cn^{\frac{1-q}{2}}\prod_{i=1}^q\left(\frac{n\pi_i}{x_i}\right)^{n\pi_i}\prod_{i=1}^q\left(\frac{n\pi_i}{x_i}\right)^{x_i-n\pi_i} \geq Cn^{\frac{1-q}{2}}\prod_{i=1}^q\left(\frac{n\pi_i}{x_i}\right)^{d_i} \tag{8}$$

Inequality (7) is due to Robbins' Stirling approximation [56], where $\lambda$ denotes ratio of circumference to diameter ($\pi$ has already been used to denote a distribution). $C$ is a constant that does not depend on $n$. Inequality (8) is because $\prod_{i=1}^q\left(\frac{n\pi_i}{x_i}\right)^{n\pi_i} = \exp\left(nD_{\text{KL}}(\pi\|\frac{\vec{x}}{n})\right)$, where $D_{\text{KL}}(\pi\|\frac{\vec{x}}{n})$ is the KL divergence of $\frac{\vec{x}}{n}$ from $\pi$, which is non-negative, meaning that $\prod_{i=1}^q\left(\frac{n\pi_i}{x_i}\right)^{n\pi_i} \geq 1$. Let $\pi_{\min} = \min_i \pi_i \geq \epsilon$.

$$\prod_{i=1}^q\left(\frac{n\pi_i}{x_i}\right)^{d_i} = \prod_{i=1}^q\frac{1}{(1+\frac{d_i}{n\pi_i})^{d_i}} \geq \prod_{i\leq q:d_i<0}\frac{1}{(1+\frac{d_i}{n\pi_i})^{d_i}} \geq \left((1-\frac{\alpha\sqrt{n}}{n\pi_{\min}})^{\alpha\sqrt{n}}\right)^q$$

$$= \left((1-\frac{\alpha}{\pi_{\min}\sqrt{n}})^{\frac{\pi_{\min}\sqrt{n}}{\alpha}}\right)^{q\alpha^2/\pi_{\min}}$$

As $\lim_{x\to\infty}(1-\frac{1}{x})^x = \frac{1}{e}$, when $n$ is large enough we have $\prod_{i=1}^q\left(\frac{n\pi_i}{x_i}\right)^{d_i} > (\frac{1}{2e})^{q\alpha^2/\pi_{\min}}$ which is a constant that does not depend on $n$. This proves that $\Pr(\vec{X}_\pi = \vec{x}) = \Omega(n^{\frac{1-q}{2}})$. $\square$

By Lemma 4, we have $\Pr(\vec{X}_{\vec{\pi}_i} = \vec{x}_i) = \Omega(\beta_i^{(1-q)/2}) = \Omega(n_l^{(1-q)/2})$.

**Finally,** we have $\Pr(\vec{X}_{\vec{\pi}} \in \mathcal{H}) \geq \Omega(n_l^{\frac{(q-1)(k-1)+q-o-1}{2}}) \times \Omega(n_l^{\frac{(1-q)}{2}})^k = \Omega(n_l^{-\frac{o}{2}})$. Note that we require $l$ to be sufficiently large such that $\frac{\zeta\sqrt{n_l}}{\rho} > 1$ in order to guarantee that $|\mathcal{X}_l|$ is large enough. Because $n_l = \Theta(l)$, there exists a constant $\hat{C}$ such that for any $l \in \mathbb{N}$, $\frac{n_{l+1}}{n_l} < \hat{C}$. Let $C = \max(n_{L+1}, \hat{C})$. This proves the tightness of the upper bound when $\mathcal{H} \neq \emptyset$ and $\mathcal{H}_{\leq 0} \cap \text{CH}(\Pi) \neq \emptyset$. $\square$

## B.1 Examples

**Example 3.** *Let $m = 3$ and $\mathcal{A} = \{1, 2, 3\}$. For any profile $P$, let $x_{123}$ denote the number of $1 \succ 2 \succ 3$ in $P$. The event "alternatives $1$ and $2$ are co-winners under Borda as well as the only two weak Condorcet winner" can be represented by the following constraints.*

$$2(x_{123} + x_{132}) + x_{213} + x_{312} = 2(x_{213} + x_{231}) + x_{123} + x_{321} \tag{9}$$

$$x_{123} + x_{132} + x_{312} = x_{213} + x_{231} + x_{321} \tag{10}$$

$$2(x_{312} + x_{321}) + x_{132} + x_{231} < 2(x_{123} + x_{132}) + x_{213} + x_{312} \tag{11}$$

$$x_{312} + x_{321} + x_{231} < x_{123} + x_{132} + x_{213} \tag{12}$$

$$x_{312} + x_{321} + x_{132} < x_{231} + x_{213} + x_{123} \tag{13}$$

*Equation (9) states that the Borda scores of $1$ and $2$ are the same; equation (10) states that $1$ and $2$ are tied in their head-to-head competition; inequality (11) states that the Borda sore of $1$ is strictly higher than the Borda score of $3$; and inequalities (12) and (13) require that $1$ and $2$ beat $3$ in their head-to-head competitions, respectively.*

**Example 4.** *We show how to obtain $\mathbf{D}$ using the setting in Example 3. Let $\vec{x} = [x_{123}, x_{132}, x_{213}, x_{231}, x_{312}, x_{321}]$. We have $\mathbf{E} = \begin{bmatrix} 1 & 2 & -1 & -2 & 1 & -1 \\ 1 & 1 & -1 & -1 & 1 & -1 \end{bmatrix}$ and $\mathbf{E}' = \begin{bmatrix} \mathbf{E} \\ \vec{1} \end{bmatrix}$.*

*The reduced echelon form of $\mathbf{E}'$ and its corresponding equations can be calculated as follows.*

$$\begin{bmatrix} 1 & 2 & -1 & -2 & 1 & -1 & | & 0 \\ 1 & 1 & -1 & -1 & 1 & -1 & | & 0 \\ 1 & 1 & 1 & 1 & 1 & 1 & | & n \end{bmatrix} \xrightarrow{R1; R2 - R1; R3 - R1} \begin{bmatrix} 1 & 2 & -1 & -2 & 1 & -1 & | & 0 \\ 0 & -1 & 0 & 1 & 0 & 0 & | & 0 \\ 0 & -1 & 2 & 3 & 0 & 2 & | & n \end{bmatrix}$$

$$\xrightarrow{R1 + 2R2; -R2; R3 - R2} \begin{bmatrix} 1 & 0 & -1 & 0 & 1 & -1 & | & 0 \\ 0 & 1 & 0 & -1 & 0 & 0 & | & 0 \\ 0 & 0 & 2 & 2 & 0 & 2 & | & n \end{bmatrix}$$

$$\xrightarrow{R1 + R2/2; R2; R3/2} \begin{bmatrix} 1 & 0 & 0 & 1 & 1 & 0 & | & \frac{n}{2} \\ 0 & 1 & 0 & -1 & 0 & 0 & | & 0 \\ 0 & 0 & 1 & 1 & 0 & 1 & | & \frac{n}{2} \end{bmatrix}$$

*The text above each arrow represents matrix operations. $R1, R2, R3$ represents the first, second, and the third row vector of the matrix on the left. For example, "$R1; R2 - R1; R3 - R1$" represents that the in the right matrix, the first row is R1 of the left matrix; the second row is obtained by subtracting $R1$ from $R2$; and the third row is obtained from subtracting $R1$ from $R3$. Let $I_0 = [x_{123}, x_{132}, x_{213}]$ and $I_1 = [x_{231}, x_{312}, x_{321}]$. We have:*

$$\mathbf{D} = \begin{bmatrix} -1 & -1 & 0 & \frac{1}{2} \\ 1 & 0 & 0 & 0 \\ -1 & 0 & -1 & \frac{1}{2} \end{bmatrix} \text{ and } \begin{bmatrix} x_{123} \\ x_{132} \\ x_{213} \end{bmatrix} = \mathbf{D} \times \begin{bmatrix} x_{231} \\ x_{312} \\ x_{321} \\ n \end{bmatrix}$$

## C Appendix: Proof of Theorem 1

**Theorem 1. (Smoothed likelihood of Codorcet's paradox).** *Let $\mathcal{M} = (\Theta, \mathcal{L}(\mathcal{A}), \Pi)$ be a strictly positive and closed single-agent preference model.*

**Smoothed avoidance of Condorcet's paradox.** *Suppose for all $\pi \in CH(\Pi)$, $UMG(\pi)$ does not contain a weak Condorcet cycle. Then, for any $n \in \mathbb{N}$, we have:*

$$\inf_{\vec{\pi} \in \Pi^n} \mathbb{E}_{P \sim \vec{\pi}} S_{NCC}(P) = 1 - \exp(-\Omega(n))$$

**Smoothed Condorcet's paradox.** *Suppose there exists $\pi \in CH(\Pi)$ such that $UMG(\pi)$ contains a weak Condorcet cycle. Then, there exist infinitely many $n \in \mathbb{N}$ such that:*

$$\inf_{\vec{\pi} \in \Pi^n} \mathbb{E}_{P \sim \vec{\pi}} S_{NCC}(P) = 1 - \Omega(1)$$

*Proof.* The theorem is proved by applying Lemma 1 multiple times, where $C^{\mathbf{ES}}$ represents the profiles whose UMG contains a specific weak Codorcet cycle. Formally, we have the following definitions.

**Definition 10** (**Variables and pairwise constraints**). *For any linear order $R \in \mathcal{L}(\mathcal{A})$, let $x_R$ be a variable that represents the number of times $R$ occurs in a profile. Let $\mathcal{X}_{\mathcal{A}} = \{x_R : R \in \mathcal{L}(\mathcal{A})\}$ and let $\vec{x}_{\mathcal{A}}$ denote the vector of elements of $\mathcal{X}_{\mathcal{A}}$ w.r.t. a fixed order. For any pair of different alternatives $a, b$, let $\mathrm{Pair}_{a,b}(\vec{x}_{\mathcal{A}})$ denote the linear combination of variables in $\mathcal{X}_{\mathcal{A}}$, where for any $R \in \mathcal{L}(\mathcal{A})$, the coefficient of $x_R$ is $1$ if $a \succ_R b$; otherwise the coefficient is $-1$.*

For any profile $P$, $\mathrm{Pair}_{a,b}(\mathrm{Hist}(P))$ is the weight on the $a \succ b$ edge in $\mathrm{WMG}(P)$. It is not hard to check that $\mathrm{Pair}_{a,b}(\vec{1}) = 0$.

**Definition 11.** *For any unweighted directed graph $G$ over $\mathcal{A}$, we define $C^G$ to be the constraints as in Definition 4 that is based on matrices $\mathbf{E}^G$ that represents $\{\mathrm{Pair}_{a,b}(\vec{x}_{\mathcal{A}}) = 0 : (a, b) \notin G \text{ and } (b, a) \notin G\}$ and $\mathbf{S}^G$ that represents $\{\mathrm{Pair}_{b,a}(\vec{x}_{\mathcal{A}}) < 0 : (a, b) \in G\}$. Let $\mathcal{H}^G$ and $\mathcal{H}^G_{\leq 0}$ denote the solutions to $C^G$ and its relaxation $\overline{C}_G$ as in Definition 4, respectively.*

We immediate have the following claim.

**Claim 3.** *For any profile $P$ and any unweighted directed graph $G$ over $\mathcal{A}$, $\mathrm{Hist}(P) \in \mathcal{H}^G$ if and only if $G = \mathrm{UMG}(P)$; $\mathrm{Hist}(P) \in \mathcal{H}^G_{\leq 0}$ if and only if $\mathrm{UMG}(P)$ is a subgraph of $G$. Moreover, $\mathrm{Rank}(\mathbf{E}^G)$ equals to the number of ties in $G$.*

*Proof.* The necessary and sufficient conditions for $\mathrm{Hist}(P) \in \mathcal{H}^G$ and $\mathrm{Hist}(P) \in \mathcal{H}^G_{\leq 0}$ follow after their definitions. Because the number of equations in $\mathbf{E}^G$ equals to the number of ties in $G$, it suffices to prove that the equations in $\mathbf{E}^G$ are independent. This is true because for any pair of alternatives $(a, b)$, the UMG of the following profile of two rankings only contains one edge $[a \to b$: $a \succ b \succ \text{others}]$ and $[\mathrm{rev} \succ a \succ b]$, where rev is the reverse order of other alternatives. $\square$

**Proof of the smoothed avoidance part.** Let $\mathcal{C}$ denote the set of all $C^G$, where $G$ is an unweighted directed graph without weak Condorcet cycles. We have the following observations.

(1) For any profile $P$, $\mathrm{S}_{\mathrm{NCC}}(P) = 0$ if and only if there exists $G \in \mathcal{C}$ such that $\mathrm{Hist}(P) \in \mathcal{H}^G$. To see this, when $\mathrm{S}_{\mathrm{NCC}}(P) = 0$, we have $\mathrm{Hist}(P) \in \mathcal{H}^{\mathrm{UMG}(P)}$ and $\mathrm{UMG}(P) \in \mathcal{C}$; and vice versa, if $\mathrm{Hist}(P) \in \mathcal{H}^G$ for some $G \in \mathcal{C}$, then $\mathrm{S}_{\mathrm{NCC}}(P) = 0$.

(2) For any graph $G$, $\mathcal{H}^G \neq \emptyset$, which follows after McGarvey's theorem [44].

(3) The total number of UMGs over $\mathcal{A}$ only depends on $m$ not on $n$, which means that $|\mathcal{C}|$ can be seen as a constant that does not depend on $n$.

The three observations imply that for any distribution of $P$, we have

$$\Pr(\mathrm{S}_{\mathrm{NCC}}(P) = 0) \leq \sum_{G \in \mathcal{C}} \Pr(\mathrm{Hist}(P) \in \mathcal{H}^G)$$

Therefore, based on observation (3) above, to prove the smoothed avoidance part of the theorem, it suffices to prove that for any $n$, any $\vec{\pi} \in \Pi^n$, and any $G \in \mathcal{C}$, we have

$$\Pr_{P \sim \vec{\pi}}(\mathrm{Hist}(P) \in \mathcal{H}^G) = \exp(-\Omega(n))$$

This follows after the exponential case in Lemma 1 applied to $C^G$. To see this, we first note that $\mathcal{H}^G \neq \emptyset$ according to observation (2) above. Also for each $\pi \in \Pi$, because $\mathrm{UMG}(\pi)$ does not contain a weak Condorcet cycle, $\mathrm{UMG}(\pi)$ is not a subgraph of $G$, which contains a Condorcet cycle. Therefore, $\mathrm{Hist}(\pi) \notin \mathcal{H}^G_{\leq 0}$ due to Claim 3, which means that $\mathcal{H}^G_{\leq 0} \cap \mathrm{CH}(\pi) = \emptyset$.

**Proof of the smoothed paradox part.** Let $\pi \in \mathrm{CH}(\Pi)$ denote any distribution such that $\mathrm{UMG}(\pi)$ contains a weak Condorcet cycle. Let $G$ denote an arbitrary supergraph of $\mathrm{UMG}(\pi)$ that contains a Condorcet cycle, e.g. by completing the weak Condorcet cycle in $\mathrm{UMG}(\pi)$. The weak smoothed paradox part is proved by applying the tightness of the polynomial case in Lemma 1 to $\mathcal{H}^G$ following a similar argument with the proof for the smoothed avoidance part. $\square$

# D    Additional Preliminaries and Examples of Group Theory

After this paper is accepted by NeurIPS, we discovered that a 2015 working paper by Doğan and Giritligil [22] has already used similar notation and ideals to provide an alternative proof for Moulin's elegant characterization of ANR impossibility [49, Problem 1] as well as obtaining a new characterization on ANR impossibility for social welfare functions (that outputs a ranking over $\mathcal{A}$). Their main results are quite different from the smoothed ANR impossibility in this paper.

In Appendix E.1, we provide an alternative proof to Moulin [49]'s characterization of ANR impossibility. At a high level the proof idea is similar to that by Doğan and Giritligil [22] though the details appear different as far as we can tell. We do not claim the group theoretic approach nor the proof in Appendix E.1 contributions of this paper, but still include Appendix E.1 for information and convenience in case a reader is curious about the proof using notation in this paper.

The *symmetric group* over $\mathcal{A} = [m]$, denoted by $\mathcal{S}_\mathcal{A}$, is the set of all permutations over $\mathcal{A}$. A permutation $\sigma$ that maps each $a \in \mathcal{A}$ to $\sigma(a)$ can be represented in two ways.

- *Two-line form:* $\sigma$ is represented by a $2 \times m$ matrix, where the first row is $(1, 2, \ldots, m)$ and the second row is $(\sigma(1), \sigma(2), \ldots, \sigma(m))$.

- *Cycle form:* $\sigma$ is represented by non-overlapping cycles over $\mathcal{A}$, where each cycle $(a_1, \cdots, a_k)$ represent $a_{i+1} = \sigma(a_i)$ for all $i \leq k - 1$, and with $a_1 = \sigma(a_k)$.

It follows that any cycle in the cycle form $(a_1, \cdots, a_k)$ is equivalent to $(a_2, \cdots, a_k, a_1)$. Following the convention, in the cycle form $a_1$ is the smallest elements in the cycle. For example, all permutations in $S_3$ are represented in two-line form and cycle form respective in the Table 1.

Table 1: $\mathcal{S}_{[3]}$ where $m = 3$.

| Two-line | $\begin{pmatrix} 1 & 2 & 3 \\ 1 & 2 & 3 \end{pmatrix}$ | $\begin{pmatrix} 1 & 2 & 3 \\ 2 & 1 & 3 \end{pmatrix}$ | $\begin{pmatrix} 1 & 2 & 3 \\ 1 & 3 & 2 \end{pmatrix}$ | $\begin{pmatrix} 1 & 2 & 3 \\ 3 & 2 & 1 \end{pmatrix}$ | $\begin{pmatrix} 1 & 2 & 3 \\ 2 & 3 & 1 \end{pmatrix}$ | $\begin{pmatrix} 1 & 2 & 3 \\ 3 & 1 & 2 \end{pmatrix}$ |
|---|---|---|---|---|---|---|
| Cycle | () or Id | $(1, 2)$ | $(2, 3)$ | $(1, 3)$ | $(1, 2, 3)$ | $(1, 3, 2)$ |

A *permutation group* $G$ is a subgroup of $\mathcal{S}_\mathcal{A}$ where the identity element is the identity permutation Id, and for any $\sigma, \eta \in \mathcal{S}_\mathcal{A}$, $\sigma \circ \eta$ is the permutation where for any linear order $V \in \mathcal{L}(\mathcal{A})$, $(\sigma \circ \eta)(V) = \sigma(\eta(V))$. For example, $(1, 2) \circ (2, 3) = (1, 2, 3)$, because $1 \succ 2 \succ 3$ is first mapped to $1 \succ 3 \succ 2$ by permutation $(2, 3)$, then to $2 \succ 3 \succ 1$ by permutation $(1, 2)$. There are six subgroups of $\mathcal{S}_{[3]}$: $\{\text{Id}\}$, $\{\text{Id}, (1, 2)\}$, $\{\text{Id}, (2, 3)\}$, $\{\text{Id}, (1, 3)\}$, $\{\text{Id}, (1, 2, 3), (1, 3, 2)\}$, and $\mathcal{S}_{[3]}$.

**Example 5.** *Table 2 shows all permutation groups over $\mathcal{S}_{[3]}$ as the result of Perm$(P)$ for some profiles.*

Table 2: Examples of Perm$(P)$, where 123 represents $1 \succ 2 \succ 3$.

| $P$ | $P[123]$ | $P[132]$ | $P[213]$ | $P[231]$ | $P[312]$ | $P[321]$ | Perm$(P)$ |
|---|---|---|---|---|---|---|---|
| $P_1$ | 1 | 2 | 2 | 2 | 2 | 2 | $\{\text{Id}\}$ |
| $P_2$ | 3 | 5 | 3 | 5 | 4 | 4 | $\{\text{Id}, (1, 2)\}$ |
| $P_3$ | 3 | 5 | 4 | 4 | 5 | 3 | $\{\text{Id}, (1, 3)\}$ |
| $P_4$ | 3 | 3 | 5 | 4 | 5 | 4 | $\{\text{Id}, (2, 3)\}$ |
| $P_5$ | 3 | 5 | 5 | 3 | 3 | 5 | $\{\text{Id}, (1, 2, 3), (1, 3, 2)\}$ |
| $P_6$ | 1 | 1 | 1 | 1 | 1 | 1 | $\mathcal{S}_{[3]}$ |

*$P_1$ in the table consists of one ranking for $1 \succ 2 \succ 3$ and two rankings for each of the remaining five linear orders. Perm$(P_1)$ only contains Id because if it contains any other permutation $\sigma$, then we must have $P[1 \succ 2 \succ 3] = P[\sigma(1 \succ 2 \succ 3)]$, which is impossible. $(12) \in \text{Perm}(P_2)$ because $P_2[1 \succ 2 \succ 3] = P_2[2 \succ 1 \succ 3] = 3$, $P_2[1 \succ 3 \succ 2] = P_2[2 \succ 3 \succ 1] = 5$, and $P_2[3 \succ 1 \succ 2] = P_2[3 \succ 2 \succ 1] = 4$. It is not hard to verify that no other permutations except Id belong to Perm$(P_2)$.*

# E   Proof of Theorem 2

**Theorem 2. (Smoothed ANR (im)possibility theorem).** Let $\mathcal{M} = (\Theta, \mathcal{L}(\mathcal{A}), \Pi)$ be a strictly positive and closed single-agent preference model. Let $\mathcal{U}_m^\Pi = \{U \in \mathcal{U}_m : \exists \pi \in \mathrm{CH}(\Pi), \forall \sigma \in U, \sigma(\pi) = \pi\}$, and when $\mathcal{U}_m^\Pi \neq \emptyset$, let $l_{\min} = \min_{U \in \mathcal{U}_m^\Pi} |U|$ and $l_\Pi = \frac{l_{\min}-1}{l_{\min}} m!$.

**Smoothed possibility.** There exist an anonymous voting rule $r_{\mathrm{ano}}$ and a neutral voting rule $r_{\mathrm{neu}}$ such that for any $r \in \{r_{\mathrm{ano}}, r_{\mathrm{neu}}\}$, any $n$, and any $\vec{\pi} \in \Pi^n$, we have:

$$\Pr_{P \sim \vec{\pi}}(S_{\mathrm{ano}}(r, P) + S_{\mathrm{neu}}(r, P) < 2) = \left\{ \begin{array}{ll} O(n^{-\frac{l_\Pi}{2}}) & \text{if } \mathcal{U}_m^\Pi \neq \emptyset \\ \exp(-\Omega(n)) & \text{otherwise} \end{array} \right.$$

**Smoothed impossibility.** For any voting rule $r$, there exist infinitely many $n \in \mathbb{N}$ such that:

$$\sup_{\vec{\pi} \in \Pi^n} \Pr_{P \sim \vec{\pi}}(S_{\mathrm{ano}}(r, P) + S_{\mathrm{neu}}(r, P) < 2) = \left\{ \begin{array}{ll} \Omega(n^{-\frac{l_\Pi}{2}}) & \text{if } \mathcal{U}_m^\Pi \neq \emptyset \\ \exp(-O(n)) & \text{otherwise} \end{array} \right.$$

*Proof.* The proof is done in three steps.

**Step 1. Identifying the source of the ANR impossibility**.

**Definition 12.** *For any $n \in \mathbb{N}$ and any $m \geq 2$, let $\mathcal{T}_{m,n}$ denote the set of $n$-profiles $P$ such that $\mathrm{Perm}(P) \in \mathcal{U}_m$. That is,*

$$\mathcal{T}_{m,n} = \{P \in \mathcal{L}(\mathcal{A})^n : \mathrm{Perm}(P) \text{ covers } \mathcal{A}\}$$

The following lemma states that profiles in $\mathcal{T}_{m,n}$ are the intrinsic source of the ANR impossibility.

**Lemma 5.** *For any voting rule $r$, any $n \in \mathbb{N}$, any $m \geq 2$, and any $P \in \mathcal{T}_{m,n}$, we have:*

$$S_{ano}(r, P) + S_{neu}(r, P) \leq 1 \tag{14}$$

*Proof.* We first partition $\mathcal{T}_{m,n}$ to $\mathrm{Hist}^{-1}(H_1) \cup \cdots \cup \mathrm{Hist}^{-1}(H_J)$, where $H_1, \ldots, H_J$ are some histograms of $n$ votes. The partition exists because for any $P \in \mathcal{T}_{m,n}$ and any profile $P'$ with $\mathrm{Hist}(P') = \mathrm{Hist}(P)$, we have $\mathrm{Perm}(P') = \mathrm{Perm}(P) \in \mathcal{U}_m$, which means that $P' \in \mathcal{T}_{m,n}$. For any $j \leq J$, if there exist $P_1, P_2 \in \mathrm{Hist}^{-1}(H_j)$ such that $r(P_1) \neq r(P_2)$, then for all $P \in \mathrm{Hist}^{-1}(H_j)$ we have $S_{\mathrm{ano}}(P) = 0$, which means that inequality (14) holds for all $P \in \mathrm{Hist}^{-1}(H_j)$. Otherwise if $r(P) = a$ for all $P \in H_j$, then by the definition of $\mathcal{T}_{m,n}$ there exists a permutation $\sigma$ over $\mathcal{A}$ such that $\mathrm{Hist}(\sigma(P)) = \mathrm{Hist}(P)$ and $\sigma(a) \neq a$. This means that $\sigma(P) \in H_j$, and therefore, $r(\sigma(P)) = a \neq \sigma(r(P))$, which means that $S_{\mathrm{neu}}(P) = 0$. Again, inequality (14) holds for $P \in \mathrm{Hist}^{-1}(H_j)$. This proves the lemma. $\qquad\square$

**Step 2. Smoothed possibility.** First, we define $r_{\mathrm{ano}}$ and $r_{\mathrm{neu}}$ by extending a voting rule $r^*$ that is defined on $\mathcal{L}(\mathcal{A})^n \setminus \mathcal{T}_{m,n}$ and is guaranteed to satisfy anonymity and neutrality simultaneously for all profiles that are not in $\mathcal{T}_{m,n}$.

More precisely, for any pair of profiles $P$ and $P'$, we write $P \equiv P'$ if and only if there exists a permutation $\sigma \in \mathcal{S}_\mathcal{A}$ such that $\mathrm{Hist}(P') = \sigma(\mathrm{Hist}(P))$. Let $(\mathcal{L}(\mathcal{A}) \setminus \mathcal{T}_{m,n}) = D_1 \cup \cdots \cup D_{L'}$ denote the partition w.r.t. $\equiv$. For each $l \leq L'$, let $P_l \in D_l$ denote an arbitrary profile. Because $P_l \notin \mathcal{T}_{m,n}$, there exists an alternative $a_l$ that is not covered by $\mathrm{Perm}(P)$. Then, for any $P' \in D_l$ and any permutation $\sigma$ such that $\mathrm{Hist}(P') = \sigma(\mathrm{Hist}(P))$, we let $r^*(P') = \sigma(a_l)$. We note that the choice of $\sigma$ does not matter, because all permutations $\sigma$ that map $\mathrm{Hist}(P)$ to $\mathrm{Hist}(P')$ have the same image for $a_l$. To see this, for the sake of contradiction, suppose $\mathrm{Hist}(P') = \sigma_1(\mathrm{Hist}(P)) = \sigma_2(\mathrm{Hist}(P))$ where $\sigma_1(a_l) \neq \sigma_2(a_l)$. Then, we have $\sigma_1^{-1} \circ \sigma_2 \in \mathrm{Perm}(P)$, and $(\sigma_1^{-1} \circ \sigma_2)(a_l) \neq a_l$, which is a contradiction to the assumption that $\mathrm{Perm}(P)$ does not cover $a_l$.

**Claim 4.** *For any profile $P \notin \mathcal{T}_{m,n}$ we have $S_{ano}(r^*, P) = 1$ and $S_{neu}(r^*, P) = 1$.*

*Proof.* By definition, for all profiles $P \notin \mathcal{T}_{m,n}$, $r^*(P)$ only depends on $\mathrm{Hist}(P)$, which means that $S_{\mathrm{ano}}(r^*, P) = 1$. To prove $S_{\mathrm{neu}}(r^*, P) = 1$, let $P \in D_l$ for some $l \leq L$ and let $\sigma \in \mathcal{S}_\mathcal{A}$ denote a permutation over alternatives. We first prove that $\sigma(P) \in D_l$. Let $P' = \sigma(P)$. For the sake of

contradiction suppose $P' \notin D_l$. Then we have $P' \in \mathcal{T}_{m,n}$. Therefore, for any alternative $a \in \mathcal{A}$, there exists $\eta \in \text{Perm}(P')$ such that $\eta(\sigma(a)) \neq \sigma(a)$. It follows that $(\sigma^{-1} \circ \eta \circ \sigma) \in \text{Perm}(P)$ and $(\sigma^{-1} \circ \eta \circ \sigma)(a) \neq a$. Therefore, $\text{Perm}(P)$ covers $\mathcal{A}$, which contradicts the assumption that $P \notin \mathcal{T}_{m,n}$. Let $P = \zeta(P_l)$, where $P_l$ is the profile chosen in the definition of $r^*$. It follows that $P' = (\sigma \circ \zeta)(P_l)$. Therefore, we have $r^*(P') = (\sigma \circ \zeta)(a_l) = \sigma(\zeta(a_l)) = \sigma(r^*(P))$. This proves the claim. $\qquad\square$

For any profile $P \notin \mathcal{T}_{m,n}$, we let $r_{\text{ano}}(P) = r_{\text{neu}}(P) = r^*(P)$. For any profile $P$ in $\mathcal{T}_{m,n}$, we let $r_{\text{ano}}(P) = a$ for an arbitrary fixed alternative $a$, and let $r_{\text{neu}}(P)$ be the top-ranked alternative of agent 1. By Claim 4, for any $r \in \{r_{\text{ano}}, r_{\text{neu}}\}$ any $n$, and any $\vec{\pi} \in \Pi^n$, $\text{Pr}_{P \sim \vec{\pi}}(S_{\text{ano}}(r, P) + S_{\text{neu}}(r, P) < 2) \geq \text{Pr}_{P \sim \vec{\pi}}(P \in \mathcal{T}_{m,n})$. Therefore, it suffices to prove the following lemma.

**Lemma 6.** *For any $r \in \{r_{ano}, r_{neu}\}$, any $n$, and any $\vec{\pi} \in \Theta^n$, we have:*

$$\text{Pr}_{P \sim \vec{\pi}}(P \in \mathcal{T}_{m,n}) = \begin{cases} O(n^{-\frac{l_\Pi}{2}}) & \text{if } \mathcal{U}_m^\Pi \neq \emptyset \\ \exp(-\Omega(n)) & \text{otherwise} \end{cases}.$$

*Proof.* The lemma is proved by applying the upper bound in Lemma 1 as in the proof of Theorem 1. For any $U \in \mathcal{U}_m$, we define constrains $\mathbf{C}^U$ as follows.

**Definition 13.** *For any $U \in \mathcal{U}_m$, we define $\mathbf{C}^U$ as in Definition 4 that is based on $\mathbf{E}^U$ and $\mathbf{S}^U$, where $\mathbf{E}^U$ represents $\{x_R - x_{\sigma(R)} = 0 : \forall R \in \mathcal{L}(\mathcal{A}), \forall \sigma \in U\}$ and $\mathbf{S}^U = \emptyset$. Let $\mathcal{H}^U$ and $\mathcal{H}^U_{\leq 0}$ denote the solutions to $\mathbf{C}^U$ and its relaxation $\overline{\mathbf{C}}_U$ as in Definition 4, respectively.*

By definition, $\mathbf{E}^U(x_R) = 0$ if and only if for all $\sigma \in U$, $x_R = \sigma(x_R)$. Because $\mathbf{S}^U = \emptyset$, we have $\mathbf{C}^U = \overline{\mathbf{C}}_U$, which means that $\mathcal{H}^U = \mathcal{H}^U_{\leq 0}$. We have the following claim about $\mathbf{C}^U$.

**Claim 5.** *For any profile $P$ and any $U \in \mathcal{U}_m$, $\text{Hist}(P) \in \mathcal{H}^U$ if and only if for all $\sigma \in U$, $\text{Hist}(P) = \sigma(\text{Hist}(P))$. Moreover, $\text{Rank}(\mathbf{E}^U) = (1 - \frac{1}{|U|})m!$.*

*Proof.* The "if and only if" part follows after the definition. We now prove that $\text{Rank}(\mathbf{E}^U) = (1 - \frac{1}{|U|})m!$. Let $\equiv_U$ denote the relationship over $\mathcal{L}(\mathcal{A})$ such that for any pair of linear orders $R_1, R_2$, $R_1 \equiv_U R_2$ if and only if there exists $\sigma \in U$ such that $R_1 = \sigma(R_2)$. Because $U$ is a permutation group, $\equiv_U$ is an equivalence relationship that partitions $\mathcal{L}(\mathcal{A})$ into $\frac{m!}{|U|}$ groups, each of which has $|U|$ linear orders. It is not hard to see that each equivalent class is characterized by $|U| - 1$ linearly independent equations represented by rows in $\mathbf{E}^U$, which means that $\text{Rank}(\mathbf{E}^U) \leq (|U| - 1)(\frac{m!}{|U|}) = (1 - \frac{1}{|U|})m!$. For any $s < (1 - \frac{1}{|U|})m!$ and any combination of $s$ rows of $\mathbf{E}^U$, denoted by $A$, it is not hard to construct $\vec{x}$ such that $A \cdot (\vec{x})^\top = (\vec{0})^\top$ but $\mathbf{E}^U \cdot (\vec{x})^\top \neq (\vec{0})^\top$, which proves that $\text{Rank}(\mathbf{E}^U) \geq (1 - \frac{1}{|U|})m!$. This proves the claim. $\qquad\square$

Let $\mathcal{C}$ denote the set of all $\mathbf{C}^U$ where $U \in \mathcal{U}_m$ as in Definition 13. We have the following observations.

(1) $\mathcal{C}$ characterizes $\mathcal{T}_{m,n}$, because by Claim 5, for any $P \in \mathcal{T}_{m,n}$ we have $\text{Hist}(P) \in \mathcal{H}^{\text{Perm}(P)}$ and $\text{Perm}(P) \in \mathcal{C}$; and vice versa, for any $\mathbf{C}^U \in \mathcal{C}$ and any $P$ such that $\text{Hist}(P) \in \mathcal{H}^U$, by Claim 5 we have $\text{Perm}(P) \supseteq U$, which means that $\text{Perm}(P)$ covers $\mathcal{A}$, and it follows that $P \in \mathcal{T}_{m,n}$.

(2) For any $U \in \mathcal{U}_m$, $\mathcal{H}^U \neq \emptyset$. This because $\vec{1} \in \mathcal{H}^U$.

(3) $|\mathcal{C}|$ can be seen as a constant that does not depend on $n$, because it is no more than the total number of permutation groups over $\mathcal{A}$.

We now prove the polynomial upper bound when $\mathcal{U}_m^\Pi \neq \emptyset$. For any $U \in \mathcal{U}_m^\Pi$, by Claim 5, we have $\text{Rank}(\mathbf{E}^U) \geq (1 - \frac{1}{|U|})m! \geq (1 - \frac{1}{l_{\min}})m!$. By applying the polynomial upper bound in Lemma 1 to all $\mathbf{C}^U \in \mathcal{C}$, for any $\vec{\pi} \in \Pi^n$, we have:

$$\text{Pr}_{P \sim \vec{\pi}}(P \in \mathcal{T}_{m,n}) \leq \sum_{\mathbf{C}^U \in \mathcal{C}} \text{Pr}_{P \sim \vec{\pi}}(\text{Hist}(P) \in \mathcal{H}^U)$$

$$= \sum_{\mathbf{C}^U \in \mathcal{C}} O\left(n^{-(\frac{|U|-1}{2|U|})m!}\right) \leq |\mathcal{C}|O\left(n^{-(\frac{l_{\min}-1}{2l_{\min}})m!}\right) = O\left(n^{-\frac{l_\Pi}{2}m!}\right)$$

The exponential upper bound is proved similarly, by applying the exponential upper bound in Lemma 1 to all $\mathbf{C}^U \in \mathcal{C}$. This proves the lemma. $\qquad\square$

**Step 3. Smoothed impossibility**. Lemma 5 implies that for any $\vec{\pi} \in \Pi^n$, $\text{Pr}_{P \sim \vec{\pi}}(\text{S}_{\text{ano}}(r, P) + \text{S}_{\text{neu}}(r, P) < 2) \geq \text{Pr}_{P \sim \vec{\pi}}(P \in \mathcal{T}_{m,n})$. Therefore, it suffices to prove the following lemma.

**Lemma 7.** *For any voting rule $r$, there exist infinitely many $n \in \mathbb{N}$ and corresponding $\vec{\pi} \in \Pi^n$, such that:*

$$\text{Pr}_{P \sim \vec{\pi}}(P \in \mathcal{T}_{m,n}) = \left\{ \begin{array}{ll} \Omega(n^{-\frac{l_\Pi}{2}}) & \text{if } \mathcal{U}_m^\Pi \neq \emptyset \\ \exp(-O(n)) & \text{otherwise} \end{array} \right. .$$

*Proof.* Let $\mathcal{C}$ be the set as defined in the proof of Lemma 6.

**Applying the lower bound in Lemma 1.** We first prove the polynomial lower bound. Suppose $\mathcal{U}_m^\Pi \neq \emptyset$ and let $U \in \mathcal{U}_m^\Pi$ denote the permutation group with the minimum size, i.e. $|U| = l_{\min}$. By Claim 5, $\text{Rank}(\mathbf{E}^U) = (1 - \frac{1}{l_{\min}})m!$. By applying the tightness of the polynomial bound part in Lemma 1 to $\mathbf{C}^U$, we have that there exists constant $C > 0$ such that for any $n' \in \mathbb{N}$, there exists $n' \leq n \leq Cn'$ and $\vec{\pi} \in \Pi^n$ such that

$$\text{Pr}_{P \sim \vec{\pi}}(\text{Hist}(P) \in \mathcal{H}^U) = \Omega\left(n^{-(\frac{l_{\min}-1}{2l_{\min}})m!}\right)$$

Note that for any profile $P$ such that $\text{Hist}(P) \in \mathcal{H}^U$, we have $\text{Perm}(P) \supseteq U$, which means that $\text{Perm}(P)$ covers $\mathcal{A}$ and therefore $P \in \mathcal{T}_{m,n}$. Therefore, we have:

$$\text{Pr}_{P \sim \vec{\pi}}(P \in \mathcal{T}_{m,n}) \geq \text{Pr}_{P \sim \vec{\pi}}(\text{Hist}(P) \in \mathcal{H}^U) \geq \Omega\left((Cn)^{-(\frac{l-1}{2l})m!}\right) = \Omega\left(n^{-(\frac{l-1}{2l})m!}\right)$$

The exponential lower bound is proved similarly, by applying the tightness of the exponential bound part in Lemma 1 to an arbitrary $U \in \mathcal{U}_m$. This proves the lemma. $\qquad\square$

This finishes the proof of Theorem 2. $\qquad\square$

### E.1 Connection to the (Non-)Existence of Anonymous and Neutral Voting Rules

As a side note, Lemma 5 and Claim 4 can be used to prove Moulin's characterization of existence of anonymous and neutral voting rules, which was stated as Problem 1 in [49] with hints on the proofs. More hints are given in [50, Problem 9.9]. We present a different proof using the group theoretic notation developed in this paper.

**Claim 6** (Problem 1 in [49])**.** *Fix any $m \geq 2$ and $n \geq 2$, there exists a voting rule that satisfies anonymity and neutrality if and only if $m$ cannot be written as the sum of $n$'s nontrivial divisors.*

*Proof.* By Lemma 5 and Claim 4, there exists a voting rule that satisfies anonymity and neutrality if and only if $\mathcal{T}_{m,n} = \emptyset$. Therefore, it suffices to prove that $\mathcal{T}_{m,n} = \emptyset$ if and only if $m$ cannot be written as the sum of $n$'s nontrivial divisors.

The "if" direction. Suppose for the sake of contradiction that $\mathcal{T}_{m,n} \neq \emptyset$. Let $P \in \mathcal{T}_{m,n} \neq \emptyset$ denote an arbitrary profile. We now partition $\mathcal{A}$ according to the following equivalence relationship $\equiv$. $a \equiv b$ if and only if there exists $\sigma \in \text{Perm}(P)$ such that $\sigma(a) = b$. The partition is well defined because $\text{Perm}(P)$ is a permutation group. In other words, if $a \equiv b$ then $b \equiv a$, because the $\text{Perm}(P)$

is closed under inversion. If $a \equiv b$ (via $\sigma_1$) and $b \equiv c$ (via $\sigma_2$) then $a \equiv c$ because $c = (\sigma_2 \circ \sigma_1)(a)$, and $\sigma_2 \circ \sigma_1 \in \text{Perm}(P)$.

Suppose $\mathcal{A} = \mathcal{A}_1 \cup \cdots \cup \mathcal{A}_L$ is divided into $L$ parts according to $\equiv$. Because $P \in \mathcal{T}_{m,n}$, which means that $\text{Perm}(P)$ covers $\mathcal{A}$, for all $l \leq L$, $|\mathcal{A}_l| \geq 2$. It is not hard to check that for any $l \leq L$, we have $|\mathcal{A}_l|$ divides $|\text{Perm}(P)|$. Also it is not hard to see that $\text{Perm}(P)$ divides $n$. This means that $m = \sum_{l=1}^{L} |\mathcal{A}_l|$ which contradicts the assumption. This proves the "if" direction.

The "only if" direction. Suppose for the sake of contradiction that $m = m_1 + \cdots + m_L$ where each $m_l \geq 2$ and divides $n$. Consider the permutation $\sigma$ whose cycle form consists of $L$ cycles whose sizes are $m_1, \cdots, m_L$, respectively. Let $U$ denote the permutation group generated by $\sigma$. It follows that $|U|$ is the least common multiple of $m_1, \cdots, m_L$, which means that $|U|$ divides $n$. Therefore, it is not hard to construct an $n$-profile $P$ such that for any ranking $R$ and any $\sigma' \in U$, we must have $P[R] = P[\sigma'(R)]$. It follows that $U \subseteq \text{Perm}(P)$, which means that $\text{Perm}(P)$ covers $\mathcal{A}$, and therefore $\in \mathcal{T}_{m,n} \neq \emptyset$. This contradicts the assumption that $\in \mathcal{T}_{m,n} = \emptyset$, which proves the "only if" direction. $\square$

## F  Proof of Lemma 2

**Lemma 2.** *For any $m \geq 2$, let $l^* = \min_{U \in \mathcal{U}_m} |U|$. We have $l^* = 2$ if $m$ is even; $l^* = 3$ if $m$ is odd and $3 \mid m$; $l^* = 5$ if $m$ is odd, $3 \nmid m$, and $5 \mid m$; and $l^* = 6$ for other $m$.*

*Proof.* We prove the lemma by discussing the following cases.

**Case 1: $2 \mid m$.** Let $\sigma = (1,2)(3,4)\cdots(m-1,m)$ denote the permutation that consists of 2-cycles. It follows that $\text{Id} = \sigma^2$ and $\{\text{Id}, \sigma\}$ is a permutation group that covers $[m]$. This means that $l^* = 2$.

**Case 2: $2 \nmid m$ and $3 \mid m$.** Let $\sigma = (1,2,3)(4,5,6)\cdots(m-2,m-1,m)$ denote the permutation that consists of 3-cycles. It follows that $\text{Id} = \sigma^3$ and $\{\text{Id}, \sigma, \sigma^2\}$ is a permutation group that covers $[m]$. This means that $l^* \leq 3$. We now prove that $l^*$ cannot be 2 by contradiction. Suppose for the sake of contradiction that $l^* = 2$ and let $G$ denote a permutation group with $|G| = 2$. Table 3 (part of Table 26.1 in [27]) lists all groups of orders 1 through 5 up to isomorphism. The order of a group is the number of its elements. For any $l \in \mathbb{N}$, a permutation group $G$ of order $l$ is isomorphic to $Z_l$ if and only if $G = \{\text{Id}, \sigma, \ldots, \sigma^{l-1}\}$, where $\sigma$ is an order-$l$ permutation, that is, $\sigma^l = \text{Id}$ and for all $1 \leq i < l$, $\sigma^i \neq \text{Id}$. A permutation group $G$ of order 4 is isomorphic to $Z_2 \oplus Z_2$ if and only if $G = \{\text{Id}, \sigma, \eta, \sigma \circ \eta\}$, where $\sigma, \eta$, and $\sigma \circ \eta$ are three order-2 permutations.

Table 3: All order 1 through 5 groups up to isomorphism.

| Order | 1 | 2 | 3 | 4 | 5 |
|---|---|---|---|---|---|
| Groups | $\{\text{Id}\}$ | $Z_2$ | $Z_3$ | $Z_4$ or $Z_2 \oplus Z_2$ | $Z_5$ |

Therefore, $G = \{\text{Id}, \sigma\}$. Because $G$ covers $\mathcal{A}$, the cycle form of $\sigma$ must consist of 2-cycles that cover all alternatives in $\mathcal{A}$, which means that $2 \mid m$, a contradiction.

**Case 3: $2 \nmid m$, $3 \nmid m$, and $5 \mid m$.** Let $\sigma = (1,2,3,4,5)\cdots(m-4,m-3,m-2,m-1,m)$ denote the permutation that consists of 5-cycles. It follows that $\text{Id} = \sigma^5$ and $\{\text{Id}, \sigma, \sigma^2, \sigma^3, \sigma^4\}$ is a permutation group that covers $[m]$. This means that $l^* \leq 5$. We now prove by contradiction that $l^*$ cannot be 2, 3, or 4. Suppose for the sake of contradiction that $l^* = 2$ and let $G$ denote a permutation group whose order is no more than 4. Because $2 \nmid m$ and $3 \nmid m$, following a similar argument with Case 2 we know that $G$ cannot be isomorphic to $Z_2, Z_3$, or $Z_4$. Therefore, by Table 3, $G$ must be isomorphic to $Z_2 \oplus Z_2$. However, the next claim shows that this is impossible. Therefore, $l^* = 5$.

**Claim 7.** *For any $m$ with $2 \nmid m$, no permutation group in $G \in \mathcal{G}_m$ is isomorphic to $Z_2 \oplus Z_2$, where $\mathcal{G}_m$ is the set of all permutation groups over $\mathcal{A} = [m]$.*

*Proof.* We prove the claim by contradiction. Suppose for the sake of contradiction $2 \nmid m$ and there exists $G \in \mathcal{G}_m$ that is isomorphic to $Z_2 \oplus Z_2$. This means that $G = \{\text{Id}, \sigma, \eta, \sigma \circ \eta\}$, where $\sigma$, $\eta$, and $\sigma \circ \eta$ only contain 2-cycles in their cycle forms, respectively. Because $2 \nmid m$, at least one alternative is not involved in any cycle in $\eta$. W.l.o.g. we let $\{1, \ldots, k\}$ denote the alternatives that

are not involved in any cycle in $\eta$, which means that they are mapped to themselves by $\eta$. It follows that the remaining $m - k$ alternatives are covered by 2-cycles in $\eta$, which means that $k$ is an odd number. For any $i \leq k$, because $G$ covers $\mathcal{A}$, $i$ must be involved in a 2-cycle in $\sigma$, otherwise none of $\eta, \sigma$, or $\sigma \circ \eta$ will map $i$ to a different alternative. Suppose $(i, j)$ is a 2-cycle in $\sigma$. We note that $(\sigma \circ \eta)(i) = j$, which means that $(i, j)$ is a 2-cycle in $\sigma \circ \eta$. This means that $\eta(j) = j$, that is, $j \leq k$. Therefore, $\{1, \ldots, k\}$ consists of 2-cycles in $\sigma$. This means that $2 \mid k$, which is a contradiction. $\quad\square$

**Case 4: $2 \nmid m$, $3 \nmid m$, and $5 \nmid m$.** Let $\sigma = (1, 2, 3)(4, 5) \cdots (m - 1, m)$. It follows that the order of $\sigma$ is 6, which means that $\{\text{Id}, \sigma, \sigma^2, \sigma^3, \sigma^4, \sigma^5\}$ is a permutation group that covers $[m]$. This means that $l^* \leq 6$. The proof for $l^* \geq 6$ is similar with the proof in Case 3. Suppose for the sake of contradiction there exists $G \in \mathcal{G}_m$ with $|G| \leq 5$. Then, by Table 3, $G$ must be isomorphic to $Z_2, Z_3, Z_4, Z_5$, or $Z_2 \oplus Z_2$. However, this is impossible because none of 2, 3, or 5 can divide $m$, and by claim 7, $G$ is not isomorphic to $Z_2 \oplus Z_2$. This proves that $l^* = 6$. $\quad\square$

### F.1 A Corollary of Theorem 2 and Lemma 2

**Corrollary 1.** *Let* $\mathcal{M} = (\Theta, \mathcal{L}(\mathcal{A}), \Pi)$ *be a strictly positive and closed single-agent preference model with* $\pi_{uni} \in CH(\Pi)$. *Let* $l^*$ *be the number in Lemma 2.*

**Smoothed possibility.** *Let* $r \in \{r_{ano}, r_{neu}\}$. *For any $n$ and any $\vec{\pi} \in \Pi^n$, we have:*

$$\text{Pr}_{P \sim \vec{\pi}}(S_{ano}(r, P) + S_{neu}(r, P) < 2) = O\left(n^{-(\frac{l^* - 1}{2l^*})m!}\right)$$

**Smoothed impossibility.** *For any voting rule $r$, there exist infinitely many $n \in \mathbb{N}$ such that:*

$$\sup_{\vec{\pi} \in \Pi^n} \text{Pr}_{P \sim \vec{\pi}}(S_{ano}(r, P) + S_{neu}(r, P) < 2) = \Omega\left(n^{-(\frac{l^* - 1}{2l^*})m!}\right)$$

In particular, Corollary 1 applies to all neutral, strictly positive, and closed models, which contains $\mathcal{M}_{\text{Ma}}^{[\varphi, 1]}$, $\mathcal{M}_{\text{Pl}}^{[\varphi, 1]}$, and IC as special cases. This is because for any neutral model, let $\pi \in \Pi$ denote an arbitrary distribution. Then, for any permutation $\sigma$ over $\mathcal{A}$, $\sigma(\pi) \in \Pi$. Therefore, $\pi_{\text{uni}} = \frac{1}{m!} \sum_{\sigma \in \mathcal{S}_{\mathcal{A}}} \sigma(\pi) \in CH(\Pi)$.

## G  Proof of Proposition 1

**Proposition 1.** *Let $r$ be a voting rule obtained from a positional scoring correspondence by applying* LEX *or* FA. *Let* $\mathcal{M} = (\Theta, \mathcal{L}(\mathcal{A}), \Pi)$ *be a strictly positive and closed single-agent preference model with* $\pi_{uni} \in CH(\Pi)$. *There exist infinitely many $n \in \mathbb{N}$ such that:*

$$\sup_{\vec{\pi} \in \Pi^n} \text{Pr}_{P \sim \vec{\pi}} (S_{ano}(r, P) + S_{neu}(r, P) < 2) = \Omega(n^{-0.5})$$

*Proof.* Suppose $r$ is obtained from a correspondence $c$ by applying LEX or FA. We first prove that any profiles $P$ with $|c(P)| = 2$ violates anonymity or neutrality under $r$. In other words, $S_{ano}(r, P) + S_{neu}(r, P) < 2$. W.l.o.g. suppose $c(P) = \{1, 2\}$. Suppose $r$ is obtained from $c$ by applying LEX-$R$. W.l.o.g. suppose $1 \succ_R 2$. Let $\sigma$ denote the permutation that exchanges 1 and 2. Because $c$ is neutral, $c(\sigma(P)) = \sigma(c(P)) = \{1, 2\}$. By LEX-$R$, $r(P) = r(\sigma(P)) = 1 \neq 2 = \sigma(r(P))$, which violates neutrality. Suppose $r$ is obtained from $c$ by applying FA-$j$. W.l.o.g. suppose $1 \succ 2$ in the $j$-th vote in $P$. Because $c(P) = \{1, 2\}$, there exists a vote in $P$ where $2 \succ 1$. Suppose $2 \succ 1$ in the $j'$-th vote. Let $P'$ denote the profile obtained from $P$ by switching $j$-th and $j'$-th vote. Because $c$ is anonymous, we have $c(P') = \{1, 2\}$. By FA-$j$, $r(P') = 2 \neq 1 = r(P)$, which violates anonymity.

Therefore, for any $n$ and any $\vec{\pi} \in \Pi^n$, we have $\text{Pr}_{P \sim \vec{\pi}} (S_{ano}(r, P) + S_{neu}(r, P) < 2) \geq \text{Pr}_{P \sim \vec{\pi}}(|c(P)| = 2)$. By applying the tightness of polynomial bound part in Lemma 1, it is not hard to prove that for any positional scoring correspondence $c$, there exist infinitely many $n \in \mathbb{N}$ and corresponding $\vec{\pi} \in \Pi^n$ such that $\text{Pr}_{P \sim \vec{\pi}}(|c(P)| = 2) = \Omega(n^{-0.5})$. This proves the proposition. $\quad\square$

# H  Proof of Theorem 3

**Theorem 3.** *Let $\mathcal{M} = (\Theta, \mathcal{L}(\mathcal{A}), \Pi)$ be a strictly positive and closed single-agent preference model with $\pi_{uni} \in CH(\Pi)$. For any voting correspondence $c$ that satisfies anonymity and neutrality, let $r_{MPSR}$ denote the voting rule obtained from $c$ by MPSR-then-TB. For any $n$ and any $\vec{\pi} \in \Pi^n$, we have:*

$$\Pr_{P \sim \vec{\pi}}(S_{ano}(r_{MPSR}, P) + S_{neu}(r_{MPSR}, P) < 2) = O(n^{-\frac{m!}{4}})$$

*Moreover, if TB satisfies anonymity (respectively, neutrality) then $r_{MPSR}$ also satisfies anonymity (respectively, neutrality).*

*Proof.* The upper bound is proved in the following two steps. First, we show that for any profile $P$ such that $\mathrm{MPSR}(P) \neq \emptyset$,

$$S_{ano}(r_{MPSR}, P) + S_{neu}(r_{MPSR}, P) = 2 \tag{15}$$

If $|c(P)| = 1$, then we have $r_{MPSR}(P) = c(P)$. (15) holds because $c$ satisfies neutrality and anonymity. If $\mathrm{MPSR}(P) \neq \emptyset$, then for any permutation $\sigma$ over $\mathcal{A}$, we have $c(\sigma(P)) = \sigma(c(P))$ and it is not hard to see that $\mathrm{MPSR}(\sigma(P)) = \sigma(\mathrm{MPSR}(P))$. Let $\mathrm{MPSR}(P) = V$. Therefore, $r_{MPSR}(\sigma(P))$ is the alternative in $\sigma(c(P))$ that is ranked highest in $\sigma(V)$, which means that $r_{MPSR}(\sigma(P)) = \sigma(r_{MPSR}(P))$. This means that $S_{neu}(r_{MPSR}, P) = 1$. For any profile $P'$ with $\mathrm{Hist}(P') = \mathrm{Hist}(P)$, it is not hard to see that $\mathrm{MPSR}(P') = \mathrm{MPSR}(P)$. Because $c$ satisfies anonymity, we have $c(P') = c(P)$. This means that $S_{ano}(r_{MPSR}, P) = 1$.

Second, we show that $\Pr_{P \sim \vec{\pi}}(\mathrm{MPSR}(P) = \emptyset) = O(n^{-\frac{m!}{4}})$ by applying the polynomial upper bound in Lemma 1 in a way similar to the proof of Theorem 2. For any partition $\mathcal{Q} = \{Q_1, \ldots, Q_L\}$ of $\mathcal{L}(\mathcal{A})$, we define $\mathbf{C}^{\mathcal{Q}}$ as in Definition 4 based on $\mathbf{E}^{\mathcal{Q}}$, which represents $\{x_R - x_{R'} = 0 : \forall l \leq L, \forall R, R' \in Q_l\}$, and $\mathbf{S}^{\mathcal{Q}} = \emptyset$. We have $\mathrm{Rank}(\mathbf{E}^{\mathcal{Q}}) = m! - |\mathcal{Q}|$. Let $\mathcal{C}$ denote the set of all $\mathbf{C}^{\mathcal{Q}}$, where each set in $\mathcal{Q}$ contains at least two linear orders. It is not hard to verify that $\mathcal{C}$ characterizes all $P$ with $\mathrm{MPSR}(P) = \emptyset$, for any $\mathbf{C}^{\mathcal{Q}} \in \mathcal{C}$, $\mathcal{H}^{\mathcal{Q}} \neq \emptyset$ and $\mathrm{Rank}(\mathbf{E}^{\mathcal{Q}}) \geq \frac{m!}{2}$, and $|\mathcal{C}|$ does not depend on $n$. Because for any $\mathbf{C}^{\mathcal{Q}} \in \mathcal{C}$, $\pi_{uni} \in \mathcal{H}^{\mathcal{Q}}_{\leq 0} \cap CH(\Pi)$, we can apply the polynomial upper bound in Lemma 1 to all $\mathbf{C}^{\mathcal{Q}} \in \mathcal{C}$, which gives us $\Pr_{P \sim \vec{\pi}}(\mathrm{MPSR}(P) = \emptyset) = O(n^{-\frac{m!}{4}})$.

The "moreover" part follows after noticing that (1) for any pair of profiles $P$ and $P'$ such that $\mathrm{MPSR}(P) = \emptyset$ and $\mathrm{Hist}(P') = \mathrm{Hist}(P)$, we have $\mathrm{MPSR}(P') = \emptyset$, and (2) for any profile $P$ with $\mathrm{MPSR}(P) = \emptyset$ and any permutation $\sigma$ over $\mathcal{A}$, we have $\mathrm{MPSR}(\sigma(P)) = \emptyset$. □

# I  Smoothed Likelihood of Other Commonly-Studied Events in Social Choice

In this section we show how to apply Lemma 1 to obtain dichotomy results on smoothed likelihood of various social choice events. Let us start with a dichotomy result on the non-existence of Condorcet cycles.

**Proposition 2 (Smoothed likelihood of non-existence of Condorcet cycles).** *Let $\mathcal{M} = (\Theta, \mathcal{L}(\mathcal{A}), \Pi)$ be a strictly positive and closed single-agent preference model.*

**Upper bound.** *For any $n \in \mathbb{N}$ and any $\vec{\pi} \in \Pi^n$, we have:*

$$\Pr_{P \sim \vec{\pi}}(S_{NCC}(P) = 1) = \begin{cases} O(1) & \text{if } \exists \pi \in CH(\Pi) \text{ s.t. } UMG(\pi) \text{ is acyclic} \\ \exp(-\Omega(n)) & \text{otherwise} \end{cases}.$$

**Tightness of the upper bound.** *There exist infinitely many $n \in \mathbb{N}$ such that:*

$$\sup_{\vec{\pi} \in \Pi^N} \Pr_{P \sim \vec{\pi}}(S_{NCC}(P) = 1) = \begin{cases} \Omega(1) & \text{if } \exists \pi \in CH(\Pi) \text{ s.t. } UMG(\pi) \text{ is acyclic} \\ \exp(-O(n)) & \text{otherwise} \end{cases}.$$

*Proof.* The proof proceeds in the following three steps.

**First step: defining $\mathcal{C}$.** Let $\mathcal{C}$ denote the set of all $\mathbf{C}^G$ where $G$ is an acyclic unweighted directed graph, as in Definition 11. We have the following observations.

(1) For any profile $P$, $\mathrm{S_{NCC}}(P) = 1$ if and only if $\mathrm{Hist}(P) \in \bigcup_{\mathbf{C}^G \in \mathcal{C}} \mathcal{H}^G$. To see this, if $\mathrm{S_{NCC}}(P) = 1$ then $\mathrm{UMG}(P)$ is acyclic, which means that $\mathrm{Hist}(P) \in \mathcal{H}^{\mathrm{UMG}(P)}$, where $\mathbf{C}^{\mathrm{UMG}(P)} \in \mathcal{C}$; and conversely, if $\mathrm{Hist}(P) \in \mathcal{H}^G$ for an acyclic graph $G$, then $\mathrm{S_{NCC}}(P) = 1$.

(2) For any graph $G$, $\mathcal{H}^G \neq \emptyset$ by McGarvey's theorem [44].

(3) $|\mathcal{C}|$ only depends on $m$, which means that $|\mathcal{C}|$ can be seen as a constant that does not depend on $n$.

**Second step, the $O(1)$ case.** The $O(1)$ upper bound is straightforward. To prove its tightness, suppose there exists $\pi \in \mathrm{CH}(\Pi)$ such that $\mathrm{UMG}(\pi)$ is acyclic. This means that there exists a topological ordering of $\mathrm{UMG}(\pi)$. Let $G$ denote an arbitrary complete acyclic supergraph of $\mathrm{UMG}(\pi)$. It follows that $\mathbf{C}^G \in \mathcal{C}$, and $\pi \in \mathcal{H}_{\leq 0}^G$ due to Claim 3. Therefore, $\mathbf{C}^G \cap \mathrm{CH}(\Pi) \neq \emptyset$. Also note that $\mathbf{E}^G = \emptyset$, which means that $\mathrm{Rank}(\mathbf{E}^G) = 0$. The tightness follows after applying the tightness of the polynomial part in Lemma 1 to $\mathbf{C}^G$.

**Third step, the exponential case.** For any $\mathbf{C}^G \in \mathcal{C}$ and any $\pi \in \mathrm{CH}(\Pi)$, we first show that $\mathcal{H}_{\leq 0}^G \cap \mathrm{CH}(\Pi) = \emptyset$. Suppose for the sake of contradiction there exists an acyclic graph $G$ such that $\pi \in \mathcal{H}_{\leq 0}^G \cap \mathrm{CH}(\Pi)$. Then, by Claim 3, $\mathrm{UMG}(\pi)$ is a subgraph of $G$, which means that $\mathrm{UMG}(\pi)$ is acyclic. This contradicts the assumption on $\mathrm{CH}(\Pi)$ in the exponential case. The upper bound (respectively, its tightness) follows after applying the exponential upper bound (respectively, its tightness) in Lemma 1 to all $\mathbf{C}^G \in \mathcal{C}$ (respectively, an arbitrary $\mathbf{C}^G \in \mathcal{C}$). $\qquad\square$

The dichotomy results in this section will be presented by the following template exemplified by Proposition 2. In the template, we will specify three components:

- EVENT, which is an event of interest that depends on the profile $P$,
- CONDITION, which is often about the existence of $\pi \in \mathrm{CH}(\Pi)$ that satisfies a weaker version of EVENT, and
- a number $l_\Pi$ that depends on the statistical model $\mathcal{M}$.

**Template for dichotomy results (Smoothed likelihood of EVENT).** *Let $\mathcal{M} = (\Theta, \mathcal{L}(\mathcal{A}), \Pi)$ be a strictly positive and closed single-agent preference model.*

**Upper bound.** *For any $n \in \mathbb{N}$ and any $\vec{\pi} \in \Pi^n$, we have:*

$$\mathrm{Pr}_{P \sim \vec{\pi}}(\text{EVENT}) = \begin{cases} O(n^{-\frac{l_\Pi}{2}}) & \text{if CONDITION holds} \\ \exp(-\Omega(n)) & \text{otherwise} \end{cases}.$$

**Tightness of the upper bound.** *There exist infinitely many $n \in \mathbb{N}$ such that:*

$$\sup_{\vec{\pi} \in \Pi^N} \mathrm{Pr}_{P \sim \vec{\pi}}(\text{EVENT}) = \begin{cases} \Omega(n^{-\frac{l_\Pi}{2}}) & \text{if CONDITION holds} \\ \exp(-O(n)) & \text{otherwise} \end{cases}.$$

For example, in Proposition 2, EVENT is "there is no Condorcet cycle", CONDITION is "there exists $\pi \in \mathrm{CH}(\Pi)$ such that $\mathrm{UMG}(\pi)$ is acyclic", and $l_\Pi = 0$. Table 4 summarizes the dichotomy results using the template. A *Condorcet winner* is the alternative who beats every other alternative in their head-to-head competition. If a Condorcet winner exists, then it must be unique. A *weak Condorcet winner* is an alternative who never loses in head-to-head competitions. Weak Condorcet winners may not be unique.

Any dichotomy result using the template is quite general, because it applies to all strictly positive and closed single-agent preference models, any $n$, and any combination of distributions. In particular, it is not hard to verify that if $\pi_{\mathrm{uni}} \in \mathrm{CH}(\Pi)$, where $\pi_{\mathrm{uni}}$ is the uniform distribution over $\mathcal{L}(\mathcal{A})$, then CONDITION is satisfied for all propositions in Table 4, which means that the polynomial bounds apply. For example, because $\pi_{\mathrm{uni}} \in \mathrm{CH}(\Pi)$ for any neutral model (for any $\pi \in \Pi$, the average of $m!$ distributions obtained from $\pi$ by applying all permutations is $\pi_{\mathrm{uni}}$), we have the following corollary.

**Corollary 2.** *The polynomial bounds in Table 4 apply to all neutral, strictly positive, and closed models including $\mathcal{M}_{Ma}^{[\varphi,1]}$ and $\mathcal{M}_{Pl}^{[\varphi,1]}$ for all $0 < \underline{\varphi} \leq 1$, and IC, which corresponds to $\Pi = \{\pi_{uni}\}$.*

Table 4: Summary of dichotomy results on smoothed likelihood of events. CH($\Pi$) is the convex hull of $\Pi$. UMG is the unweighted majority graph.

| Prop. | EVENT | CONDITION | $l_\Pi$ |
|---|---|---|---|
| 2 | No Condorcet cycles | $\exists \pi \in$ CH($\Pi$) s.t. UMG($\pi$) is acyclic | 0 |
| 3 | $\exists$ Condorcet cycle of length $k$ | $\exists \pi \in$ CH($\Pi$) s.t. UMG($\pi$) contains a weak Condorcet cycle of length $k$ | 0 |
| 4 | $\exists$ Condorcet winner | $\exists \pi \in$ CH($\Pi$) that has at least one weak Condorcet winner | 0 |
| 5 | No Condorcet winner | $\exists \pi \in$ CH($\Pi$) and a supergraph $G$ of UMG($\pi$) that has no weak Condorcet winner | 0 or 1 |
| 6 | $\exists$ exactly $k$ weak Condorcet winners | $\exists \pi \in$ CH($\Pi$) that contains at least $k$ weak Condorcet winners | $\dfrac{k(k-1)}{2}$ |
| 7 | No weak Condorcet winners | $\exists \pi \in$ CH($\Pi$) and a supergraph $G$ of UMG($\pi$) that has no weak Condorcet winner | 0 |

In light of Corollary 2 and as a result of Proposition 3, the likelihood of Condorcet voting paradox is asymptotically maximized under IC, among all i.i.d. distributions over $\mathcal{L}(\mathcal{A})$. This givens an asymptotic answer to an open questions by Tsetlin et al. [63].

If EVENT is desirable, such as "Condorcet winner" or "No Condorcet cycles", then a polynomial (sometimes $\Theta(1)$) smoothed likelihood is desirable; if EVENT is undesirable, such as "No Condorcet winner" or "there exists a Condorcet cycle of length $k$", then an exponential likelihood is desirable.

**Overview of proof techniques.** All propositions are proved by applying Lemma 1 in the following three steps exemplified by the proof of Proposition 2. **First, we define a set** $\mathcal{C}$ **of constraints** $C^{ES}$'s such that

(1) EVENT is characterized by $\bigcup_{C^{ES} \in \mathcal{C}} \mathcal{H}$ in the sense that EVENT holds for a profile $P$ if and only if Hist($P$) $\in \mathcal{H}$ for some $C^{ES} \in \mathcal{C}$,

(2) for each $C^{ES} \in \mathcal{C}$, $\mathcal{H} \neq \emptyset$, and

(3) $|\mathcal{C}|$ is a constant that does not depend on $n$ (but may depend on $m$).

**Second, for the polynomial bound**, the upper bound is proved by applying the polynomial upper bound in Lemma 1 to all (constant number of) $C^{ES} \in \mathcal{C}$. The tightness is proved by explicitly choosing $C^{ES} \in \mathcal{C}$, often as a function of some $\pi \in$ CH($\Pi$), so that $\pi \in \mathcal{H}_{\leq 0}$, which implies $\mathcal{H}_{\leq 0} \cap$ CH($\Pi$) $\neq \emptyset$, and then applying the tightness of the polynomial bound in Lemma 1. **Third, for the exponential bound**, we first prove that for any $C^{ES} \in \mathcal{C}$, $\mathcal{H}_{\leq 0} \cap$ CH($\Pi$) $= \emptyset$. Then, the upper bound (respectively, its tightness) is proved by applying the exponential bound (respectively, its tightness) in Lemma 1 to all $C^{ES} \in \mathcal{C}$ (respectively, an arbitrary $C^{ES} \in \mathcal{C}$).

**Definition 14.** *For any $3 \leq k \leq m$, we let $S_{CC=k}(P) = 1$ (respectively, $S_{WCC=k}(P) = 1$) if there exists a Condorcet cycle (respectively, weak Condorcet cycle) of length $k$ in $P$; otherwise $S_{CC=k}(P) = 0$ (respectively, $S_{WCC=k}(P) = 0$).*

**Proposition 3 (Smoothed likelihood of existence of Condorcet cycles with length $k$).** *Let $\mathcal{M} = (\Theta, \mathcal{L}(\mathcal{A}), \Pi)$ be a strictly positive and closed single-agent preference model.*

**Upper bound.** *For any $n \in \mathbb{N}$, any $\vec{\pi} \in \Pi^n$, and any $3 \leq k \leq m$, we have:*

$$\Pr_{P \sim \vec{\pi}}(S_{CC=k}(P)) = \begin{cases} O(1) & \text{if } \exists \pi \in CH(\Pi) \text{ s.t. } S_{WCC=k}(\pi) = 1 \\ \exp(-\Omega(n)) & \text{otherwise} \end{cases} .$$

**Tightness of the upper bound.** *For any $3 \leq k \leq m$, there exist infinitely many $n \in \mathbb{N}$ such that:*

$$\sup_{\vec{\pi} \in \Pi^N} \Pr_{P \sim \vec{\pi}}(S_{CC=k}(P)) = \begin{cases} \Omega(1) & \text{if } \exists \pi \in CH(\Pi) \text{ s.t. } S_{WCC=k}(\pi) = 1 \\ \exp(-O(n)) & \text{otherwise} \end{cases} .$$

*Proof.* **First step, defining $\mathcal{C}$.** For any length-$k$ cycle $p = a_1 \to a_2 \to \cdots \to a_k \to a_1$ in $\mathcal{A}$, we define $\mathrm{C}^p$, where $\mathbf{E}^p = \emptyset$ and $\mathbf{S}^p$ represents the $k$ constraints

$$\{\mathrm{Pair}_{a_2,a_1}(\vec{x}_{\mathcal{A}}) < 0, \mathrm{Pair}_{a_3,a_2}(\vec{x}_{\mathcal{A}}) < 0, \ldots, \mathrm{Pair}_{a_1,a_k}(\vec{x}_{\mathcal{A}}) < 0\}$$

Let $\mathcal{C}$ denote all such $\mathrm{C}^p$. We have the following observations.

(1) For any profile $P$, $\mathrm{S}_{\mathrm{CC}=k}(P) = 1$ if and only if $\mathrm{Hist}(P) \in \bigcup_{\mathrm{C}^p \in \mathcal{C}} \mathcal{H}^p$. To see this, if $\mathrm{S}_{\mathrm{CC}=k}(P) = 1$ then there exists a length-$k$ cycle $p$ in $\mathrm{UMG}(P)$, which means that $\mathrm{Hist}(P) \in \mathcal{H}^p$, where $\mathrm{C}^p \in \mathcal{C}$; and conversely, if $\mathrm{Hist}(P) \in \mathcal{H}^p$ for some length-$k$ cycle $p$, then $p$ is a length-$k$ Condorcet cycle in $P$, which means that $\mathrm{S}_{\mathrm{CC}=k}(P) = 1$.

(2) by McGarvey's theorem [44], for each $\mathrm{C}^p \in \mathcal{C}$, there exists a profile $P$ where $p$ is a cycle in $\mathrm{UMG}(P)$, which means that $\mathcal{H}^p \neq \emptyset$.

(3) The total number of length-$k$ cycles in $\mathcal{A}$ only depends on $m$ and $k$, which means that $|\mathcal{C}|$ can be seen as a constant that does not depend on $n$.

**Second step, the polynomial case.** The $O(1)$ upper bound is straightforward. To prove the tightness, suppose there exists $\pi \in \mathrm{CH}(\Pi)$ with $\mathrm{S}_{\mathrm{WCC}=k}(\pi) = 1$. Let $p$ denote an arbitrary length-$k$ weak Condorcet cycle in $\mathrm{UMG}(\pi)$. It follows that $\mathbf{S}^p \cdot (\pi)^\top \leq (\vec{0})^\top$, which means that $\mathcal{H}^p_{\leq 0} \cap \mathrm{CH}(\Pi) \neq \emptyset$, because $\mathbf{E}^p = \emptyset$. The tightness follows after applying the tightness of the polynomial lower bound in Lemma 1 to $\mathrm{C}^p$, where $\mathrm{Rank}(\mathbf{E}^p) = 0$.

**Third step, the exponential case.** For any $\mathrm{C}^p \in \mathcal{C}$ and any $\pi \in \mathrm{CH}(\Pi)$, we first prove that $\mathcal{H}^p_{\leq 0} \cap \mathrm{CH}(\Pi) = \emptyset$. Suppose for the sake of contradiction that there exists a length-$k$ cycle $p$ such that $\pi \in \mathcal{H}^p_{\leq 0} \cap \mathrm{CH}(\Pi)$. Then, because $\mathbf{S}^p \cdot (\pi)^\top \leq (\vec{0})^\top$, for any edge $a_i \to a_{i+1}$ in $p$ we must have $\pi[a_i, a_{i+1}] - \pi[a_{i+1}, a_i] \geq 0$, which means that $p$ is a length-$k$ weak Condorcet cycle in $\mathrm{UMG}(\pi)$, meaning that $\mathrm{S}_{\mathrm{WCC}=k}(\pi) = 1$. This contradicts the assumption on $\mathrm{CH}(\Pi)$ in the exponential case. The upper bound (respectively, its tightness) follows after applying the exponential bound (respectively, its tightness) in Lemma 1 to all (respectively, an arbitrary) $\mathrm{C}^p \in \mathcal{C}$. $\square$

**Definition 15.** *For any profile $P$, let $\mathrm{S}_{CW}(P) = 1$ if $P$ has a Condorcet winner; otherwise let $\mathrm{S}_{CW}(P) = 0$. Let $WCW(P)$ denote the number of weak Condorcet winners in $P$.*

**Proposition 4 (Smoothed likelihood of existence of Condorcet winner).** *Let $\mathcal{M} = (\Theta, \mathcal{L}(\mathcal{A}), \Pi)$ be a strictly positive and closed single-agent preference model.*

**Upper bound.** *For any $n \in \mathbb{N}$ and any $\vec{\pi} \in \Pi^n$, we have:*

$$\Pr_{P \sim \vec{\pi}}(\mathrm{S}_{CW}(P) = 1) = \begin{cases} O(1) & \text{if } \exists \pi \in CH(\Pi) \text{ s.t. } WCW(\pi) \geq 1 \\ \exp(-\Omega(n)) & \text{otherwise} \end{cases}.$$

**Tightness of the upper bound.** *There exist infinitely many $n \in \mathbb{N}$ such that:*

$$\sup_{\vec{\pi} \in \Pi^n} \Pr_{P \sim \vec{\pi}}(\mathrm{S}_{CW}(P) = 1) = \begin{cases} \Omega(1) & \text{if } \exists \pi \in CH(\Pi) \text{ s.t. } WCW(\pi) \geq 1 \\ \exp(-O(n)) & \text{otherwise} \end{cases}.$$

*Proof.* **First step: defining $\mathcal{C}$.** Let $\mathcal{C} = \{\mathrm{C}^G : \mathrm{S}_{CW}(G) = 1\}$, that is, $\mathcal{C}$ contains all $\mathrm{C}^G$ (Definition 11) where $G$ contains a Condorcet winner. We have the following observations.

(1) For any profile $P$, $\mathrm{S}_{CW}(P) = 1$ if and only if $\mathrm{Hist}(P) \in \bigcup_{\mathrm{C}^G \in \mathcal{C}} \mathcal{H}^G$. To see this, if $\mathrm{S}_{CW}(P) = 1$ then $\mathrm{Hist}(P) \in \mathcal{H}^{\mathrm{UMG}(P)} \in \mathcal{C}$, where $\mathrm{C}^G \in \mathcal{C}$; and conversely, if $\mathrm{Hist}(P) \in \mathcal{H}^G$ for some $\mathrm{C}^G \in \mathcal{C}$, then $P$ has a Condorcet winner.

(2) For any $G$ with $\mathrm{S}_{CW}(G) = 1$, $\mathcal{H}^G \neq \emptyset$ due to McGarvey's theorem [44].

(3) $|\mathcal{C}|$ can be seen as a constant that does not depend on $n$.

**Second step: the polynomial case.** The $O(1)$ upper bound trivially holds. To prove the tightness, suppose there exits $\pi \in \mathrm{CH}(\Pi)$ such that $\mathrm{WCW}(\pi) \geq 1$. Let $a$ denote an arbitrary weak Condorcet winner in $\mathrm{UMG}(\pi)$. We obtain a complete graph $G^*$ from $\mathrm{UMG}(\pi)$ by adding $a \to b$ for all tied

pairs $(a, b)$ in UMG($\pi$), and then adding arbitrary edges between other tied pairs in UMG($\pi$). It follows that $S_{CW}(G^*) = 1$ and $\mathbf{E}^{G^*} = \emptyset$ because $G^*$ is complete, which means that $\text{Rank}(\mathbf{E}^{G^*}) = 0$. The tightness follows after applying the tightness of the polynomial bound in Lemma 1 to $\mathbf{C}^{G^*}$.

**Third step: the exponential case.** For any $\mathbf{C}^G \in \mathcal{C}$ and any $\pi \in \text{CH}(\Pi)$, we now prove that $\mathcal{H}_{\leq 0}^G \cap \text{CH}(\Pi) = \emptyset$. Suppose for the sake of contradiction there exist $\mathbf{C}^G \in \mathcal{C}$ and $\pi \in \mathcal{H}_{\leq 0}^G \cap \text{CH}(\Pi)$. By Claim 3, UMG($\pi$) is a subgraph of $G$, which means that the Condorcet winner in $G$ is a weak Condorcet winner in UMG($\pi$), which contradicts the assumption that $\text{WCW}(\pi) = 0$ in the exponential case. The upper bound (respectively, its tightness) follows after applying the exponential bound (respectively, its tightness) in Lemma 1 to all (respectively, an arbitrary) $\mathbf{C}^G \in \mathcal{C}$. $\square$

**Proposition 5** (**Smoothed likelihood of non-existence of Condorcet winner**). *Let $\mathcal{M} = (\Theta, \mathcal{L}(\mathcal{A}), \Pi)$ be a strictly positive and closed single-agent preference model. Let $\mathcal{G}_\Pi$ denote the set of all unweighted directed graphs $G$ over $\mathcal{A}$ such that (1) there exists $\pi \in CH(\Pi)$ such that $UMG(\pi) \subseteq G$, and (2) $S_{CW}(G) = 0$. When $\mathcal{G}_\Pi \neq \emptyset$, we let $l_\Pi = \min_{G \in \mathcal{G}} Ties(G)$, where $Ties(G)$ denote the number of unordered pairs that are tied in $G$.*

**Upper bound.** *For any $n \in \mathbb{N}$ and any $\vec{\pi} \in \Pi^n$, we have:*

$$\Pr_{P \sim \vec{\pi}}(S_{CW}(P) = 0) = \begin{cases} O(n^{-\frac{l_\Pi}{2}}) & \text{if } \mathcal{G}_\Pi \neq \emptyset \\ \exp(-\Omega(n)) & \text{otherwise} \end{cases}.$$

**Tightness of the upper bound.** *There exist infinitely many $n \in \mathbb{N}$ such that:*

$$\sup_{\vec{\pi} \in \Pi^N} \Pr_{P \sim \vec{\pi}}(S_{CW}(P) = 0) = \begin{cases} \Omega(n^{-\frac{l_\Pi}{2}}) & \text{if } \mathcal{G}_\Pi \neq \emptyset \\ \exp(-O(n)) & \text{otherwise} \end{cases}.$$

*Proof.* **First step: defining $\mathcal{C}$.** Let $\mathcal{C} = \{\mathbf{C}^G : S_{CW}(G) = 0\}$, where $\mathbf{C}^G$ is defined in Definition 11. That is, $\mathcal{C}$ contains all $\mathbf{C}^G$ where $G$ does not contain a Condorcet winner. We have the following observations.

(1) For any profile $P$, $S_{CW}(P) = 0$ if and only if $\text{Hist}(P) \in \bigcup_{\mathbf{C}^G \in \mathcal{C}} \mathcal{H}^G$. To see this, if $S_{CW}(P) = 0$ then $\text{Hist}(P) \in \mathcal{H}^{\text{UMG}(P)}$, where $\mathbf{C}^{\text{UMG}(P)} \in \mathcal{C}$; and conversely, if $\text{Hist}(P) \in \mathcal{H}^G$ for some $\mathbf{C}^G \in \mathcal{C}$, then $P$ does not has a Condorcet winner.

(2) For any $G$ with $S_{CW}(G) = 0$, we have $\mathcal{H}^G \neq \emptyset$ due to McGarvey's theorem [44].

(3) $|\mathcal{C}|$ can be seen as a constant that does not depend on $n$.

**Second step: the polynomial case.** To prove the upper bound, we note that for any $G$ such that (1) there is no Condorcet winner and (2) $G$ is a supergraph of the UMG of some $\pi \in \text{CH}(\Pi)$, the number of ties in $G$ is at least $l_\Pi$, which means that $\text{Rank}(\mathbf{E}^G) \geq l_\Pi$. Therefore, according to observation (1) above, we have:

$$\Pr_{P \sim \vec{\pi}}(\text{WCW}(P) = k) \leq \sum_{G : \mathbf{C}^G \in \mathcal{C}} \Pr_{P \sim \vec{\pi}}(\text{Hist}(P) \in \mathcal{H}^G) = O\left(n^{-\frac{l_\Pi}{2}}\right)$$

The last part follows after applying the polynomial upper bound in Lemma 1 to all $\mathbf{C}^G \in \mathcal{C}$ and the observation (3) above. In particular, for any graph $G$ that is not a supergraph of the UMG of any $\pi \in \text{CH}(\Pi)$, $\Pr_{P \sim \vec{\pi}}(\text{Hist}(P) \in \mathcal{H}^G)$ is exponentially small due to Claim 3 and the exponential upper bound in Lemma 1 applied to $\mathbf{C}^G$.

To prove the tightness, let $\pi \in \text{CH}(\Pi)$ denote a distribution such that there exists a supergraph $G^* \in \mathcal{G}_\Pi$ of UMG($\pi$) where $G^*$ contains $l_\Pi$ ties. By Claim 3, $\pi \in \mathcal{H}_{\leq 0}^{G^*}$, which means that $\mathcal{H}_{\leq 0}^{G^*} \cap \text{CH}(\Pi) \neq \emptyset$. Also by Claim 3, $\text{Rank}(\mathbf{E}^{G^*}) = l_\Pi$. The tightness follows after applying the polynomial tightness in Lemma 1 to $\mathbf{C}^{G^*}$.

**Third step: the exponential case.** For any $\mathbf{C}^G \in \mathcal{C}$ and any $\pi \in \text{CH}(\Pi)$, we first prove that $\mathcal{H}_{\leq 0}^G \cap \text{CH}(\Pi) = \emptyset$. Suppose for the sake of contradiction that such $\mathbf{C}^G \in \mathcal{C}$ and $\pi \in \mathcal{H}_{\leq 0}^G \cap \text{CH}(\Pi)$

exist. It follows from Claim 3 that UMG($\pi$) is a subgraph of $G$, which means that $G \in \mathcal{G}_\Pi$. This contradicts the assumption that $\mathcal{G} = \emptyset$. The upper bound (respectively, its tightness) follows after applying the exponential bound (respectively, its tightness) in Lemma 1 to all (respectively, an arbitrary) $C^G \in \mathcal{C}$. $\square$

The following claim implies that $l$ in Proposition 5 can only be 0 or 1.

**Claim 8.** *For any unweighted directed graph $G$ over $\mathcal{A}$ that does not contain a Condorcet winner, there exists a supergraph of $G$, denoted by $G^*$, such that $G^*$ does not contain a Condorcet winner, and the number of ties in $G^*$ is no more than one. The upper bound of one is tight.*

*Proof.* Let $G^*$ denote a supergraph of $G$ without a Condorcet winner and with the minimum number of ties. If there is no tie in $G^*$ then the claim is proved. For any pair of tied alternatives $a$ and $b$ in $G^*$, adding $a \to b$ to $G^*$ must lead to a Condorcet winner due to the minimality of $G^*$, and $a$ must be the Condorcet winner. This means that $a$ beats all alternatives other than $b$ in $G^*$. Similarly, $b$ beats all alternatives other than $a$ in $G^*$. This means that $\{a, b\}$ is the only tie in $G^*$ because if there exists another tie $\{c, d\}$, then adding $c \to d$ to $G^*$ will not introduce a Condorcet winner, which contradicts the minimality of $G^*$. Therefore, the number of ties in $G^*$ is upper bounded by 1. The tightness of the upper bound is proved by letting $G$ be a graph where $a$ and $b$ are the only weak Condorcet winners. $\square$

**Proposition 6** (**Smoothed likelihood of exactly $k$ weak Condorcet winners**). *Let $\mathcal{M} = (\Theta, \mathcal{L}(\mathcal{A}), \Pi)$ be a strictly positive and closed single-agent preference model.*

**Upper bound.** *For any $n \in \mathbb{N}$, any $1 \le k \le m$, and any $\vec{\pi} \in \Pi^n$, we have:*

$$\Pr_{P \sim \vec{\pi}}(WCW(P) = k) = \begin{cases} O(n^{-\frac{k(k-1)}{4}}) & \text{if } \exists \pi \in CH(\Pi) \text{ s.t. } WCW(\pi) \ge k \\ \exp(-\Omega(n)) & \text{otherwise} \end{cases}.$$

**Tightness of the upper bound.** *For any $1 \le k \le m$, there exist infinitely many $n \in \mathbb{N}$ such that:*

$$\sup_{\vec{\pi} \in \Pi^N} \Pr_{P \sim \vec{\pi}}(WCW(P) = k) = \begin{cases} \Omega(n^{-\frac{k(k-1)}{4}}) & \text{if } \exists \pi \in CH(\Pi) \text{ s.t. } WCW(\pi) \ge k \\ \exp(-O(n)) & \text{otherwise} \end{cases}.$$

*Proof.* **First step: defining $\mathcal{C}$.** Let $\mathcal{C} = \{C^G : WCW(G) = k\}$, where $C^G$ is defined in Definition 11. That is, $\mathcal{C}$ contains all $C^G$ where $G$ has exactly $k$ weak Condorcet winners. We have the following observations.

(1) For any profile $P$, $WCW(P) = k$ if and only if $\text{Hist}(P) \in \bigcup_{C^G \in \mathcal{C}} \mathcal{H}^G$. To see this, if $WCW(P) = k$ then $\text{Hist}(P) \in \mathcal{H}^{\text{UMG}(P)}$, where $\text{UMG}(P) \in \mathcal{C}$; and conversely, if $\text{Hist}(P) \in \mathcal{H}^G$, then $G = \text{UMG}(P)$ contains exactly $k$ weak Condorcet winners, which means that $WCW(P) = k$.

(2) by McGarvey's theorem [44], for each $C^G \in \mathcal{C}$, there exists a profile $P$ with $\text{UMG}(P) = G$, which means that $\mathcal{H}^G \ne \emptyset$.

(3) The total number of unweighted directed graphs over $\mathcal{A}$ only depends on $m$, which means that $|\mathcal{C}|$ can be seen as a constant that does not depend on $n$.

**Second step: the polynomial case.** To prove the upper bound, we note that for any $G$ that has exactly $k$ weak Condorcet winners, the number of ties is at least $\frac{k(k-1)}{2}$, which means that $\text{Rank}(\mathbf{E}^G) \ge \frac{k(k-1)}{2}$. Therefore, according to observation (1) above, we have:

$$\Pr_{P \sim \vec{\pi}}(WCW(P) = k) \le \sum_{G:C^G \in \mathcal{C}} \Pr_{P \sim \vec{\pi}}(\text{Hist}(P) \in \mathcal{H}^G) = O\left(n^{-\frac{k(k-1)}{4}}\right)$$

The last part follows after applying the polynomial upper bound in Lemma 1 to all $C^G \in \mathcal{C}$ and the observation (3) above.

To prove the tightness, suppose there exists $\pi \in \text{CH}(\Pi)$ with $\text{WCW}(\pi) \geq k$. This means that there exists a supergraph of $\text{UMG}(\pi)$ over $\mathcal{A}$, denoted by $G^*$, that has exactly $k$ weak Condorcet winners, and there is an edge from any weak Condorcet winner to any other alternative. By Claim 3, $\pi \in \mathcal{H}_{\leq 0}^{G^*}$ and $\text{Rank}(\mathbf{E}^{G^*}) = \frac{k(k-1)}{2}$. The tightness follows after applying the tightness of the polynomial bound in Lemma 1 to $\mathbf{C}^{G^*}$.

**Third step: the exponential case.** For any $\mathbf{C}^G \in \mathcal{C}$ and any $\pi \in \text{CH}(\Pi)$, we first prove that $\mathcal{H}_{\leq 0}^G \cap \text{CH}(\Pi) = \emptyset$. Suppose for the sake of contradiction there exist $\mathbf{C}^G \in \mathcal{C}$ and $\pi \in \mathcal{H}_{\leq 0}^G \cap \text{CH}(\Pi)$. It follows that $\text{UMG}(\pi)$ is a subgraph of $G$, which means that all weak Condorcet winners in $G$ must also be weak Condorcet winners in $\pi$. Because there are $k$ weak Condorcet winners in $G$, we have $\text{WCW}(\pi) \geq k$, which is a contradiction. The lower (respectively, upper) bound follows after applying Lemma 1 to an arbitrary (respectively, all) $\mathbf{C}^G \in \mathcal{C}$. $\qquad\square$

Consider the special case where $\pi_{\text{uni}} \in \text{CH}(\Pi)$. Notice that all edge weights in $\pi_{\text{uni}}$ are 0, which means that $\pi_{\text{uni}}$ satisfies all pairwise constraints $\text{Pair}_{a,b}$ defined in Definition 10. Consequently, for any $\mathbf{E}$ and $\mathbf{S}$ that only contain pairwise constraints, we have $\mathcal{H}_{\leq 0} \cap \text{CH}(\Pi) \neq \emptyset$. This observation leads to the following corollary of Proposition 6.

**Corollary 3.** *Let $\mathcal{M} = (\Theta, \mathcal{L}(\mathcal{A}), \Pi)$ be a strictly positive and closed single-agent preference model with $\pi_{\text{uni}} \in \text{CH}(\Pi)$. For any $1 \leq k \leq m$, any $n \in \mathbb{N}$, and any $\vec{\pi} \in \Pi^n$, we have $\text{Pr}_{P \sim \vec{\pi}}(\text{WCW}(P) = k) = O(n^{-\frac{k(k-1)}{4}})$. The bound is tight for infinitely many $n \in \mathbb{N}$ and corresponding $\vec{\pi} \in \Pi^n$.*

When $\mathcal{M}$ is neutral, we have $\pi_{\text{uni}} \in \text{CH}(\Pi)$. This is because for any $\pi \in \Pi$, the average of $m!$ distributions obtained from $\pi$ by applying all permutations is $\pi_{\text{uni}}$. Therefore, Corollary 3 applies to all neutral, strictly positive, and closed models including $\mathcal{M}_{\text{Ma}}^{[\varphi,1]}$ and $\mathcal{M}_{\text{Pl}}^{[\varphi,1]}$ for all $0 < \varphi \leq 1$.

**Proposition 7 (Smoothed likelihood of non-existence of weak Condorcet winners).** *Let $\mathcal{M} = (\Theta, \mathcal{L}(\mathcal{A}), \Pi)$ be a strictly positive and closed single-agent preference model.*

**Upper bound.** *For any $n \in \mathbb{N}$ and any $\vec{\pi} \in \Pi^n$, we have:*

$$\text{Pr}_{P \sim \vec{\pi}}(\text{WCW}(P) = 0) = \begin{cases} O(1) & \text{if } \exists \pi \in \text{CH}(\Pi) \text{ and } G \supseteq \text{UMG}(\pi) \text{ s.t. } \text{WCW}(G) = 0 \\ \exp(-\Omega(n)) & \text{otherwise} \end{cases}$$

**Tightness of the upper bound.** *There exist infinitely many $n \in \mathbb{N}$ such that:*

$$\sup_{\vec{\pi} \in \Pi^N} \text{Pr}_{P \sim \vec{\pi}}(\text{WCW}(P) = 0) = \begin{cases} \Omega(1) & \text{if } \exists \pi \in \text{CH}(\Pi) \text{ and } G \supseteq \text{UMG}(\pi) \\ & \text{s.t. } \text{WCW}(G) = 0 \\ \exp(-O(n)) & \text{otherwise} \end{cases}$$

*Proof.* **First step: defining $\mathcal{C}$.** Let $\mathcal{C} = \{\mathbf{C}^G : \text{WCW}(G) = 0\}$, where $\mathbf{C}^G$ is defined in Definition 11. That is, $\mathcal{C}$ contains all $\mathbf{C}^G$ where $G$ has no weak Condorcet winners. We have the following observations.

(1) For any profile $P$, $\text{WCW}(P) = 0$ if and only if $\text{Hist}(P) \in \bigcup_{\mathbf{C}^G \in \mathcal{C}} \mathcal{H}^G$. To see this, if $\text{WCW}(P) = 0$ then $\text{Hist}(P) \in \mathcal{H}^{\text{UMG}(P)}$, where $\text{UMG}(P) \in \mathcal{C}$; and conversely, if $\text{Hist}(P) \in \mathcal{H}^G$, then $G = \text{UMG}(P)$ does not contain a weak Condorcet winner, which means that $\text{WCW}(P) = 0$.

(2) $\mathcal{H}^G \neq \emptyset$ due to McGarvey's theorem [44].

(3) $|\mathcal{C}|$ can be seen as a constant that does not depend on $n$.

**Second step: the polynomial case.** The upper bound trivially holds. To prove the tightness, suppose there exit $\pi \in \text{CH}(\Pi)$ and a supergraph $G$ of $\text{UMG}(\pi)$ that does not contain a Condorcet winner. Let $G^*$ denote an arbitrary complete supergraph of $G$. It follows that $G^*$ is a supergraph of $\text{UMG}(\pi)$ and $G^*$ does not contain a Condorcet winner, which means that $\mathbf{C}^{G^*} \in \mathcal{C}$. By Claim 3, $\pi \in \mathcal{H}_{\leq 0}^{G^*}$, which means that $\mathcal{H}_{\leq 0}^{G^*} \cap \text{CH}(\Pi) \neq \emptyset$. Note that $\mathbf{E}^{G^*} = \emptyset$, which means that $\text{Rank}(\mathbf{E}^{G^*}) = 0$. The tightness follows after applying the tightness of the polynomial bound in Lemma 1 to $\mathbf{C}^{G^*}$.

**Third step: the exponential case.** For any $\mathrm{C}^G \in \mathcal{C}$ and any $\pi \in \mathrm{CH}(\Pi)$, we first prove that $\mathcal{H}^G_{\leq 0} \cap \mathrm{CH}(\Pi) = \emptyset$. Suppose for the sake of contradiction that there exist $\mathrm{C}^G \in \mathcal{C}$ and $\pi \in \mathcal{H}^{\overline{G}}_{\leq 0} \cap \mathrm{CH}(\Pi)$, which means that $\mathrm{WCW}(G) = 0$. By Claim 3, $\mathrm{UMG}(\pi)$ is a subgraph of $G$, which contradicts the assumption of the exponential case, that all supergraphs of $\mathrm{UMG}(\pi)$ contains at least one weak Condorcet winner. The upper bound (respectively, its tightness) follows after applying the exponential bound (respectively, its tightness) in Lemma 1 to all (respectively, an arbitrary) $\mathrm{C}^G \in \mathcal{C}$. $\qquad\square$