[Reviews · NeurIPS 2020]

Review 1

Summary and Contributions: This paper introduces a smoothed analysis framework to analyze various impossibility results from social choice theory in large-scale settings i.e. when the number of voters is large. Compared to the existing framework of domain restrictions or likelihood analysis, I find the smoothed analysis framework to be quite exciting because it takes a worst average-case approach to understand various impossibility theorems. The framework does not have to limit the set of possible preferences, nor it has to fix a particular distribution. In this framework, the authors characterized the smoothed likelihood of several social choice phenomena, including the likelihood of non-existence of Condorcet cycles. They also proved a smoothed version of a folklore impossibility theorem which says that a voting rule cannot simultaneously satisfy both anonymity and neutrality. The authors then show that under popular tie-breaking scheme, this probability vanishes to zero, but at a slow rate for scoring rules. Then they propose a novel tie-breaking rule, which coupled with a scoring rule, shows the exponentially small probability of the impossibility theorem. Overall, I think the paper proposes a great framework to improve our understanding of various social choice impossibility results. The general template can be useful in proving the smoothed likelihood of other impossibility theorems in social choice. 


Strengths: I think the main strength of the paper is its technique. The authors provide a generic template based on the satisfaction of linear constraints. The main motivation behind this template is that many phenomena in social choice are effectively characterized by the histogram of rankings, where the histogram just records the number of occurrences of each ranking in a profile. Once we focus on the histogram, an event can be formulated as the satisfaction of a set of linear constraints. Based on the set of distributions over the histograms and the solution set of the system of constraints, the authors prove matching upper and lower bound for the probability of satisfying the system of constraints.

Weaknesses: I see one weakness in the noise model. The underlying ground truth profile is adversarially chosen, but the authors assume that conditioned on the ground truth, the noise of the voters are iid drawn. Even though this is usually the case in smoothed analysis of various TCS problems, it might not be realistic for the social choice setting, particularly if the voters are situated on a network. But I believe such extensions can be handled in another paper. 


Correctness: I haven’t read the proof fully in the supplementary material, but I think the claims are correct.

Clarity: Yes, the paper is very well-written. Even though I am not familiar with the smoothed analysis framework, I understood the main results and proof techniques quite easily. Also they did a great job in highlighting the importance of smoothed analysis framework in the introduction.


Relation to Prior Work: To the best of my knowledge, this paper is the first one to introduce the smoothed analysis framework in the context of social choice. So this paper builds a new framework entirely from scratch. But they did cover other approaches to characterize impossibility results in social choice theory and highlight the need of the smoothed analysis framework.

Reproducibility: Yes

Additional Feedback: In theorem 1, the required condition says that UMG(\pi) does not contain a weak Condorcet cycle. Is this easy to check if we just have sample access to the preferences, instead of the entire distribution? Additionally, is it always true for standard noise models like Mallows or Plackett-Luce model? -----After Rebuttal------ Thanks for the rebuttal! I am happy with the answers to my question. Overall, I think the paper makes interesting contributions to social choice theory and recommend accepting the paper.


Review 2

Summary and Contributions: The authors introduce smoothed analysis in social choice, specifically, in voting. Traditional approaches to circumvent classic impossibility results (such as Condorcet's paradox or the simple impossibility that no rule can be simultaneously anonymous and neutral) by showing that the likelihood of bad events happening is exponentially small when votes are coming from a distribution. The paper avoids making parametric distributional assumptions by deriving more general results that apply to a broad family of distributions. Using a new technical lemma, the authors show that Condorcet's paradox either happens with at least a constant probability or exponentially rarely, depending on whether the underlying distribution over the votes is paradoxical or not. They also show that there exist rules that satisfy both anonymity and neutrality with high probability. They also show how to construct such rules by considering positional scoring rules with a novel tie-breaking scheme; in contrast, the authors show that using conventional tie-breaking schemes does not work.

Strengths: The paper has several strengths. - The results are novel (to the best of my knowledge) and important. - The new technical lemma is certainly very useful and seems to have broad applications beyond social choice. While I am no expert in this area, I was not able to find this result with a 10-minute google scholar search. - The authors are able to establish tight bounds on the limiting probability, which is nice.

Weaknesses: There are two major weakness. - The paper has a loose fit with NeurIPS. While there is a statistical element, there is no learning element in the paper. But recently, NeurIPS has been expanding in its scope, and statistical social choice has had increased representation in NeurIPS. So perhaps this is a minor weakness. - A major weakenss is that I think Theorem 1 was effectively already known (see additional feedback). This reduces the contribution of the paper as Theorem 1 is one of three main theorems. That said, the usefulness of the lemma is incontrovertible as I do not see Theorems 2 or 3 being provable with conventional techniques.

Correctness: - I did not check the proofs, which are in the appendix.

Clarity: - While there are spots where the authors could have improved clarity, the authors have done a fantastic job writing the paper clearly and formally, especially given the limited space and the vast number of concepts involved in the paper.

Relation to Prior Work: - This is very well laid out. For the last note on distortion, you may wish to point out that distortion is a worst-case analysis, and analyzing "smoothed distortion" (expected distortion for profiles near an arbitrary profile) seems to be an interesting open direction.

Reproducibility: Yes

Additional Feedback: Major comments: 1) Smoothed analysis: Calling your analysis "smoothed analysis" is a bit confusing. Smoothed analysis would take worst-case over profiles, and then expectation over noise added. But as you say in your related work, you're taking worst-case over distributions coming from a family. Such analysis was already done in the past. For example, consider work that analyzed the probability of Condorcet's paradox under any distribution from the Mallows family. Of course, your work replaces this narrow family with a much broader *nonparametric* family, and due to that, is strictly more general than smoothed analysis (because reasonable noise added to any profile would satisfy your assumptions). So this is not an objection to your technical contribution. While it is probably too much to change the whole paper and position it this way, perhaps you can add a paragraph pointing this out. It is also worth remarking that generalizing from narrow parametric families (e.g. Mallows' model) to broad nonparametric families is also not a novel contribution. The following papers have attempted this in the past, for example. However, these papers are on the "objective" setting with a ground truth, whereas your work introduces this approach in the "subjective" setting. I view this as the right way to define the novelty of your contribution. "When do noisy votes reveal the truth?" Caragiannis et al., TEAC 2016 "Simple, Robust and Optimal Ranking from Pairwise Comparisons", Shah and Wainwright, JMLR, 2018 2) Theorem 1: In Theorem 3.8 of the TEAC paper cited above, the authors show that given O(ln(m/\epsilon)) votes from Mallows' model, UMG becomes acyclic w.p. >= 1-\epsilon. In other words, given n votes, it becomes acyclic w.p. = 1-m*exp(-\Omega(n)). While they state their result only for the Mallows' model, their proof only uses the assumption that if P[a>b]-P[b>a] > 0, then it is at least some fixed constant. This is true in your model because you don't allow the distributions to change with n. Hence, I believe that your Theorem 1 was essentially already known. Nonetheless, I think capturing this argument in the "easy" case of your broader lemma, and then deriving the "hard" case (that uses the rank of the constraint matrix) is still a significant contribution. 3) You should perhaps acknowledge that your formulation of anonymity and neutrality as a "per profile" property is a bit nonstandard as these properties are usually thought of as either true (across all profiles) or false (if violated anywhere). For example, a rule that is not neutral and biased towards candidate X may not be adopted in practice even though on most profiles you may not see such a violation. I do not see this as a major downside because we already know that full neutrality is impossible, so this is the best possible positive result one can hope for. 4) Use of big-Oh notation: When you write O(n^{-rank/2}) or O(n^{-\ell_{\Pi}/2}), you should check if you're using the notation correctly, or whether the bounds are supposed to be O(n)^{-rank/2} and O(n)^{-\ell_{\Pi}/2}. For example, in L638 in the appendix, you have an exponent k outside of the \Omega notation, which, I think, would prevent you from writing \Omega(n_{\ell}^{-o/2}) and instead require you to write \Omega(n_{\ell})^{-o/2}? Minor comments: - Perhaps the introduction could be a bit gentler. For example, the a > b notation, Condorcet's paradox, or the mention of "agents" and "alternatives" are a bit abrupt and may be unfamiliar to a reader who does not work in social choice. - You mention cardinal utilities in our contributions, but the paper does not consider it. Please remove. - When you mention distortion in related work, it is worth noting that this is a worst-case approach, and looking at "smoothed distortion", where you consider the average distortion in profiles near an arbitrary profile, is an interesting direction. - L178: Mention that WMG is a complete graph. - L186: Which supergraph? You perhaps mean the one obtained by adding a -> b and b -> a edges when P[a>b]=P[b>a]. - L190: Have you defined \sigma(P) operation? - L196: It may be helpful to replace d with \epsilon because d is usually used for "dimension" (an integer > 1), whereas in your work, d \in (0,1) and usually very close to 0. This also prevents confusion later when you say that O((0.9 d n)^k) = O(n^k). - L197: What is the definition of limit you're using for a sequence of probability distributions to defined what "closed" means? Is this pointwise convergence? - L198: You use V here, but R before to denote a linear order. - L259: "namely" -> specifically - L262: Have you defined \vec{\pi} \cdot \vec{1}? You perhaps mean that by \vec{\pi}, you're also denoting the vector of fractional histograms of the distributions. Wasn't that supposed to be (Hist(\pi_1),...,Hist(\pi_n))? So shouldn't it be defined as Hist(\vec{\pi})? - L357: "ramking" -> ranking - It's a bit weird to have future work as a paragraph of section 5. When you have more space in camera ready, please make it a section. UPDATE AFTER REBUTTAL: I think the paradox part of Theorem 1 should be provable easily. Certainly, the strong edges of UMG remain whp. While each weak edge also persists wp 1/2, I understand that these are correlated. So yes, something like Berry-Essen may be needed. However, from a conceptual point of view, even stating the paradox part for when the UMG consists of a strong cycle would send the same message, but with a much simpler proof. So I still view Theorem 1 as not a significant contribution of the paper. I agree that smoothed analysis (worst-case underlying profile, stochastic noise on top of that) is a special case of your analysis (observed profile can come from any distribution from a wide family), as I already stated in my review. My point was that analysis of the style presented in the paper was already done in prior work, so calling it smoothed analysis seems to be trying to inflate the conceptual novelty of the paper. I had completely missed that you treat m as fixed. I believe this is only mentioned once in the paper. I would recommend emphasizing this a bit more (e.g. certainly after the first line of the model section).


Review 3

Summary and Contributions: This paper examines a type of smoothed analysis in social choice.

Strengths: +Gives improved analysis of the folklore impossibility theorem with certain tie breaking rules. + Well written +Develops a certain amount of mathematical machinery to solve the tasks at hand.

Weaknesses: Results are incremental: - Results (especially on Condorcet’s paradox) are very expected. So expected, that I don’t know why it is a good idea to apply smoothed analysis here. That is, Condorcet's only goes away when you choose very particular models over agent's signals that never violate certain contraints on the marginals. - While the basic results on the folklore impossibility theorem are also expected (by my intuition), the paper gives nice a bound there. However, I don’t think that this is sufficient to warrant a publication in this venue.

Correctness: Seems correct, but I did not read appendix, which wold be required to actually make this claim.

Clarity: yes

Relation to Prior Work: yes

Reproducibility: Yes

Additional Feedback: The paper’s results are too incremental to accept. While I think that the result on the folklore impossibility theorem is nice. It even gives an improvement for certain tie breaking rules. The results on condorcet’s theorem are very expected (given prior work) and not very surprising---at least at a high level. The exact rates of convergence are indeed contributions, but what things converge to is very expected. In fact, it is a rather silly thing to apply smoothed analysis too. The power of smoothed analysis is that if you just have a few bad inputs, it can smooth them away (like in the folklore impossibility theorem case). However, for Condorcet’s theorem, after smoothing, things look the same. So I do not understand the point. Other comments: The paper claims that worst case impossibility theorems are of little relevance given power of AI to learn users preference and make decision on their behalf. However, these results say that you cannot design ANY system that has some very reasonable properties. So, your fancy AI system will have to do stupid things sometimes, no matter how fancy. And the sometimes depends on the preferences of the agents, not anything about the system itself. Typoes: line 357 ******AFTER REBUTTAL********* >I didn't understand your comment about Condorcet's paradox remaining the same before and after the application of smoothed analysis. *What (I think) I meant was that UMG(\pi) is basically a marginal distribution for each pair. Then you violate condorcet with good probability if and only if you have a pi such that expectation of the marginals violates Condorcet. Note this is not at all what happens in smoothed analysis on LPs. Here you can have things that definitely do not work, however, with a small perturbation, they work with high probability. Nothing like that is happening in the Condorcet model. *(comment to other reviewers) *I had originally misread the paper. I thought they improved other's results on the Folk Theorem by using a new tie breaking rule. Instead, they improve the results of this very paper. So that makes more sense as a substantial contribution. I increased my score to reflect this. *(comment to other reviewers)* In any event, it seems my tastes in social choice problems are a little idiocycratic (not just this paper), and others here are more familiar with the literature than I am. So I am completely fine being overruled. I don't see anything "wrong" with this paper besides my taste not agreeing with it.


Review 4

Summary and Contributions: This paper uses smoothed analysis to better understand a classic paradox and impossibility result in social choice. The paper's conclusions are that even though the Condorcet paradox and the impossibility of having both neutrality and anonymity hold in the worst case, in general they are likely not a great concern.

Strengths: There is a lot to like about the paper. To the best of my knowledge it is the first significant attempt at using smoothed analysis for better understanding some of the impossibility results and paradoxes from social choice. This will open up future research in the area. The move away from the Impartial Culture assumption in the analysis is also an important contribution. While Imp Cul has made earlier analysis of social choice problems easier, it was always acknowledged as a highly unrealistic assumption. The tie-breaking rule will also be of broader interest to the social choice community, with practical ramifications. The Condorcet paradox and impossibility results align with people's intuitions. I think this is a good thing since now there is a principled framework to understand why many practitioners weren't losing a lot of sleep over Condorcet cycles.

Weaknesses: The main weakness of the paper is that it does not address Arrow's Theorem, since that is the big one. That said, I think it is entirely appropriate to have a paper like this setting up the framework and showing how it can be useful on some other problems before tackling the major question in the field. The presentation could certainly be improved -- particularly in Section 3 which is setting up the central technical result. The definitions were used to really introduce notation and I found them hard to parse. If more space was needed, then the discussion on Mallows and Plackett-Luce earlier in the paper could have been moved to an appendix, with just a short mention that the setup of the analysis encompasses standard preference models.

Correctness: The social choice aspects of the paper are technically correct. I am not an expert on smoothed analysis and so can not comment on that aspect.

Clarity: See my comment under "Weaknesses". The paper is dense in places and it would be possible to move some content to an appendix allowing for more space for the technical content.

Relation to Prior Work: To the best of my knowledge this is the first paper to use smoothed analysis to investigate Condorcet's paradox and this impossibility theory. I thought the authors did a thorough job at explaining how their work related to the other literature in the field, at least from the social choice perspective.

Reproducibility: Yes

Additional Feedback: I read the author's response. I remain supportive of this paper.

[Author Response · NeurIPS 2020]

We thank all reviewers for their helpful comments and suggestions! We will address all minor issues. Please see our responses to major issues below.

**R#1. Re. correlations among noise.** We totally agree with the reviewer that correlated noise is an important topic for future research. Actually our results only require independent (not necessarily identical) noise, which is a classical assumption in many literatures as briefly discussed in L126–127. We will add references in the revision. In the context of social networks, our framework can model the setting where agent's preferences are mostly determined by his/her neighbors except for a small random noise.

**R#1. Re. UMG($\pi$) has cycles.** We havn't thought about checking it from data, which is an interesting statistical hypothesis testing question for future research. For all neutral models such as Mallows and PL studied in this paper, we have $\pi_{uni} \in CH(\overline{\Pi})$ (L338), which means that UMG($\pi_{uni}$), which is the empty graph, contains a weak Condorcet cycle.

**R#2. Re. Theorem 1 is effectively known.** We totally agree with the reviewer that the smoothed avoidance can be proved by existing techniques without using Lemma 1. Thanks for the reference and we will discuss it in the revision. We are not sure if the smoothed paradox part of Theorem 1 can be proved by techniques in the TEAC paper (though it can be proved by multi-variate Berry-Esseen because its $O(\sqrt{n})$ error is small enough), and we'd greatly appreciate it if the reviewer can shed some light.

**R#2. Re. calling the proposed framework "smoothed analysis" is confusing.** Thanks for the very insightful comments! The "worst-case over profiles, and then expectation over noise added" part in smoothed complexity analysis is actually a special non-parametric model, which is a special case of our framework in the following sense: the parameters are all possible ground truth (e.g. all $n$-profiles), and the family of distributions are distributions over $n$-profiles obtained from adding noise to each ground truth—the setting where each agent chooses a (possibly different) distribution from single-agent Mallows (L199) is an example. In accordance with the reviewer's comment about subjective settings, we view the loss function in our framework (the per-profile satisfaction of axioms as the reviewer pointed out) the main conceptual novelty compared to previous work in preference learning, statistical/epistemic social choice (e.g. the JMLR paper and the TEAC paper, which adopt popular statistical loss functions that measures quality of decisions w.r.t. the ground truth, such as MSE and 0-1 loss), and smoothed complexity analysis (whose loss function is the runtime that does not depend on the ground truth). We call the proposed framework "smoothed" for better presentation, because (1) the idea behind smoothed complexity analysis is well accepted in CS, and (2) the conceptual innovation of our framework is similar to that of smoothed complexity analysis: while the modeling of uncertainty and the minimax nature of inference are standard in statistical decision theory, the smoothed (complexity or social choice) analysis provide principled ways and new results to resolve criticisms on average-case analysis in disciplines beyond statistics (algorithm design and social choice, respectively). We will expand the discussions about the second point in L132–134 in the revision, and will add more detailed and technical discussions in the full version.

**R#2. Re. distortion, Big-Oh.** Distortion is related to the single-agent PL (L204–207) whose ground truth is cardinal utilities and whose votes are rankings. $k \leq m$ is treated as a constant because all results are proved for fixed $m$. We'd certainly follow the reviewer's suggestions in the revision. Thanks so much for the suggestions!

**R#3. Re. "results are incremental".** We agree that some high-level messages behind our results align well with people's intuition, but like R#4, we view it a strength of the paper as for smoothed complexity analysis (the simplex algorithm was widely believed to be fast before it was formally proved by Spielman and Teng). We will make it clear in the revision. On a more technical level, it is unclear whether anything (impossibility and possibility) should be expected, because both hold under certain conditions. Our technical contribution is not to prove whether the impossibilities vanish or not, but is to characterize when do they vanish and at what rate. The non-triviality of such characterizations is evident in the vast literature on this topic and many open questions such as the conjecture in Section 4 of Tsetlin et al. [49], which we gave an asymptotic answer (see L140–141).

**R#3. Re. "basic results on the folklore impossibility theorem are also expected".** We are puzzled by the reviewer's comment because we are not aware of any similar study even under IC, as discussed in L142–143. We'd greatly appreciate it if the reviewer could elaborate on the reasoning and references behind his/her expectation.

**R#3. Re. "for Condorcet's theorem, after smoothing, things look the same".** If we are not mistaken, this is not true because (1) Condorcet paradox holds in the worst case, and (2) our Theorem 1 (L280–282) states that under certain conditions, Condorcet's paradox *disappears* after smoothing, which is different from before smoothing.

**R#3. Re. "the role of AI".** Application of AI is used to motivate the large-scale and frequently-used features, which naturally lead to the study of asymptotic satisfaction under realistic statistical models (please see L49–51). We did not mean to hint that AI can break mathematical facts and will make it clearer in the revision.

**R#4.** We totally agree that smoothed Arrow and other impossibility theorems are natural and important next steps! We will follow your suggestion to move Mallows and PL to the appendix to improve the presentation in the revision.

[Meta-Review · NeurIPS 2020]

The reviewers carefully considered your paper, and had a robust discussion about its merits. There was broad agreement that it has some very nice ideas, and is worthy of acceptance at NeurIPS this year. But please take a look at the details comments from the reviewers, particularly Reviewer 2, when composing your camera-ready.